# Landscape of Thoughts: Visualizing the Reasoning Process of Large Language Models

**Zhanke Zhou**[1,2*]   **Zhaocheng Zhu**[3,4*]   **Xuan Li**[1*]   **Mikhail Galkin**[6]
**Xiao Feng**[1]   **Sanmi Koyejo**[2]   **Jian Tang**[3,5]   **Bo Han**[1†]

[1]TMLR Group, Hong Kong Baptist University   [2]Stanford University
[3]Mila - Québec AI Institute   [4]Université de Montréal   [5]HEC Montréal   [6]Intel AI Lab

## Abstract

Numerous applications of large language models (LLMs) rely on their ability to perform step-by-step reasoning. However, the reasoning behavior of LLMs remains poorly understood, posing challenges to research, development, and safety. To address this gap, we introduce *landscape of thoughts* (LoT), the first landscape visualization tool to inspect the reasoning trajectories with certain reasoning methods on any multi-choice dataset. We represent the *textual states* in a trajectory as *numerical features* that quantify the states' distances to the answer choices. These features are then visualized in two-dimensional plots using t-SNE. Qualitative and quantitative analysis with the landscape of thoughts effectively distinguishes between strong and weak models, correct and incorrect answers, as well as different reasoning tasks. It also uncovers undesirable reasoning patterns, such as low consistency and high uncertainty. Additionally, users can adapt LoT to a model that predicts the property they observe. We showcase this advantage by adapting LoT to a lightweight verifier that evaluates the correctness of trajectories. Empirically, this verifier boosts the reasoning accuracy and the test-time scaling effect. The code is publicly available at: https://github.com/tmlr-group/landscape-of-thoughts.

## 1 Introduction

Large language models (LLMs) have revolutionized problem-solving. Many practical applications, *e.g.*, LLMs as agents (Schick et al., 2023; Lewis et al., 2020; Yao et al., 2023b), critically depend on step-by-step reasoning (Wei et al., 2022; Kojima et al., 2022). Despite progress in advanced models like OpenAI o1 (Jaech et al., 2024) and decoding methods such as test-time scaling (Snell et al., 2024), the underlying *reasoning behavior* of LLMs remains poorly understood, hindering the development of these models and posing deployment risks (Anwar et al., 2024).

A few pioneering attempts (Wang et al., 2023a; Saparov & He, 2023; Saparov et al., 2023; Dziri et al., 2024) probe LLM reasoning, but their insights often hinge on specific decoders and tasks. In practice, practitioners debug by *manually reading* the reasoning trajectories generated by LLMs, which has two drawbacks: (i) **scalability**, *i.e.*, human inspection does not scale (*e.g.*, at 30s per trajectory, 100 trajectories require 50min); and (ii) **aggregation**, *i.e.*, deriving reliable, dataset-level conclusions (*e.g.*, from 10,000 trajectories) is difficult, yielding subjective and even biased summaries. These costs compound during iterative development, where fast, interpretable feedback is essential. Consequently, there is a clear need for general, reusable tools to analyze LLM reasoning in users' own settings. This tool can potentially benefit *engineers* by accelerating iteration, *reasoning researchers* by informing decoder improvements, and *safety researchers* by monitoring model behaviors.

To this end, we introduce the *landscape of thoughts* (LoT), a visualization of LLM reasoning trajectories that delivers automatic, objective analysis from single examples to full datasets. Analogous to how t-SNE (Van der Maaten & Hinton, 2008) reveals structure in high-dimensional data, LoT exposes patterns in the reasoning space of LLMs. By pairing qualitative landscapes with quantitative metrics (consistency, uncertainty, and perplexity), LoT enables comparison and reveals insights beyond manual inspection or metric analysis.

---

[*]Equal Contribution.
[†]Correspondence to Bo Han (bhanml@comp.hkbu.edu.hk).

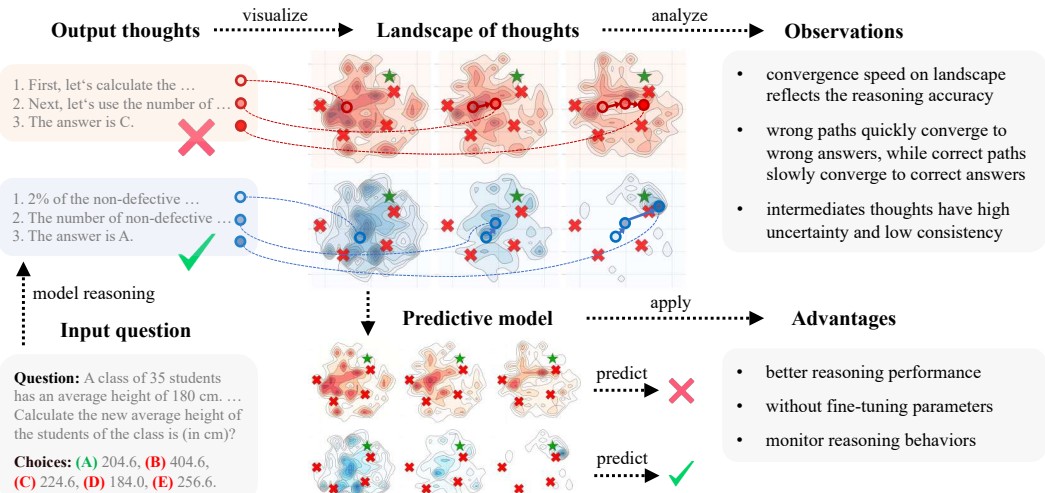

Figure 1: Landscape of thoughts for visualizing the reasoning steps of LLMs. Note that the red landscape represents wrong reasoning cases, while the blue indicates the correct ones. The darker regions in landscapes indicate more thoughts, with ✖ indicating incorrect answers and ★ marking correct answers. Specifically, given a question with multiple choices, we sample a few thoughts from an LLM and divide them into two categories based on correctness. We visualize the landscape of each category by projecting the thoughts into a two-dimensional feature space, where each density map reflects the distribution of states at a reasoning step. With these landscapes, users can intuitively discover the reasoning patterns of an LLM or a decoding method. In addition, a predictive model is applied to predict the correctness of landscapes and can help improve the accuracy of reasoning.

Specifically, given any multiple-choice reasoning dataset, LoT visualizes the distribution of intermediate states in any reasoning trajectories of interest *w.r.t.* the answer choices, which enables users to uncover reasoning patterns in both success and failure trajectories (Fig. 1). The core idea is to characterize the *states of textual thoughts* in a trajectory as *numerical features* that quantify the states' distances to the answer choices. These distances are estimated via the perplexity metric, using the same LLM that generates the thoughts. Then, these state features (i) produce three metric plots and (ii) are projected into a two-dimensional space with t-SNE to generate the landscape plots.

We examine LoT with different dimensions of model sizes, decoding methods, and reasoning datasets. LoT reveals several insightful observations regarding the reasoning behaviors of LLMs. Some notable observations include: 1) The convergence speed of trajectories towards correct answers reflects the accuracy, regardless of the base model, decoding method, or dataset; 2) The convergence speed of trajectories in success and failure cases differs markedly, indicating that we may use the convergence speed of a reasoning trajectory to predict its accuracy; 3) Low consistency and high uncertainty are generally observed in the intermediate thoughts, revealing the instability of the reasoning process. These patterns are uncovered by connecting per-trajectory inspection with dataset-level reasoning analysis, which is not reported by prior text inspection or metric analysis.

Since our tool is built on top of state features, it can be adapted to a machine-learning model to quantitatively predict certain properties, such as the findings mentioned above. We showcase this advantage by training a lightweight model to predict the success and failure cases, which is equivalent to verifiers commonly used in LLM reasoning (Cobbe et al., 2021). Despite being lightweight compared to typical LLM-based verifiers, it consistently improves reasoning performance across the majority of model, method, and dataset combinations in our experiments. Hence, users can leverage this framework to predict task-specific properties in their own settings.

In summary, our main contributions are three-fold:

- We introduce the first tool for automatic and scalable visualization of the LLM reasoning procedure, applicable to any open-source models and decoding methods on multiple-choice datasets (Sec. 2).
- Our tool reveals several new insights into the reasoning behaviors of different language models, decoding methods, and datasets (Sec. 3).
- Our tool can also be adapted into a predictive model to estimate certain properties and guide the reasoning process, improving LLM reasoning without modifying the model parameters (Sec. 4).

## 2 LANDSCAPE OF THOUGHTS

### 2.1 PROBLEM FORMULATION

Our goal is to *visualize* the reasoning trajectories of LLMs across a variety of task domains. Specifically, we target datasets consisting of *multiple-choice questions*, where each datapoint $(x, y, \mathcal{C})$ comprises a question $x$, a correct answer $y$, and a finite set of candidate choices $\mathcal{C} = \{c_j\}_{j=1}^k$, all represented in texts. [1] The visualization tool applies to the following models and methods.

**Language models.** To explore the landscape of thoughts generated by an LLM $p_{\text{LLM}}(\cdot)$, the model should produce diverse reasoning trajectories for solving a problem. In each trajectory, the reasoning thoughts are decoded autoregressively as $\hat{t}_i \sim p_{\text{LLM}}(t_i|x, \mathcal{C}, \hat{t}_1, \ldots, \hat{t}_{i-1})$: each thought $\hat{t}_i$ is conditioned on the question $x$, the candidate set $\mathcal{C}$, and the sequence of preceding thoughts $\hat{t}_1, \ldots, \hat{t}_{i-1}$. To characterize intermediate states within these trajectories, the LLM must also function as a likelihood estimator, enabling the computation of the probability $p_{\text{LLM}}(\hat{y}|x, \mathcal{C}, \hat{t}_1, \ldots, \hat{t}_i)$ of any answer $\hat{y}$. These two requirements are generally satisfied by open-source LLMs, such as Llama (Dubey et al., 2024) and DeepSeek (Liu et al., 2024). However, closed-source LLMs like GPT-4 (Achiam et al., 2023) and Gemini (Team et al., 2023) are excluded, as their likelihood estimation is not publicly supported.

**Reasoning methods.** While there are many approaches to solving reasoning problems with LLMs (Creswell et al., 2022; Kazemi et al., 2023), this work focuses on chain-of-thought (CoT) (Wei et al., 2022) and its derivatives (Zhou et al., 2023; Yao et al., 2023a), given their widespread adoption. These decoding methods generally guide the model in generating a structured trajectory of intermediate reasoning thoughts before arriving at the final answer. Note that to visualize a large number of reasoning thoughts effectively, these thoughts should be automatically parsed into distinct units (*e.g.*, via sentence tokenization). This requirement can be satisfied by most LLMs. [2]

### 2.2 QUALITATIVE VISUALIZATION WITH LANDSCAPES

Given a collection of reasoning trajectories generated by an LLM, [3] our tool seeks to visualize, within a two-dimensional (2D) space, how different trajectories lead to either correct or incorrect answers, as illustrated in Fig. 1. A key challenge lies in the absence of a direct mapping from the *textual* space of thoughts to *numerical* 2D coordinates. To address this gap, we utilize the same LLM to represent intermediate states as numerical features. These state features are then projected into a 2D space for visualization. For simplicity, we denote a thought as $t_i$ instead of $\hat{t}_i$.

**Characterizing the states.** Here, the intermediate *thoughts* $\{t_i\}_{i=1}^n$ in a reasoning trajectory naturally define a sequence of *states* $\{s_i\}_{i=0}^n$, where $s_0 = [x]$ and $s_i = [x, t_1, t_2, \ldots, t_i]$. We characterize the states as features using the likelihood function of the LLM. Specifically, the $k$-dim feature $\boldsymbol{f}_i$ for state $s_i$ indicates the relative distances from the state $s_i$ to all possible choices $\{c_j\}_{j=1}^k$:

$$\boldsymbol{f}_i \triangleq [d(s_i, c_1), d(s_i, c_2), \ldots, d(s_i, c_k)]^\top, \tag{1}$$

where $d(s_i, c_j)$ measures the *distance* between state $s_i$ and choice $c_j$. To normalize the token lengths across choices, we calculate $d(s_i, c_j)$ through the perplexity metric (Shannon, 1948; Manning & Schutze, 1999):

$$d(s_i, c_j) \triangleq \exp\left(-\frac{1}{|c_j|}\sum_{t=1}^{|c_j|} \log p_{\text{LLM}}(c_j[t]|s_i, c_j[: t])\right) = p_{\text{LLM}}(c_j|s_i)^{-1/|c_j|}, \tag{2}$$

where $|c_j|$ is the number of tokens in $c_j$, and $p_{\text{LLM}}(c_j|s_i)$ is the accumulated probability in an autoregressive manner. Assuming $|c_j| = T$, we have $p_{\text{LLM}}(c_j|s_i) = p_{\text{LLM}}(c_j[1]|s_i) \cdot p_{\text{LLM}}(c_j[2]|s_i, c_j[1]) \cdot p_{\text{LLM}}(c_j[3]|s_i, c_j[1], c_j[2]) \ldots p_{\text{LLM}}(c_j[T]|s_i, c_j[1], c_j[2] \ldots c_j[T-1])$. The token-level probabilities are normalized over the entire vocabulary; $c_j[1]$ is the first token of $c_j$, and $c_j[T]$ is the last token.

We normalize the vector $\boldsymbol{f}_i$ to have a unit $\ell_1$ norm. Additionally, we encode each choice as a landmark feature vector for visualization. Notably, the perplexity decreases as the model's prediction confidence increases. To align with this observation, we define the feature vector $\boldsymbol{f}_j^c$ for a choice $c_j$ as:

$$\boldsymbol{f}_j^c \triangleq \frac{1}{k}[\mathbb{1}(j \neq 1), \ldots, \mathbb{1}(j \neq k)]^\top. \tag{3}$$

---

[1] LoT is positioned for multi-choice questions. Appendix E.11 discusses its extension to open-ended tasks.

[2] We empirically verify the robustness of LoT if this requirement does not hold (please see Appendix H.9).

[3] Our method provides post-hoc analyses; it never intervenes or alters the model's reasoning trajectory.

For $r$ trajectories, each with $n$ states, we compute the feature vectors for all $r \cdot n$ states. Here, the zero entry in $\boldsymbol{f}_j^c$ indicates zero distance to the choice $c_j$ itself, and the nonzero entries (each equal to $1/k$) encode the assumption that distances among different choices are equal. [4] Together with the feature vectors of $k$ choices, we obtain a feature matrix $\boldsymbol{F} \in \mathbb{R}^{k \times (r \cdot n + k)}$ as:

$$\boldsymbol{F} \triangleq [\boldsymbol{f}_1^{(1)}, \ldots, \boldsymbol{f}_n^{(1)}, \ldots, \boldsymbol{f}_1^{(r)}, \ldots, \boldsymbol{f}_n^{(r)}, \boldsymbol{f}_1^c, \ldots, \boldsymbol{f}_k^c]. \tag{4}$$

Note that a sufficiently large number of trajectories is necessary to generate a comprehensive visualization of the reasoning landscape. For computational efficiency, we sample $d$ trajectories per question across all questions, yielding $r = d \times N_q$ total trajectories, where $N_q$ is the number of questions. We then normalize feature vectors by reordering choices so the correct answer appears in the first dimension across all questions. In this way, we can visualize the landscape of multiple questions by putting their trajectories together, which is more efficient than visualizing by generating enough trajectories for one question.

**Visualization.** After constructing the feature matrix $\boldsymbol{F}$, we project the states and choices into a 2D space for visualization. This step can be performed using various existing dimensionality reduction methods (Pearson, 1901; Van der Maaten & Hinton, 2008; McInnes et al., 2018). We employ t-SNE (Van der Maaten & Hinton, 2008) due to its effectiveness in preserving local neighborhood structure from the original high-dimensional space and its robustness to a wide range of transformations. [5] By applying t-SNE to the $k$-dim $\boldsymbol{F}$, we obtain the 2D coordinates $\bar{\boldsymbol{F}} \in \mathbb{R}^{2 \times (rn+k)}$. The two projected dimensions correspond to directions in the original answer space; each state's 2D coordinates thus reflect its relative distance to different answers. Finally, the coordinates of the states define a discrete density function in the 2D space, represented by the color depth in landscapes.

### 2.3 QUANTITATIVE VISUALIZATION WITH METRICS

Beyond the qualitative visualization, we introduce three quantitative metrics to help understand the LLMs' behavior. These metrics are defined based on the intermediate states in Sec. 2.2.

**Consistency.** To understand whether the LLM knows the answer before generating all thoughts, we compute the consistency of state $s_i$ by checking whether $\boldsymbol{f}_i$ and $\boldsymbol{f}_n$ agree:

$$\text{Consistency}(s_i) = \mathbb{1}(\arg\min \boldsymbol{f}_i = \arg\min \boldsymbol{f}_n). \tag{5}$$

**Uncertainty.** To know how confident the LLM is about its predictions at intermediate steps, we compute the uncertainty of state $s_i$ as the entropy of $\boldsymbol{f}_i$ (note $\sum_{d \in \boldsymbol{f}_i} d = 1$)

$$\text{Uncertainty}(s_i) = -\sum_{d \in \boldsymbol{f}_i} d \cdot \log d. \tag{6}$$

**Perplexity.** We also measure the LLM's confidence in its generated thoughts using perplexity, which is comparable across thoughts of different lengths:

$$\text{Perplexity}(t_i) = p_{\text{LLM}}(t_i | s_{i-1})^{-1/|t_i|}. \tag{7}$$

**Remark 2.1.** While perplexity is traditionally used at the token level for language modeling, we apply it at the thought level. Additionally, we introduce consistency and uncertainty metrics tailored to reasoning trajectories, offering a new perspective for the community. Appendix F introduces related works in detail. The following section demonstrates that the LoT, containing the qualitative landscape and the quantitative metrics, is effective for automatic and scalable visualization of reasoning trajectories.

## 3 RESULTS AND OBSERVATIONS

In this section, we utilize the landscape of thoughts to analyze the reasoning behavior of LLMs by comparing the visualizations across three dimensions: (1) diverse scales and types of *language models* in Sec. 3.1, (2) different *reasoning tasks* in Sec. 3.2, and (3) various *reasoning methods* in Sec. 3.3. Unless stated otherwise, we employ Llama-3.1-70B with CoT as the default configuration in evaluations. All the visualizations are built upon the model's estimation of its own thoughts. [6]

---

[4]LoT can be applied to trajectories with different numbers of states. We assume $n$ states for demonstrations.

[5]Appendix H.8 shows that LoT is compatible and robust with different methods of dimensionality reduction.

[6]Appendix H.1 validates each qualitative observation from LoT. Full visualizations are in Appendix I.

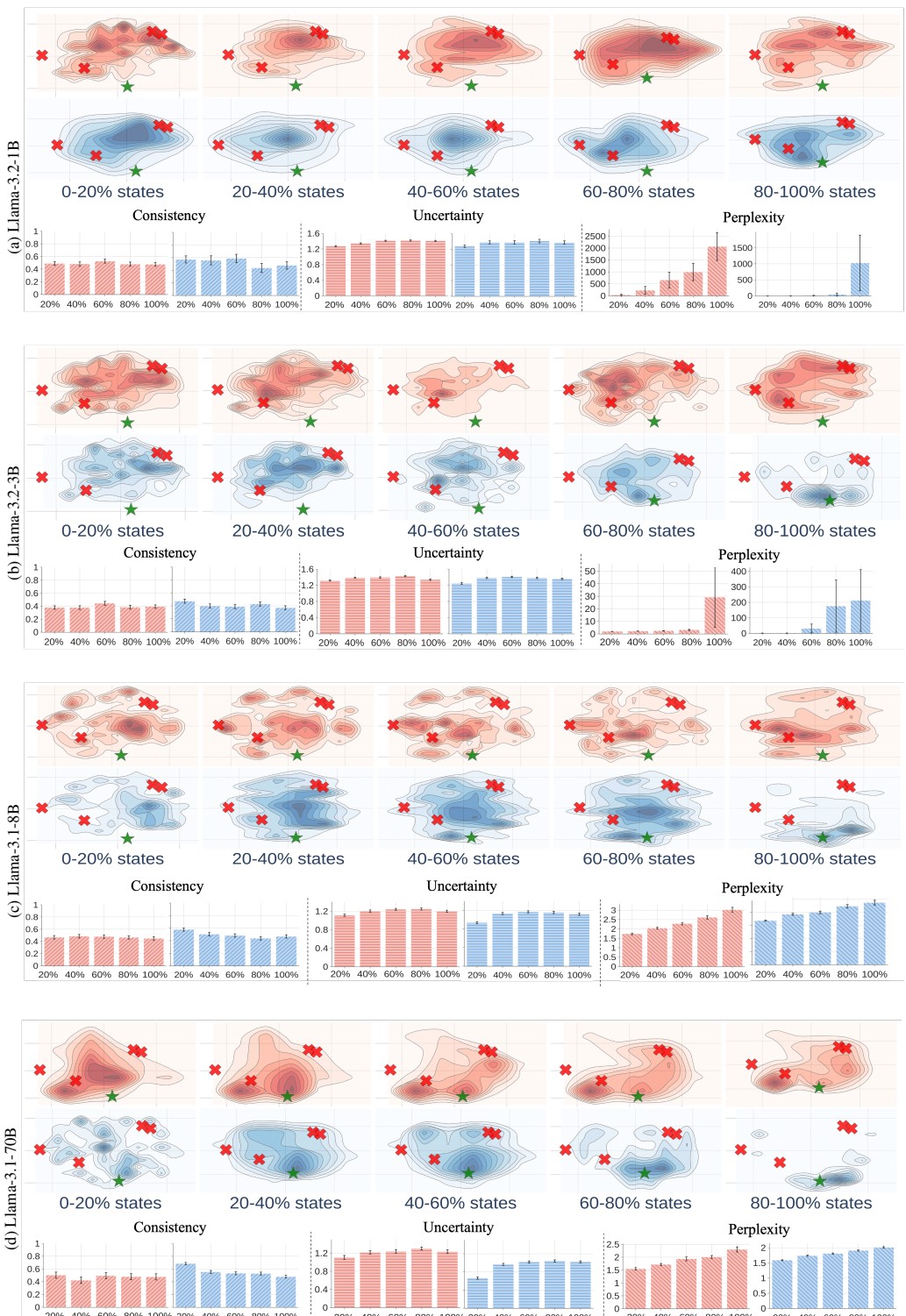

Figure 2: Comparing the LoT of different language models (with CoT on the AQuA dataset). Darker regions represent higher state density, with ✖ indicating incorrect answers and ★ marking the correct ones. Through the reasoning trajectories, spanning from early (0-20% states) to the later stages (80-100% states), the visualization shows correct cases (bottom rows in blue) with incorrect cases (top rows in red). Metrics are calculated *w.r.t.* each bin, *e.g.*, 20% - 40% of states. The reasoning accuracy of the four subfigures is: (a) 15.8%, (b) 42.0%, (c) 53.2%, and (d) 84.4%.

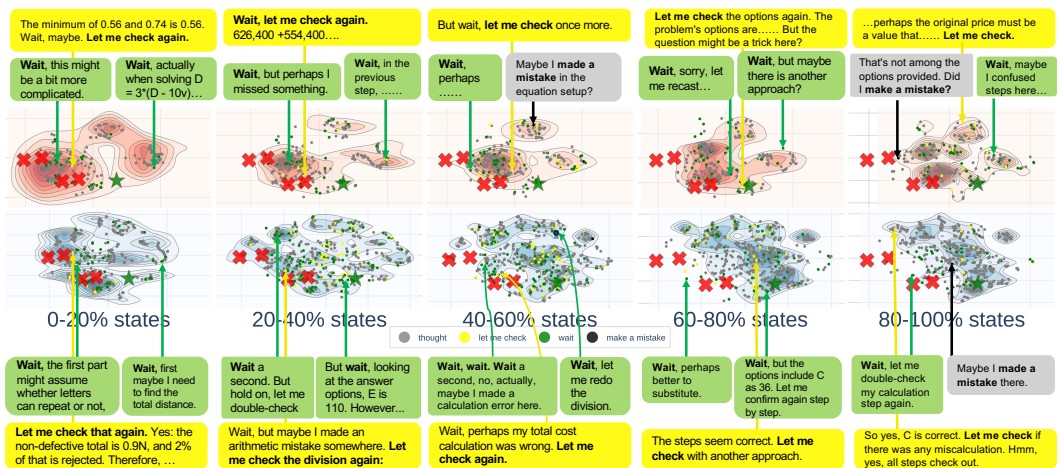

Figure 3: The LoT of the reasoning model QwQ-32B (using CoT prompting on the AQuA dataset).

## 3.1 COMPARISON ACROSS LANGUAGE MODELS

We study several LLMs' behavior across parameter scales (from 1B, 3B to 70B). We run each model with CoT prompting on 50 randomly selected problems from the mathematical reasoning dataset AQuA. Their landscapes are shown in Fig. 2, from which we have the following observations. [7]

**Observation 3.1** (*The landscape converges faster as the model size increase*). As model parameters scale from 1B to 70B, the corresponding landscape demonstrates faster convergence to the correct answers with higher density in the last 20% states, aligning with the increasing accuracy. With more parameters, larger models can store broader knowledge (Allen-Zhu & Li, 2024). This leads to more confident solutions, demonstrated by more focused answer patterns and lower uncertainty.

**Observation 3.2** (*Larger models have higher consistency, lower uncertainty, and lower perplexity*). As the model size increases, the consistency increases; at the same time, the uncertainty and perplexity decrease significantly. This also aligns with the higher accuracy for the large models. [8]

In addition, we apply LoT to the recent reasoning model QwQ-32B (Team, 2025) and observe:

**Observation 3.3** (*Reasoning models present more-complex reasoning behaviors in landscapes.*). In Fig. 3, the landscapes can capture complex reasoning patterns such as self-evaluation and self-correction. Specifically, correct trajectories tend to include more instances of self-evaluation and self-correction compared to incorrect ones. These behaviors often occur early in the reasoning process, when the model is far from the correct answer. Compared to non-reasoning models, correct trajectories here show greater diversity, with green and yellow points more widely scattered.

## 3.2 COMPARISON ACROSS REASONING TASKS

Besides the AQuA dataset, we include MMLU, CommonsenseQA, and StrategyQA datasets. We run the default model with CoT on 50 problems per dataset. These observations are derived from Fig. 4:

**Observation 3.4** (*Similar reasoning tasks exhibit similar landscapes*). The landscapes of AQuA, MMLU, and StrategyQA in Fig. 4 exhibit organized search behavior with higher state diversity, while CommonSenseQA presents concentrated search regions, reflecting direct retrieval of common-sense knowledge rather than step-by-step reasoning processes. These distinct landscape patterns demonstrate the potential to reveal underlying domain relationships across different reasoning tasks.

**Observation 3.5** (*Different reasoning tasks present significantly different patterns in consistency, uncertainty, and perplexity*). The histograms in Fig. 4 show that the perplexity consistently increases as reasoning progresses across all datasets. Specifically, datasets such as AQuA and MMLU show relatively higher levels of uncertainty compared to StrategyQA and CommonSenseQA. As for StrategyQA, correct trajectories show increasing consistency that surpasses incorrect trajectories at

---

[7]All claims are defined in the original answer distance space and visualized in 2D space (see Appendix E.7.)

[8]Appendix H.3 presents additional analyses of the consistency metric: the consistency does not relate to the length of the trajectory. Tab. 5 in Appendix H.5 supports the validity of comparing perplexity across models.

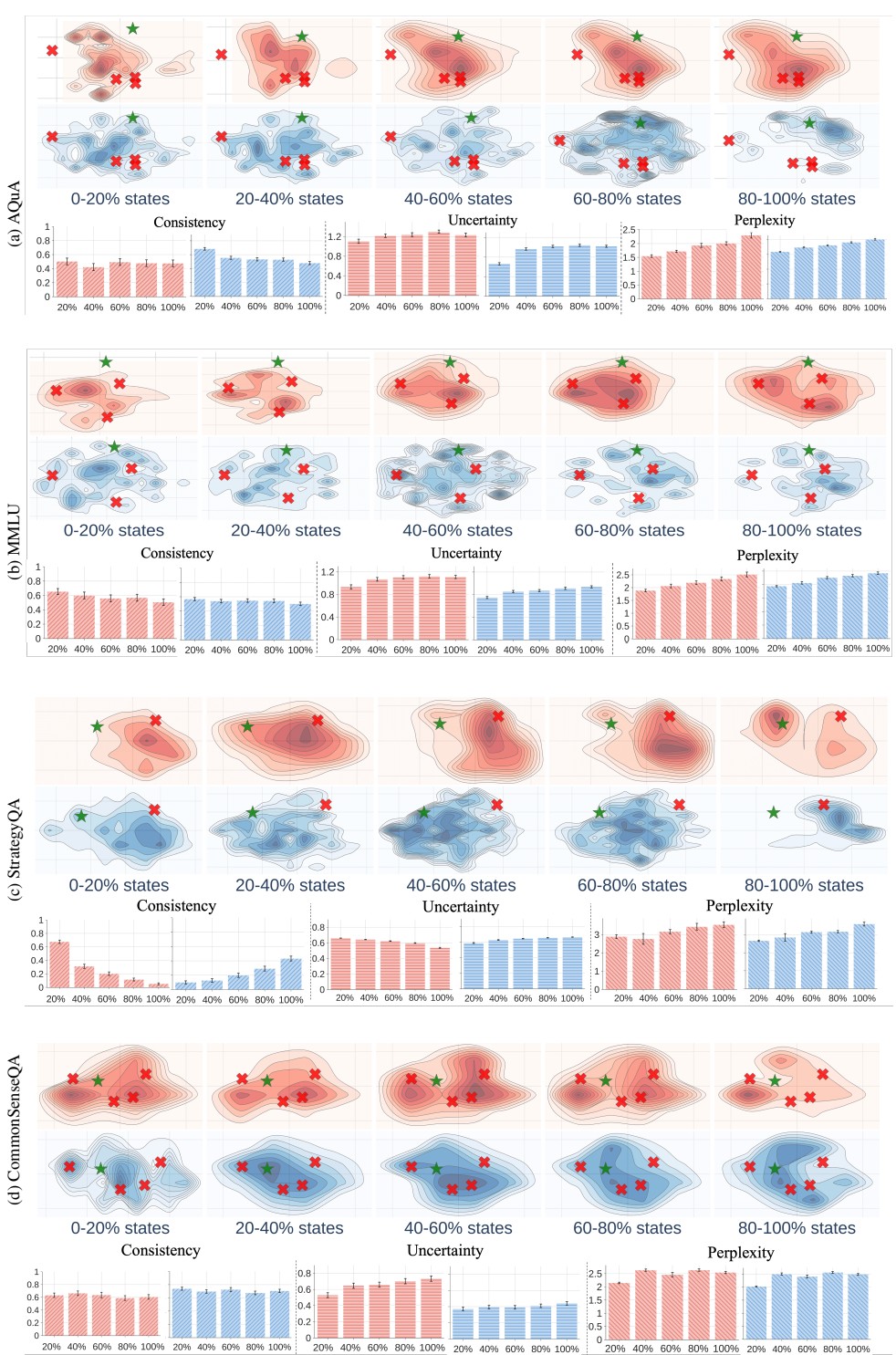

Figure 4: Comparing the LoT of different datasets (using Llama-3.1-70B with CoT). The accuracy of reasoning for the four subfigures is: (a) 84.4%, (b) 80.2%, (c) 75.8%, and (d) 64.8%.

around 60% states, while incorrect trajectories show decreasing consistency. However, when the trajectory is longer than the ground truth trajectory, the later stages (60-100% of states) exhibit both increasing perplexity and decreasing uncertainty. [9]

---

[9]We show detailed analysis for trajectories in StrategyQA in Appendix H.4.

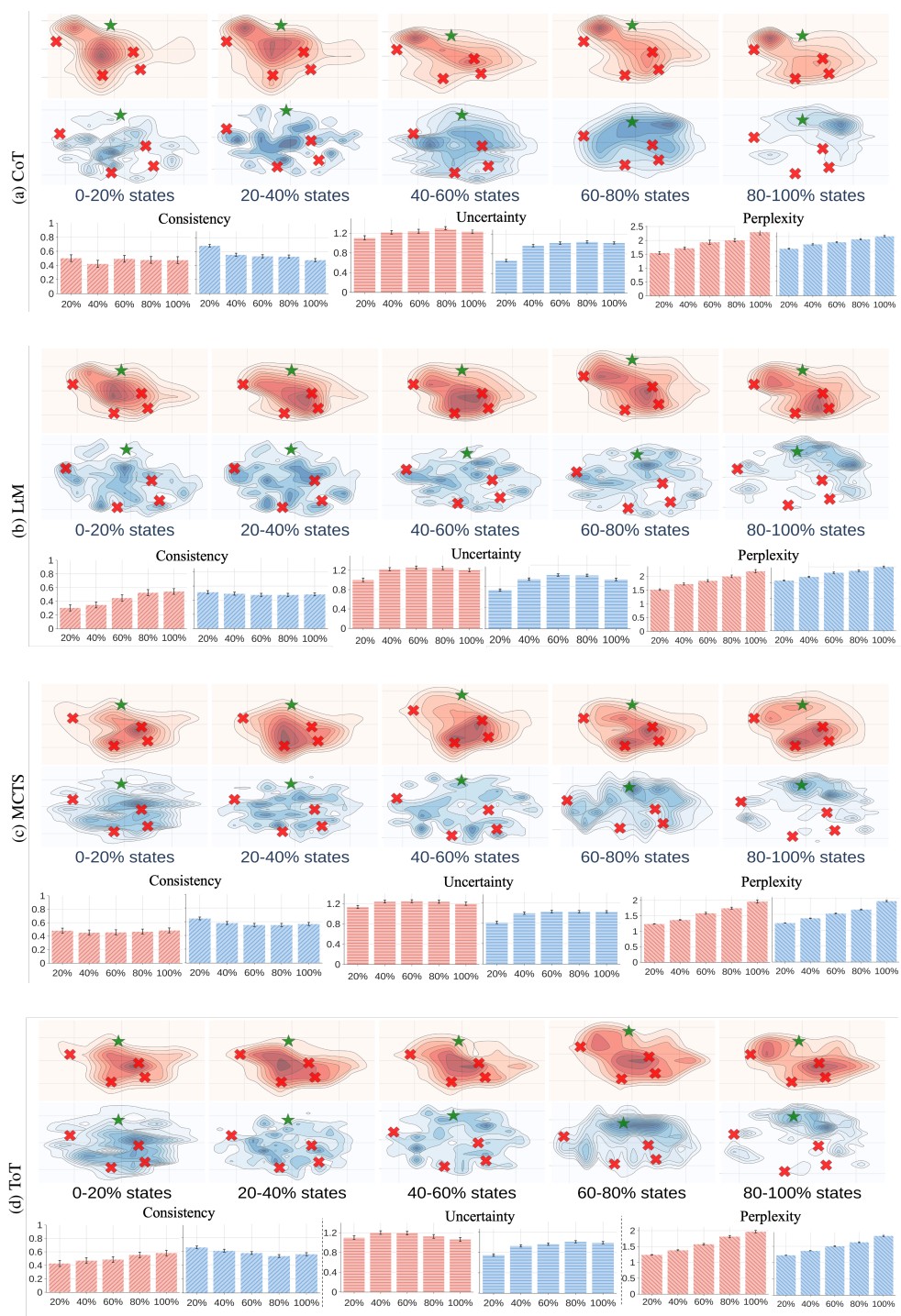

Figure 5: Comparing the LoT of four reasoning methods (using Llama-3.1-70B on the AQuA dataset). The reasoning accuracy is: (a) 84.4%, (b) 82.2%, (c) 75.8%, and (d) 81.6%, respectively.

## 3.3 COMPARISON ACROSS REASONING METHODS

**Setup.** We evaluate the default model with four reasoning methods: chain-of-thought (CoT) (Wei et al., 2022), least-to-most (LtM) (Zhou et al., 2023), MCTS (Zhang et al., 2024), and tree-of-thought (ToT) (Yao et al., 2023a). We run these methods on 50 problems from AQuA and observe that:

**Observation 3.6** (*Cross-method comparison: Among correct reasoning trajectories, methods with faster convergence to correct answers achieve higher accuracy.*). From Fig. 5, we observe that the

states are widely dispersed at early stages and gradually converge to correct (or incorrect) answers in later stages. Here, convergence means the trend of a reasoning trajectory approaching one answer. Generally, methods with more scattered landscapes (that converge more slowly) present lower accuracy than those that converge faster. For example, the blue landscape in Fig. 5(a) converges faster than the blue landscapes in Fig. 5(c), and the former achieves higher accuracy than the latter.

**Observation 3.7** (*Within-method comparison: For any single method, incorrect trajectories converge faster to wrong answers than correct trajectories converge to right answers.*)**.** In Fig. 5, failure trajectories usually converge to the wrong answers at earlier stages of reasoning, e.g., 20-40% of states in Fig. 5(c). By contrast, the states in the success trajectories converge to the correct answers in the later 80-100% of states. This implies that early states of the reasoning process can lead to any potential answers (from a model perspective), while the correct answers are usually determined at the end of reasoning trajectories. In addition, Fig. 34 showcases the corresponding text of thoughts. [10]

**Observation 3.8** (*Compared to failure trajectories, the intermediate states in correct trajectories have higher consistency w.r.t. the final state*)**.** By comparing the consistency plots in Fig. 5, we find that the model generally has low consistency between the intermediate states and the final state. Notably, the consistency of wrong trajectories is significantly lower than that of correct trajectories. This implies that the reasoning process can be quite unstable. Even though decoding methods like CoT and LtM are designed to solve a problem directly (without exploration), the thoughts generated by these methods do not consistently guide the reasoning trajectory to the answer.

## 4 ADAPTING VISUALIZATION TO PREDICTIVE MODELS

One advantage of our method is that it can be adapted into a predictive model to predict any property users observe. Here, we show how to convert our method to a lightweight verifier for voting trajectories, following the observations in Sec. 3. Note that this methodology is not limited to verifiers. Users can use this technique to adapt the visualization tool to monitor the properties in their scenarios.

### 4.1 A LIGHTWEIGHT VERIFIER

Observations 3.7 and 3.8 show that the convergence speed and consistency of intermediate states can distinguish correct and wrong trajectories. Inspired by these observations, we build a model $g : \mathbb{R}^{(k+1) \times n} \to \{0, 1\}$ to predict the correctness of a trajectory based on the state features $\{\boldsymbol{f}_i\}_{i=1}^n$ and consistency metric $\{\text{Consistency}(s_i)\}_{i=1}^n$. The insight is that the state features, used to compute the 2-D visualization, encode rich location information of the states and can be used to estimate the convergence speed. Due to the small dimensionality of these features, we parameterize $g$ with a random forest (Breiman, 2001) to avoid overfitting. We use this model as a verifier to enhance LLM reasoning (Cobbe et al., 2021). Unlike popular verifiers (Lightman et al., 2024) that involve a moderately sized language model on textual thoughts, our verifier operates on state features and is quite lightweight. We train a verifier on thoughts sampled from the training split of each dataset and apply it to vote trajectories at test time. Given $q$ trajectories sampled by a decoding method, the final prediction is produced by a weighted majority voting:

$$\hat{y} = \arg\max_{c \in \mathcal{C}} \sum_{i=1}^{q} \mathbb{1}(\hat{y}^{(i)} = c) \cdot g(\{\boldsymbol{f}_i\}_{i=1}^n, \{\text{Consistency}(s_i)\}_{i=1}^n). \tag{8}$$

### 4.2 EXPERIMENTAL RESULTS

We evaluate our numerical verifier against an unweighted voting baseline (Wang et al., 2023b) with various models, decoding methods, and reasoning datasets. We report the accuracy here instead of the commonly used pass@k metric, which can be easily inflated by random guessing or exhaustive enumeration of candidates to obtain a high score. Detailed settings of experiments are in Appendix G. We also provide ablation studies on training the verifier and discuss and compare the variance of the verifier in Appendix H.6, and experiment on the scaling effect with different features in Appendix H.7.

**Effectiveness of the verifier.** We first compare our verifier against the unweighted voting baseline, each applied to 10 trajectories. As shown in Fig. 6, our verifier consistently enhances the reasoning performance of all models and decoding methods, even though our verifier does not use any pre-trained language model. Notably, smaller language models (1B and 3B) show significant performance

---

[10]In Appendix H.2, only a few incorrect trajectories (1.8%) are close to the correct answer in middle thoughts.

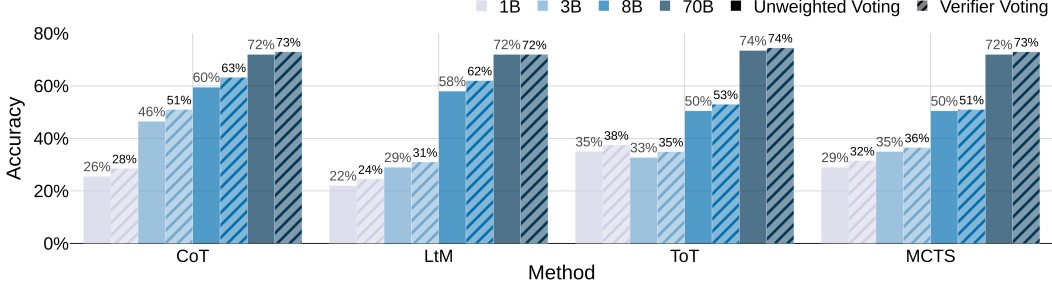

Figure 6: The accuracy of reasoning under different decoding methods and model scales (averaging across all four datasets). Results for each dataset are in Appendix I.

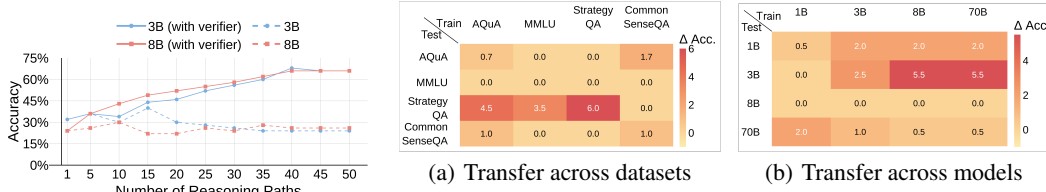

(a) Transfer across datasets      (b) Transfer across models

Figure 7: Demonstration of the inference-time scaling effect of the verifier. We show the voting accuracy (%) on StrategyQA scales with the number of trajectories.

Figure 8: Absolute accuracy changes ($\Delta$ Acc) with the verifier, compared to performance in Fig. 6 (without the verifier). The verifier is trained on each column (dataset or model) and evaluated on all rows (other datasets or models). Positive values indicate improvement in accuracy with the verifier.

gains with the verifier's assistance, achieving substantial improvements over their original reasoning capabilities. We also compare the verifiers across reward-guided methods.

**Test-time scaling.** While the improvement of the verifier seems marginal with 10 trajectories, our verifier can provide a substantial performance gain with more trajectories. We adjust the number of trajectories from 1 to 50, and plot the results of the verifier and the unweighted voting baseline in Fig. 7. Models with our verifier exhibit significantly stronger scaling behaviors, achieving over 65% accuracy. In contrast, the performance of the baseline saturates around 30% accuracy. These results suggest that our state features, which are used in both the visualization tool and the verifier, capture important information about the reasoning behavior of LLMs. Thus, the verifier can boost test-time scaling, especially in solving complex problems.

**Cross-dataset and cross-model transferability.** One interesting property of the state features and metrics is that their shape and range are independent of the model and dataset, suggesting that we may deploy the verifier trained on one dataset or model in another setting. As illustrated in Fig. 8, we evaluate how the verifier transfers across reasoning datasets (*e.g.*, train on AQuA and test on MMLU) and model scales (*e.g.*, train on 1B model and test on 70B model). We observe some positive transfers across datasets and models. For example, a verifier trained on AQuA can improve the performance of StrategyQA by 4.5%. A verifier trained on the 70B model also improves the performance of the 3B model by 5.5%. However, some cases do not benefit from the transferred verifiers. We leave improving the transferability of the state features and metrics as future work.

## 5 CONCLUSION

This paper introduces the landscape of thoughts, a visualization tool for analyzing the reasoning trajectories produced by large language models. Built on top of feature vectors of intermediate states in trajectories, our tool reveals several insights into LLM reasoning, such as the relationship between convergence and accuracy, and issues of low consistency and high uncertainty. Our tool can also be adapted to predict the answer of reasoning trajectories based on the observed property, which is demonstrated by a lightweight verifier developed based on the feature vectors and our observations for distinguishing the correctness of trajectories. We foresee that this tool will create several opportunities to monitor, understand, and improve the LLM reasoning.

ACKNOWLEDGEMENT

ZKZ, XL, XF, and BH were supported by RGC Young Collaborative Research Grant No. C2005-24Y, RGC General Research Fund No. 12200725, and HKBU CSD Departmental Incentive Scheme. SK was partially supported by NSF 2046795 and 2205329, IES R305C240046, ARPA-H, the MacArthur Foundation, Schmidt Sciences, Stanford HAI, RAISE Health, OpenAI, Microsoft, and Google.

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

APPENDIX

## A  ETHIC STATEMENT

The study does not involve human subjects, data set releases, potentially harmful insights, applications, conflicts of interest, sponsorship, discrimination, bias, fairness concerns, privacy or security issues, legal compliance issues, or research integrity issues.

## B  IMPACT STATEMENT

This work aims to advance the field of trustworthy machine learning and large language models, especially the interpretability of machine reasoning. Our work presents a tool for visualizing and understanding reasoning steps in LLMs. We foresee that our work will introduce more interpretability and transparency into the development and deployment of LLMs, advancing us toward more trustworthy machine learning. We do not find any negative societal consequences of our work.

## C  REPRODUCTION STATEMENT

The experimental setups for training and evaluation are described in detail in Appendix G.1, and the experiments are all conducted using public datasets. We provide the link to our source codes to ensure the reproducibility of our experimental results: https://github.com/tmlr-group/landscape-of-thoughts.

## D  LLM USAGE DISCLOSURE

This submission was prepared with the assistance of LLMs, which were utilized for polishing content and checking grammar. The authors assume full responsibility for the entire content of the manuscript. It is confirmed that no LLM is listed as an author.

## E  FURTHER DISCUSSIONS

### E.1  CHALLENGES IN ANALYZING LLM'S REASONING AUTOMATICALLY

Currently, the fundamental mechanisms behind both successful and unsuccessful reasoning attempts in LLMs remain inadequately understood. Traditional performance metrics, such as accuracy, provide insufficient insights into model behavior. While human evaluation has been employed to assess the quality of sequential thoughts (e.g., logical correctness and coherence), such approaches are resource-intensive and difficult to scale. We identify three challenges in developing *automated analysis systems* for LLMs' reasoning:

*Challenge 1: Bridging the token-thought gap.* Current explanatory tools, including attention maps (Clark et al., 2019; Kobayashi et al., 2020), probing (Alain & Bengio, 2016; Tenney et al., 2019; Hewitt & Liang, 2019), and circuits (Yao et al., 2024), primarily operate at the token-level explanation. While these approaches offer valuable insights into model inference, they struggle to capture the emergence of higher-level reasoning patterns from lower-level token interactions. Additionally, the

discrete nature of natural language thoughts poses challenges for traditional statistical analysis tools designed for continuous spaces. Understanding how thought-level patterns contribute to complex reasoning capabilities requires new analytical frameworks that can bridge this conceptual gap.

*Challenge 2: Analyzing without training data access.* Existing investigations into LM reasoning have predominantly focused on correlating test questions with training data (Ippolito et al., 2022; Wang et al., 2024a). This approach becomes particularly infeasible given the reality of modern LLMs: many models are closed-source, while some offer only model weights. Therefore, a desired analysis framework should operate across varying levels of model accessibility.

*Challenge 3: Measuring reasoning quality.* Beyond simple performance metrics, we need new ways to evaluate the quality and reliability of model reasoning. This includes developing techniques to understand reasoning paths, creating intermediate representations that capture both token-level and thought-level patterns, and designing metrics that can assess the logical coherence and validity of reasoning steps.

Consequently, we propose that a viable analysis of reasoning behavior should satisfy multiple criteria: it should operate in a post-hoc manner with varying levels of model access, bridge the gap between token-level and thought-level analysis, and provide meaningful metrics for evaluating reasoning quality. Given the absence of tools meeting these requirements, we identify the need for a new analytical framework that can address these challenges while providing useful insights for improving model reasoning capabilities.

### E.2 A COMPARISON BETWEEN LANDSCAPE VISUALIZATION AND TEXTUAL ANALYSIS

Notably, for the language model, one could manually examine the responses to individual questions, as their responses are interpretable by humans. However, this approach has two major limitations:

*Limitation 1: Lack of Scalability.* Analyzing the individual question is time-consuming and labor-intensive. In general, text-based analysis requires human evaluators to carefully read long reasoning chains word by word. For example, if it takes 30 seconds to understand a single problem, reviewing 100 problems would require around 50 minutes of focused human effort. This burden grows quickly, especially as researchers often repeat this process many times while developing models and methods. In practice, researchers need quick, easily interpretable feedback, such as accuracy, when experimenting with changes to models and methods.

*Limitation 2: Lack of Aggregation.* It is difficult to aggregate insights across multiple problems to understand model behavior at the dataset level. Summarizing model behavior across multiple problems presents another challenge. Suppose one researcher has 100 reasoning chains; it is hard for him/her to reliably synthesize the model's overall behavior. Different researchers may arrive at different, subjective summaries, which hinders consistency and interpretability.

By contrast, our visualization method provides a more objective and automatic way to analyze a model, making it much easier for researchers to analyze the model's reasoning behavior. Similar to the t-SNE (Van der Maaten & Hinton, 2008), the visualization enables a more comprehensive analysis of multiple reasoning problems instead of only one problem. The visualization uniquely combines human-readable paths with quantitative, scalable metrics for reasoning process analysis, enabling both model comparisons and mechanistic insights beyond manual text inspection.

Notably, the landscape provides unique insights into LLM reasoning that text analysis alone cannot capture. This power source bridges the gap between localized text understanding and global reasoning behavior. Our analysis in Sec. 3 reveals insights that are not revealed by previous text-based analysis. These insights include structural patterns across many reasoning paths, a strong correlation between early consistency and accuracy, and model-level differences where larger models explore more broadly than smaller ones.

### E.3 THE INTRINSIC RELATIONSHIP BETWEEN VISUALIZATION AND METRICS

In the modeling of this work, we project each thought (state in a trajectory) from text space to numerical space, with the thought's feature vector that each dimension indicates the distance to a particular answer (see Eqn. 1). We compute the feature vectors of all the thoughts from multiple trajectories and then obtain the feature matrix $F$. Then, based on this feature matrix, we compute

(1) the landscape visualization through dimension reduction and (2) the metrics of consistency and uncertainty. From this view, the metrics' information can actually be seen from the landscape. In this work, we mainly focus on the landscapes and also use the metrics plots to help analyze.

In addition, landscape visualizations preserve the information of metrics, including the consistency, uncertainty, convergence, and many other metrics that are not covered in this work. The landscape provides a "global" view of the overall reasoning trajectories, while each metric provides a "local" view of a particular aspect. Note that humans naturally prefer visual matters like figures and videos, e.g., researchers prefer to use t-SNE in understanding the classification models. We recommend using landscape as a visualization tool to help understand the LLM reasoning, while the metric plots can further help inspect some particular aspects.

### E.4    DISCUSSION ON RESULTS AND OBSERVATIONS

In the landscape visualizations, red regions map out the reasoning trajectories that end in incorrect answers, while blue regions map out those that end correctly. The contour lines and the depth of color together convey the density of reasoning states at each step: darker shades mean more trajectories passing through that region. As you observe a landscape evolve from its initial scatter of states toward later clustering, you're seeing whether and how quickly the model's reasoning paths lock onto an answer.

Observation 3.6 arises when we compare only the blue (correct) landscapes of different methods in Fig. 5. Early in the process, all methods scatter widely, exploring many possibilities; over time, though, some methods' contours tighten more rapidly than others. Here, the landscape in Fig. 5(a) converges to its correct region much sooner—and with a denser cluster—than the landscapes in Fig. 5(b) to 5(c), and this faster, tighter convergence corresponds to its higher accuracy. Namely, methods with more scattered landscapes (converge more slowly) present lower accuracy than those that converge faster.

A related pattern appears when we compare models of different sizes in Fig. 2 (Observation 3.1). As we scale from the 1B model to the 70B model, the last 20% of the reasoning steps show increasingly dense blue clusters. Larger models, with greater capacity to store and retrieve information, steer their reasoning more directly and confidently toward the right answer, mirroring their higher accuracy. This further supports the positive correlation between convergence speed (of correct landscapes) and reasoning accuracy, which is revealed in Observation 3.6.

Observation 3.7 emerges from contrasting the red and blue landscapes of the same algorithm in Fig. 5. Here, failure trajectories (red) often settle into a wrong answer by roughly 20-40% of the reasoning process, while success trajectories (blue) only coalesce around the correct answer toward the very end—around 80-100% of the states. This indicates that early reasoning states are exploratory and can drift toward incorrect conclusions, whereas correct solutions only converge late in the trajectories. This convergence-speed disparity between red and blue landscapes also holds across multiple datasets in Fig. 4.

Finally, Fig. 4 shows that each reasoning task leaves a distinct landscape "fingerprint," supporting Observation 3.4. In AQuA, MMLU, and StrategyQA, the landscapes trace wide, structured sweeps of reasoning states—clear evidence of step-by-step deduction and exploration of intermediate hypotheses. By contrast, CommonSenseQA produces a tightly clustered trajectory from the outset, indicating direct retrieval of knowledge rather than an iterative trajectory. This divergence mirrors the tasks themselves: AQuA, MMLU, and StrategyQA require exploratory traversal through multiple reasoning steps, resulting in diverse yet organized state distributions, whereas CommonSenseQA depends on straightforward recall. These task-specific structures demonstrate how our landscape visualizations can uncover both shared patterns and fundamental differences across reasoning challenges.

In addition, each of these qualitative observations is further supported by statistical analyses in Appendix H.1, and we provide full visualizations, including annotated state trajectories (Figs. 32 to 35) and additional model comparisons (Figs. 37 to 38).

### E.5 How to Explain the Increasing Uncertainty and Perplexity?

In short, uncertainty and perplexity in our framework capture two different aspects of the process (belief over answer options vs. predictability of reasoning text), and their mostly increasing trends in Fig. 2 reflect exploratory and increasingly specific reasoning, rather than a monotone loss of commitment. In the following, we clarify this interpretation, emphasize the small late-stage drop in uncertainty, and connect these plots to existing findings in the literature.

**Uncertainty measures the spread of belief over answer options, and its trend reflects exploration plus late-stage commitment.** For each state $s_i$, we form a normalized distance vector $f_i \in \mathbb{R}^k$ over the $k$ answer choices and define $\text{Uncertainty}(s_i) = -\sum_{j=1}^{k} f_i(j) \log f_i(j)$ with $\sum_j f_i(j) = 1$. Low uncertainty means the model strongly prefers one option; high uncertainty means it spreads belief across several. In Fig. 2, the average uncertainty over trajectories tends to increase as reasoning progresses, indicating that intermediate thoughts often open up or rebalance multiple candidates instead of simply sharpening an initial guess. Importantly, in the final stage (roughly the last 20% of thoughts), we observe a slight drop in uncertainty, consistent with the model only firmly committing to its final answer near the end of reasoning.

**Perplexity is defined over the reasoning text itself and naturally increases as thoughts become longer, more specific, and less templated.** For each thought $t_i$, we define $\text{PPL}(t_i) = p_{\text{LLM}}(t_i \mid s_{i-1})^{-1/|t_i|}$, using the model's own conditional probabilities. Early steps often use very generic templates (for example, "Let us think step by step"), which are highly probable and hence low perplexity. Later thoughts contain detailed computations, question-specific entities, and idiosyncratic phrasing, which are rarer under the model and thus yield higher perplexity. This explains why perplexity tends to increase with step index, even though larger models have systematically lower perplexity plots than smaller ones at the same step (as seen in Fig. 2 and Appendix H.5).

**Similar phenomena have been noted in prior work: high-quality or human-like text does not always correspond to the lowest-perplexity generations, and useful content often resides in relatively lower-probability regions of the model's distribution.** For example, Holtzman et al. (2020) observe that pushing perplexity too low leads to degenerate, repetitive text and argue that high-quality text typically has moderate perplexity rather than minimal perplexity. Recent work on chain-of-thought also uses stepwise perplexity to identify "critical" reasoning steps, showing that changes in perplexity along the chain correlate with decision quality (Cui et al., 2025). Our increasing-perplexity plots are consistent with the view that deeper reasoning steps are rarer but important.

**The increasing trends of uncertainty and perplexity are most informative when interpreted together with consistency and accuracy, especially across model scales.** In Fig. 2, larger models exhibit higher consistency (their intermediate states agree with the final answer more often and earlier in the trajectory) while still showing increasing uncertainty and perplexity. This suggests a pattern of "early latent commitment plus exploratory justification": larger models tend to settle on a preferred answer relatively early (reflected in high consistency), then generate more complex, problem-specific reasoning that (i) briefly reconsiders alternatives, raising average uncertainty, and (ii) uses less frequent language, raising perplexity. Smaller models, in contrast, exhibit lower consistency together with similar or higher uncertainty and perplexity, indicating that their exploration is less controlled and more often accompanied by changes in the preferred answer. Our verifier in Sec. 4 leverages exactly these temporal patterns in uncertainty and perplexity (along with consistency) to predict correctness, which further supports that these metrics capture meaningful dynamics rather than noise.

In summary, the non-decreasing uncertainty and increasing perplexity in Fig. 2 do not imply that models simply become less confident as they reason. Rather, they show that models (i) explore and hedge among multiple answer options in the middle of trajectories, then slightly reduce uncertainty when committing at the end, and (ii) move from generic, high-probability templates to rarer, more specialized reasoning text as depth increases, with larger models doing so at overall lower perplexity levels.

### E.6 THE CONVERGENCE SPEED OF CORRECT/INCORRECT TRAJECTORIES AND SMALL/LARGE MODELS

The key point is that they refer to different conditionings: Fig. 1 compares correct versus incorrect trajectories within a fixed model, while Observation 3.1 compares convergence behavior across model sizes, averaged over trajectories. Once this distinction is explicit, the two statements are consistent.

**Within a fixed model, wrong trajectories converge prematurely, and correct trajectories converge later.** Fig. 1 describes a within-model phenomenon: for a given model and dataset, trajectories that eventually answer incorrectly tend to "lock in" to a wrong answer region earlier, while correct trajectories stay exploratory for longer and only settle near the correct region toward the end. We formalize this using a convergence statistic based on high-dimensional state features. For a trajectory with states $f_1, \ldots, f_n$, we measure the distance $d_i = \|f_i - f_n\|_2$ between each state and its own final state, fit a log-linear model $\log d_i \approx \alpha + \beta i$, and use $e^\beta$ as a convergence coefficient, where smaller $e^\beta$ means faster convergence. Tab. 1(a) shows that, under the same model and method, incorrect paths have significantly smaller $e^\beta$ than correct paths (p value = 0.008), which matches the intuition in Fig. 1 that wrong paths converge "too fast" to wrong answers, while correct paths take more steps before stabilizing.

**Across model sizes, larger models converge more efficiently to correct regions on average.** Observation 3.1 studies a different axis, comparing convergence across model scales on AQuA. Here we keep the dataset, prompt, and reasoning method fixed and vary only the model size (for example, Llama 3.2 1B, 3B, 8B, and Llama 3.1 70B). We quantify how directly trajectories move from initial to final states using the speed metric $\text{speed} = \dfrac{\|\bar{s}_n - \bar{s}_0\|_2}{\sum_{j=1}^n \|\bar{s}_j - \bar{s}_{j-1}\|_2} \in [0, 1]$, where $\bar{s}_i$ is the 2D position of state $i$. Higher speed means less wandering and a more direct path. Appendix H.1 shows that this speed correlates very strongly with accuracy (p-value about $9.4 \times 10^{-11}$), and that larger, more accurate models have higher speeds than smaller ones. Intuitively, as the model scales up, a larger fraction of its trajectories quickly head toward the correct region and follow relatively straight paths, so the landscape "converges faster" in the sense of Observation 3.1.

**The two statements are compatible once we distinguish between conditional and marginal views.** Putting these together, we have two different but compatible facts. Within each fixed model, if we condition on outcome, incorrect trajectories tend to converge earlier than correct ones, that is, $\mathbb{E}[e^\beta_{\text{wrong}}] < \mathbb{E}[e^\beta_{\text{correct}}]$, which is Fig. 1 and Observation 3.7. Across models of different sizes, when we look at all trajectories or at correct trajectories in aggregate, larger models have higher speed and smaller convergence coefficients; that is, they reach their final regions, especially the correct region, more efficiently than smaller models, which is Observation 3.6. In practice, a 70B model still has some early wrong convergences, but these are fewer, and its successful trajectories travel more directly to the correct cluster and dominate the overall landscape. An analogy is that a stronger solver both has more correct solutions and tends to reach them more quickly on average, while for any fixed solver, the quickest answers are often overconfident mistakes.

In summary, Fig. 1 describes a within-model effect (wrong paths in that model converge earlier than right paths), while Observation 3.1 describes an across-model effect (larger models' trajectories, especially correct ones, converge faster and more directly than those of smaller models).

### E.7 THE CLAIMS OF POSITIONS OF LOT

**All core claims are defined and validated in the original answer distance space and visualized in 2D space.** For each intermediate state $s_i = [x, t_1, \ldots, t_i]$, we construct a feature vector $f_i \in \mathbb{R}^k$ with components $f_i(j) = d(s_i, c_j)$, where $d(s_i, c_j)$ is the length-normalized perplexity-based distance to the candidate answer $c_j$, followed by $\ell_1$ normalization so that $f_i$ lies in a probability simplex-like space over the $k$ options. This vector is exactly the model's internal assessment of how the current partial reasoning aligns with each answer. Our metrics are defined directly on $f_i$ and on the thoughts: consistency checks whether $\arg\min_j f_i(j)$ matches $\arg\min_j f_n(j)$, uncertainty is the entropy of the normalized $f_i$, and convergence is captured by the margin and stability of the best answer over steps. Using these quantities, we show that incorrect trajectories tend to converge earlier and more sharply to a wrong answer (Observation 3.7), while correct trajectories exhibit higher intermediate consistency and lower uncertainty (Observation 3.8), as summarized in the consistency and uncertainty plot in

Figs. 2, 4, 5 and statistically verified in Appendix H.1 to H.3. None of these results relies on any geometric property of the 2D representations.

**Our consistency metric tracks the stability of answer preference and is largely insensitive to global distribution sharpness.** Consistency depends only on $\arg\min f_i$ and $\arg\min f_n$, that is, on the ranking of distances across choices. Any monotone transform that uniformly sharpens probabilities, for example, $p_j \mapsto p_j^\alpha$ with $\alpha > 1$ or temperature rescaling $p'(c_j \mid s_i) \propto p(c_j \mid s_i)^{1/T}$ with $T < 1$, preserves this ranking and leaves $\text{Consistency}(s_i)$ unchanged. A generic increase in sharpness would therefore not by itself increase consistency, nor would it create the systematic gaps we observe between correct and incorrect trajectories or between small and large models. For consistency to increase, intermediate states must more often agree in their top choice with the final answer, which is exactly what we interpret as more stable belief dynamics.

**The visible clusters and paths align with these high-dimensional metrics and labels, which indicates that they reflect real structure.** Distances to the correct answer and to distractors are explicit coordinates of $f_i$, and we also embed answer anchors as landmarks (Eqn. 3). Regions that appear in 2D as clusters around a particular anchor correspond to states where $f_i$ places most mass on that option, and consistency with the final prediction is high. Similarly, trajectories that visually move from diffuse areas to a compact region near the correct anchor correspond to sequences where distances to the correct answer decrease and the margin over other options increases, precisely what our convergence coefficients and consistency plots quantify. Observation 3.6 and Tab. 1(a) show that incorrect trajectories converge earlier and more strongly toward their final answer region than correct trajectories, and Observation 3.7 and Tab 1(b) show that paths with higher speed tend to be more accurate. This alignment between geometric patterns and scalar metrics suggests that the 2D clusters and paths are manifestations of structure already present in the original feature space.

**LoT is explicitly framed as a behavioral diagnostic of belief dynamics rather than a direct probe of latent cognitive mechanisms.** The log probabilities reflect surface likelihoods rather than an explicit symbolic proof tree. But for an autoregressive language model, the conditional likelihood distribution is what governs behavior, so any latent reasoning process must ultimately manifest as changes in these scores. By tracking the sequence $\{f_i\}_{i=1}^T$, we analyze how the model's own scoring function reallocates probability mass across the $k$ candidate answers as thoughts are generated. Phenomena such as incorrect trajectories converging earlier to wrong options than correct trajectories converging to the right one, and the existence of mid-trajectory states with low consistency and high entropy, are expressed in terms of margins and entropy in this original feature space and are not trivial consequences of simply preferring high-probability answers. LoT therefore makes a deliberate choice to study belief evolution over answer options as an operational view of reasoning behavior, without claiming to reconstruct a hidden conceptual reasoning structure.

**The t-SNE is used only to visualize an already low-dimensional, semantically anchored space, and all qualitative patterns in the landscapes are corroborated by projection-independent analyses and alternative projectors.** Each state is represented by the $k$-dimensional vector $f_i = (d(s_i, c_1), \ldots, d(s_i, c_k))$ after length and $\ell_1$ normalization, and the answer options themselves are embedded as anchors in the same space. Two states are close if they induce similar distributions over answers, so this space already has a clear probabilistic interpretation before any projection. We then apply t-SNE to map this interpretable space to 2D primarily to illustrate how trajectories move from regions where multiple answer clusters are mixed to regions dominated by a single cluster and how correct and incorrect trajectories end near different anchors, as shown in Figs. 2, 4, and 5. These are coarse neighborhood and cluster properties rather than fine-grained claims about global distances or angles. Moreover, Appendix H.8 shows that replacing t-SNE with UMAP or PaCMAP yields qualitatively similar patterns of early diffuse exploration versus later convergence and similar separation of fast-converging wrong trajectories from slower-converging correct ones. The corresponding one-dimensional plots in the original space (distance to correct answer, entropy, consistency over steps) show the same trends. This indicates that the observed paths are rooted in the underlying answer distance features rather than artifacts of a particular projection algorithm or random seed.

**The analyses of model scale explicitly separate global distribution sharpness from genuine differences in trajectory behavior by using normalized features, within-model comparisons, and additional controls.** Larger models indeed tend to produce sharper output distributions, and we observe lower uncertainty and perplexity in Fig. 2. To mitigate this effect, we always apply $\ell_1$

normalization to each $f_i$, that is, $f_i \leftarrow f_i/|f_i|_1$, so that we focus on the relative geometry among options rather than absolute log probability scales. More importantly, many key comparisons are within a fixed model, such as correct versus incorrect trajectories or different reasoning methods and datasets evaluated with the same model, where global calibration and sharpness are essentially held fixed. In these settings, we still find that failure trajectories typically converge earlier and remain highly consistent around a wrong option, while success trajectories converge later but more reliably to the correct one, which cannot be attributed to differences in global sharpness. As shown in Tab. 5, while there is a slight variation in perplexity across model scales, the values all fall within a comparably narrow range (from 1.42 to 1.96), which demonstrates that for decoding the same CoTs, different models in the Llama-3 family produce similar and comparable perplexity scores. This supports the validity of comparing perplexity across models in our study.

**The predictive power of the lightweight verifier trained on LoT features provides independent evidence that these representations capture meaningful structure in reasoning behavior rather than only visualization artifacts.** The lightweight verifier is a simple random forest that operates solely on the likelihood-based state features and consistency statistics derived from $f_i$, without access to raw text or hidden activations. Despite this simplicity, it significantly improves accuracy over unweighted self-consistency and yields much stronger test time scaling as the number of trajectories increases, as shown in Figs. 6 and 7. If the features only reflected superficial sharpness or spurious projection effects, it would be difficult for such a simple model to reliably distinguish correct from incorrect trajectories across datasets, methods, and models. The verifier's effectiveness supports the view that LoT's representation retains stable and informative signals about reasoning quality.

In summary, LoT should be interpreted as a method for visualizing and quantifying belief dynamics over candidate answers rather than for reconstructing a latent "conceptual reasoning structure." It builds a well-defined model of the internal representation of each thought in the answer distance space $f_i$, defines trajectory-based metrics and statistical tests that do not depend on t-SNE, and uses 2D landscapes that are robust to the choice of projector and consistent with these metrics.

### E.8   How is Cross-question Comparability Achieved?

LoT does not assume a single global semantic manifold over all texts. Instead, each state is embedded into *a shared belief space over answer options*, so cross-question comparability comes from a common probability simplex structure and per-question normalization, and all quantitative results are computed in this space before any 2D projection.

**(1) LoT works in a decision space over answer options, not a universal text embedding space.** For a multiple-choice question with candidates $C = \{c_1, \ldots, c_k\}$ and an intermediate state $s_i = [x, t_1, \ldots, t_i]$, we construct a feature vector $f_i \in \mathbb{R}^k$ with components $f_i(j) = d(s_i, c_j)$, where $d(s_i, c_j)$ is the length-normalized perplexity-based distance (Eq. (2)), followed by $\ell_1$ normalization so that $\sum_j f_i(j) = 1$ (Section 2.2). Thus $f_i$ is a point in a probability simplex over answer indices $\{1, \ldots, k\}$, encoding how the model distributes belief over the options at state $s_i$, rather than an embedding of raw text into a semantic space.

**(2) Cross-question comparability is defined in terms of belief geometry over answer indices, not direct semantic similarity of answer texts.** For a fixed dataset like AQuA, all questions share the same number of options $k$, so every state feature $f_i$ lies in the same $k$-dimensional simplex. Two states from different questions are close if the model exhibits similar belief patterns, such as strong confidence in one option, confusion between two options, or near-uniform uncertainty. Our core metrics use exactly this structure. Distances to the correct answer are computed by selecting the coordinate corresponding to the ground-truth index. These quantities are therefore comparable across questions because they measure how belief mass concentrates and stabilizes over indices, not over specific strings.

**(3) Each landscape is learned from a single global embedding over the shared simplex, not from stitched per-question subspaces.** For a given dataset and model, we obtain state features $\{f_i\}$ from all questions and trajectories. We normalize feature vectors by reordering choices so the correct answer appears in the first dimension across all questions. These state features, together with the choice anchors (Eqn. 2), are pooled into one matrix in $\mathbb{R}^k$ and apply a single dimensionality reduction $g : \mathbb{R}^k \to \mathbb{R}^2$ (by default, t-SNE). Figs. 2, 4, and 5 are produced by this single mapping per dataset, so all points in a figure lie in the same underlying belief space. For datasets with different $k$ (for

example, AQuA versus StrategyQA), we keep landscapes separate and compare them via metrics such as consistency, uncertainty, and histogram intersection scores in Tab. 1(c), rather than forcing them into a shared manifold.

**(4) Our quantitative conclusions do not rely on interpreting arbitrary distances between states from different questions as semantic similarity.** All core metrics and statistical tests are defined within each question's simplex and then aggregated. For example, the convergence coefficient in Appendix H.1 is obtained by fitting $\log d_i^{(q)} \approx \alpha_q + \beta_q i$ for each question $q$, where $d_i^{(q)}$ is the distance from $s_i$ to the correct answer for that question, and then analyzing the distribution of $\exp(\beta_q)$ across questions (Tab. 1(a)). Similarly, the consistency and uncertainty plot in Figs. 2, 4, and 5 are computed by first evaluating these metrics per state and per question, then averaging over questions. None of these analyses requires treating the Euclidean distance between a state from question $q$ and a state from question $q'$ as semantically meaningful; they depend only on how beliefs evolve within each local answer simplex.

**(5) Geometrically, the global landscape can be viewed as many aligned local simplexes embedded in a common feature space, and we interpret it in that way.** Each question induces a $k$-vertex simplex of belief states over its own options. Because we use the same coordinates (option indices and normalized distances) for every question in a dataset, these simplexes live in the same ambient space $\mathbb{R}^k$. When we apply t-SNE to the pooled set of $\{f_i\}$, we obtain a 2D projection in which clusters correspond to structurally similar belief states, such as "high confidence in the chosen option" or "persistent confusion between two options". We do not claim that this 2D embedding is a globally faithful semantic reasoning space; it is an intuitive visualization of how belief states populate the probability simplex, while our formal claims are grounded in the per-question metrics described above.

In summary, LoT embeds thoughts into a dataset-specific belief space over answer indices that is shared across questions, and our conclusions are derived from within-question geometry and aggregated statistics rather than from assuming a universal semantic manifold over text.

### E.9   POTENTIAL EXTENSION TO PRUNING UNPROMISING TRAJECTORIES

We showcase that our tool can be utilized to identify potentially incorrect reasoning trajectories at test time. In Section 3.3, we build up a lightweight verifier, which is based on the thoughts' feature vectors and the consistency metric from the landscape of thoughts. This verifier indeed aims to predict the correctness of a reasoning trajectory, in order to boost the reasoning accuracy at test time. It is proven to be beneficial to the voting of multiple reasoning trajectories, as shown in Sec. 4.2.

Further, this verifier (together with the visualization tool) can be adopted to prune unpromising reasoning trajectories in tree-based searching. For instance, in methods like tree-of-thoughts and MCTS, a model explores multiple reasoning trajectories and usually uses the same model to identify the promising paths to search for the ultimate solution. Here, by leveraging features from the landscape of thoughts and the consistency metric, our verifier can identify flawed trajectories early during reasoning, acting as an efficient pruning mechanism to boost the search efficiency and reasoning performance.

Therefore, our tool can be integrated into the reasoning methods to monitor particular reasoning patterns (e.g., the correctness) and help understand as well as boost reasoning. There are multiple directions that deserve future exploration, including the one to identify and prune the potentially incorrect reasoning trajectories (Han et al., 2025). Such capability is particularly relevant in safety-critical applications, where challenges like jailbreak defense (Li et al., 2023b), safety alignment (Cao et al., 2025), and knowledge retention (Wang et al., 2025a) demand robust monitoring of LLM reasoning behavior.

### E.10   POTENTIAL EXTENSION TO IDENTIFY POST-HOC TRAJECTORIES

In the following, we discuss the feasibility of detecting post-hoc trajectory using our framework, particularly in defining the post-hoc trajectory. A post-hoc trajectory refers to the trajectory that the model exhibits high confidence in a single answer in the early states and maintains high consistency across states in the trajectory. Specifically,

- the "early state" correspond to the "very early tokens of the response";
- the "high confidence in a single answer" corresponds to the "model has chosen its answer";
- the "high consistency across states in the trajectory" corresponds to the "trajectory is produced as a consequence of that decision".

Namely, the post-hoc trajectory can be potentially identified by inspecting the confidence and consistency of particular positions of states in our framework. Then, we elaborate on the more detailed definitions for the three components above.

- For defining the "early states", it should have an absolute threshold of states index, e.g., early 10 states, or a relative threshold, e.g., early 10% of states. This threshold should be chosen deliberately, and the states with an index smaller than this threshold are categorized as "early states".
- Similarly, a clear threshold is necessary for defining the "high confidence" or "high consistency", e.g., over 80% confidence and 60% consistency. With the metrics defined in Section 2.3, here, we should examine (1) the confidence of the early states in the trajectory and (2) the consistency across all states of the trajectory. Here, only the trajectory that exceeds the confidence threshold as well as the consistency threshold can be classified as a post-hoc trajectory.

In conclusion, our framework shows promise for identifying post-hoc trajectories. Meanwhile, we should note that it still needs (1) to choose particular thresholds for the precise definition of post-hoc trajectory and (2) to collect a set of reliable data to verify the effectiveness in identifying post-hoc trajectory. These are quite challenging to conduct. Although it goes beyond the scope of work, we believe investigating post-hoc trajectory in reasoning is valuable and merits exploration in future work.

### E.11 LIMITATIONS AND FUTURE DIRECTIONS

**Scope.** While the *Landscape of Thoughts* offers a practical lens on model reasoning, its current instantiation is limited to multiple-choice settings. Extending LoT to open-ended reasoning—including mathematical problem solving, code generation, and planning—requires handling less structured and more entangled reasoning paths, especially as LLM training infrastructure continues to evolve (Tang et al., 2025; 2026). Two complementary threads of future work are: (i) improving accessibility by producing intuitive visual and textual explanations that help non-experts inspect and trust model behavior, and (ii) developing automated, scalable detectors of reasoning failures to improve reliability across applications.

**Key challenge: synthesizing options.** The central obstacle is the quality of the synthesized answer options. Human-authored distractors are carefully calibrated to be plausible, exposing distinctions between (1) correct reasoning and (2) reasonable-but-wrong reasoning (e.g., overlooking information or making arithmetic slips). In contrast, LLM-generated distractors can be implausible and thus trivially eliminated when juxtaposed with the correct option, yielding visualizations that over-emphasize the correct trace and limit diagnostic value. Moreover, LLMs may reuse similar reasoning patterns, producing near-duplicate error modes across incorrect options and reducing the comprehensiveness of the analysis.

**Mitigations.** To address these issues, we can elicit higher-quality distractors with state-of-the-art LLMs (e.g., OpenAI o3, Gemini 2.5 Pro) and tune sampling hyperparameters (temperature, top-$p$) to promote diversity and explore alternative solution trajectories.

**Binary reformulation.** A practical alternative is to recast multiple-choice prompts as binary (yes/no) queries. For example, the question "What is the capital of France?" can be reformulated as "Is Paris the capital of France?" with options *Yes* or *No*. Under this framing, both options remain prima facie plausible: the incorrect choice admits coherent yet flawed rationales, and the variety of "No" trajectories preserves diversity without resorting to obviously implausible distractors.

**Beyond multiple choice.** Although open-ended tasks are beyond the present scope, LoT is, in principle, extendable. The key requirement is to construct a candidate set of answers by querying the model (a non-trivial step that is given for free in multiple-choice tasks). Treat the ground-truth answer as one option and generate additional plausible alternatives using LLMs; LoT can then analyze the induced reasoning behaviors in these open-ended scenarios.

**Case: code generation.** Code generation introduces additional challenges: there is typically no single ground-truth program, and evaluation proceeds via test suites. Candidate programs are diverse and do not naturally discretize into options. We propose the following procedure: (i) sample multiple candidate solutions from the model under evaluation; (ii) score each by the number of tests passed; (iii) apply a threshold to separate more-correct from less-correct solutions; (iv) embed and cluster solutions within each partition; and (v) use cluster centroids as anchors for "correct" and "incorrect" choices. Cluster quality can be assessed with the Silhouette Score and the Davies-Bouldin Index. These anchors enable a LoT-style visualization over the solution space and provide insight into reasoning behaviors.

In summary, our visualization framework is adaptable beyond multiple-choice scenarios. To our knowledge, LoT is the first landscape visualization tool aimed at analyzing LLM reasoning; it is imperfect and remains open to improvement and extension. We believe it constitutes a small but meaningful step toward understanding and improving the reasoning processes of LLMs.

### E.12  A COMPARISON BETWEEN LIGHTWEIGHT VERIFIER AND REWARD-GUIDED ALGORITHMS

It is worth noting that our goal is not to build a sophisticated verifier, but rather to demonstrate how the feature vectors from the landscape visualization can be effectively used.

In general, reward-guided algorithms are more computationally efficient than the path landscape. Specifically, for a reasoning path with $n$ thoughts and $c$ answer choices, constructing the landscape requires $n \times c$ forward passes through the reasoning model. In contrast, a reward-guided approach typically makes a single call to a reward model that evaluates the entire reasoning chain at once.

Meanwhile, it's important to consider the overhead involved in training the reward models in reward-guided algorithms. Notably, for Process-Reward Models (PRMs) (Luo et al., 2024; Xu et al., 2025), collecting high-quality training data often requires detailed, fine-grained annotations of reasoning steps, which can be costly and time-consuming. Moreover, training a reward model (often itself an LLM) incurs significant computational expense. In contrast, our lightweight verifier is much more efficient to train, as it requires no human annotations and uses easily obtainable data.

## F  RELATED WORK

**Reasoning with large language models.** Chain-of-Thought (CoT) prompting (Wei et al., 2022; Kojima et al., 2022) has empowered LLMs to tackle multi-step reasoning problems by generating intermediate steps before producing a final answer. Building upon CoT, numerous methods have been proposed to address various challenges, including compositional generalization (Zhou et al., 2023; Khot et al., 2023), planning (Yao et al., 2023a; Hao et al., 2023), knowledge reasoning (Liang et al., 2024a; 2025; 2024b), graph analytics (Shang & Huang, 2025; Shang et al., 2025), rule learning (Zhu et al., 2023), and active reasoning (Zhou et al., 2025) within the CoT reasoning. Beyond solving reasoning tasks, CoT has also emerged as a foundational framework for other techniques, such as fine-tuning LLMs (Zelikman et al., 2022; Wang et al., 2025b; Zhang et al., 2025a), enabling LLM-based agents (Yao et al., 2023b), and facilitating test-time scaling (Snell et al., 2024; Zhang et al., 2025b). Nevertheless, most of these approaches are developed in a trial-and-error manner, largely due to the absence of proper tools for analyzing the CoT.

**Understanding chain-of-thought reasoning.** There are a few studies that explore what makes CoT prompting effective by perturbing its exemplars. To be specific, Madaan & Yazdanbakhsh (2022) found that the text and patterns of exemplars help CoT generate sentences resembling correct answers. Besides, Wang et al. (2023a) highlighted the importance of maintaining the correct order of reasoning steps, while Ye et al. (2022) demonstrated that using complementary exemplars can enhance reasoning performance. Furthermore, CoT can benefit from longer reasoning chains, even without new information to the prompt (Jin et al., 2024). Another line of research investigates CoT's general behavior (Tang et al., 2023; Saparov & He, 2023; Saparov et al., 2023; Shi et al., 2023). For example, CoT heavily depends on the semantic structure of the problem to perform reasoning (Tang et al., 2023), struggles with planning and unification in deductive reasoning (Saparov & He, 2023), has difficulty generalizing to longer reasoning paths (Saparov et al., 2023), and can be easily misled by irrelevant information in the context (Shi et al., 2023; Zhou et al., 2024). However, these observations

are derived from specific reasoning tasks and prompt settings, limiting their applicability to other scenarios. In contrast, we introduce a general-purpose tool that allows users to analyze reasoning in their contexts.

**Tools for analyzing chain-of-thought.** To the best of our knowledge, the only existing tool for analyzing CoT is gradient-based feature attribution (Wu et al., 2023), which computes a saliency score for each input token based on the model's output. However, these token-level saliency scores do not directly capture the thought-level, multi-step reasoning process of LLMs. Consequently, the main finding in (Wu et al., 2023) is that CoT stabilizes saliency scores on semantically relevant tokens compared to direct prompting. Metrics designed to quantify CoT performance (Chen et al., 2024; Ton et al., 2024) can also be used to analyze the reasoning behaviors of LLMs. For instance, Ton et al. (2024) employs information gain to identify failure modes in reasoning paths, aligning with Observation 3.7 in this paper. However, our 2-D visualization offers significantly deeper insights than a single information gain metric. Additionally, the verifier derived from our tool is conceptually related to outcome-supervised reward models (Cobbe et al., 2021).

**Measuring uncertainty and consistency in LLM reasoning.** Several works in this research line compute metrics (such as confidence and perplexity) by leveraging the features from LLMs to measure and detect hallucination in reasoning (Li et al., 2023a; Chuang et al., 2024; Yang et al., 2025). Specifically, low confidence and high perplexity often indicate unreliable reasoning, enabling the development of lightweight detectors to guide reasoning and mitigate hallucinations. However, these metrics have limitations (Xiong et al., 2024; Zhang et al., 2023): they can exhibit over-confidence or low perplexity in incorrect responses, their reliability relies heavily on the models' capability, and they cannot provide more comprehensive insights into the multiple reasoning trajectories. By contrast, our landscape of thoughts offers a holistic approach, integrating several existing metrics. This framework enables global qualitative analysis, including measures of perplexity, consistency, and uncertainty. In addition, the landscape of thoughts enables the development of advanced tools to enhance reasoning by using the features and metrics, as mentioned in Sec. 3.3.

## G  EXPERIMENT SETTINGS

### G.1  SETUP

Visualizing the landscape of thoughts fundamentally relies on the decoding probability of LLMs. To this end, we adopted four open-source models with varying parameter sizes, namely `Llama-3.2-1B`, `Llama-3.2-3B`, `Llama-3.1-8B`, and `Llama-3.1-70B`. We repeatedly sample 10 times from the target LLM using the same reasoning strategy as self-consistency (Wang et al., 2023b).

For visualization purposes, we randomly sample 50 questions from the testing split of each dataset and generate reasoning paths with the setup described above. For simplicity, we compute distances only between each state and all candidate answers. To visualize multiple problems in a shared space, we always place the distance to the correct answer as the first element of each feature vector. This alignment allows joint analysis across problems, as introduced in the paragraph below Equation 4. We then aggregate feature vectors from all problems into a feature matrix (Equation 2), which is passed to t-SNE to compute the pairwise distance between any two states and then outputs the 2D coordinate of each state.

For training the lightweight verifier, we randomly sample 20 questions from the training split of each dataset to obtain the feature matrix $S$. We extract these features using three model scales: `Llama-3.2-3B`, `Llama-3.1-8B`, and `Llama-3.1-70B`. Despite the relatively small training set, it proves sufficient for our lightweight verifier, which we subsequently evaluate on the data for visualization in Sec. 3.

### G.2  DATASETS

**AQuA** (Ling et al., 2017). This dataset develops to challenge language models' quantitative reasoning capabilities. The AQuA presents complex algebraic word problems in a multiple-choice format, where only one is correct. Each problem requires numerical computation, deep linguistic understanding,

and logical inference. It provides a nuanced assessment of a model's ability to translate textual information into algebraic reasoning.

**MMLU** (Hendrycks et al., 2021). Spanning 57 distinct academic and professional domains, MMLU provides a rigorous test of language models' capabilities across humanities, social sciences, hard sciences, and technical disciplines.

**StrategyQA** (Geva et al., 2021). This dataset is designed to evaluate implicit reasoning and multi-hop question answering. The dataset is characterized by yes/no questions that demand implicit reasoning strategies. Unlike straightforward factual queries, these questions require models to construct elaborate reasoning paths, showing hidden logical connections.

**CommonsenseQA** (Talmor et al., 2019). This dataset assesses commonsense reasoning through multi-choice questions derived from the ConceptNet knowledge graph (Speer et al., 2017). The dataset aims to test a model's understanding of commonsense concepts and ability to make logical inferences. However, the questions often require the model to incorporate external knowledge to select the correct answer from plausible distractors.

Note that AQuA, MMLU, and StrategyQA all demand exploratory traversal of intermediate reasoning states, resulting in diverse but structured landscapes. CommonsenseQA, conversely, represents a distinct domain where answers depend on static knowledge rather than emergent reasoning pathways.

### G.3 DECODING ALGORITHMS

**Chain of Thought (CoT)** (Wei et al., 2022). CoT elicits the LLM's reasoning capabilities by incorporating few-shot examples that demonstrate explicit reasoning steps. It provides the model with exemplar reasoning traces to guide its problem-solving process.

**Zero-shot CoT** (Kojima et al., 2022). The core idea of this prompt strategy lies in adding simple instructions, e.g., "Let's think step by step." to the prompt, enabling models to generate reasoning traces without assigned task-specific examples.

**Least-to-Most (LtM)** (Zhou et al., 2023). LtM is an innovative reasoning approach that systematically breaks down complex problems into progressively simpler subproblems. This approach mirrors human cognitive problem-solving strategies, where individuals naturally break down complex tasks into smaller, more comprehensible parts.

**Tree-of-Thought (ToT)** (Yao et al., 2023a). ToT expanded this concept by creating a more sophisticated, multi-branching reasoning framework. While CoT follows a linear path of reasoning, ToT introduces a more dynamic exploration, allowing models to generate multiple reasoning paths simultaneously, evaluate them, and strategically prune less promising trajectories.

**Monte Carlo tree search (MCTS)** (Zhang et al., 2024). MCTS is a powerful computational algorithm originally developed for game-playing strategies, particularly in complex decision-making environments like chess and Go. The method uses probabilistic sampling and tree exploration to systematically navigate potential solution spaces, balancing exploring new possibilities with exploiting promising paths. We adopt the task-agnostic node expansion and evaluation prompt from ReST-MCTS (Zhang et al., 2024) to conduct our experiment across different tasks.

## H SUPPLEMENTARY RESULTS AND ANALYSIS

### H.1 STATISTICAL VERIFICATION OF THE OBSERVATIONS

In this part, we conduct extra experiments and statistically verify Observations 3.1, 3.4, 3.6, and 3.7, while the other Observations 3.2, 3.5, and 3.8 have been quantitatively verified by the metrics in Sec. 2.3.

To verify Observations 3.6, we calculate the convergence coefficient ($e^\beta$) by fitting a log-linear regression model to the sequence of distances $d_i$ between each state and the final answer as $\log(d_i) \approx \alpha + \beta i$, where $\alpha$ is the intercept term; $\beta$ is the slope coefficient that quantifies convergence behavior; $i$ represents the position index in the reasoning chain. Lower values of $e^\beta$ indicate faster convergence. For Observations 3.1 and 3.7, we measure the speed of a reasoning path moving from start to end as

Table 1: Statistical verification of the observations in Sec. 3.

(a) Verifying Observation 3.6

|  | Correct | Incorrect |
|---|---|---|
| CoT | 1.026 | 0.975 |
| L2M | 1.026 | 0.989 |
| ToT | 1.004 | 0.987 |
| MCTS | 1.002 | 0.985 |

(b) Verifying Observation 3.7 and 3.1

|  | Speed | Accuracy |
|---|---|---|
| CoT | 0.322 | 84.4% |
| L2M | 0.224 | 82.2% |
| ToT | 0.205 | 81.6% |
| MCTS | 0.198 | 75.8% |

(c) Verifying Observation 3.4

|  | AQuA | MMLU | StrategyQA | Common SenseQA |
|---|---|---|---|---|
| AQuA | 1.0 | 0.914 | 0.895 | 0.859 |
| MMLU | 0.914 | 1.0 | 0.870 | 0.843 |
| StrategyQA | 0.895 | 0.870 | 1.0 | 0.889 |
| Common SenseQA | 0.859 | 0.843 | 0.889 | 1.0 |

speed $= \frac{\|\bar{s}_n - \bar{s}0\|}{\sum_{j=1^n} \|\bar{s}j - \bar{s}j-1\|} \in [0, 1]$, where $\bar{s}_i$ represents the 2D coordinate of the state $i$. Whereas Observation 3.4, we compute pairwise histogram intersection scores of the density distributions. Lower scores indicate greater dissimilarity between landscapes.

Notably, for Tab. 1(a), we found that correct paths consistently show slight divergence, while incorrect paths show more convergence (p-value = 0.008), thus verifying Obs. 3.6. As shown in Tab. 1(b), speed and accuracy correlate strongly (p-value = 9.421e-11), thus verifying Observation 3.7. This is also applicable for verifying Observation 3.1. Tab. 1(c) shows that lower scores indicate greater dissimilarity between landscapes, which verifies Observation 3.4, i.e., AQuA, MMLU, and StrategyQA are more similar, while CommonSenseQA exhibits distinct patterns.

## H.2 ANALYSIS OF REASONING TRAJECTORY CONVERGENCE

We aim to investigate Observation 3.7 quantitatively to show its consistency with the statistical result. Specifically, we analyzed all questions from the AQuA dataset using the Llama-3.1-8B-Instruct model with the CoT method. Among the 500 reasoning trajectories (50 questions, with 10 trajectories per problem), we observed that cases where a reasoning chain initially approached the correct answer's path but later diverged to an incorrect conclusion were quite rare: only 4 questions (8% of all questions) exhibited this phenomenon, accounting for just 9 reasoning trajectories (1.8% of all trajectories). This indicates that such failure cases are infrequent within the overall set of generated reasoning trajectories.

Thoughts that diverge from the correct answer exhibit remarkable proximity at certain states. We quantify the proximity by measuring the distance between states: a smaller distance indicates that the state is closer to the correct answer. Notably, in the following example, the chain's reasoning reached a minimum distance of just 0.077 from the correct path before drifting to a final distance of 0.182. This reveals that even incorrect answers may closely track the correct reasoning at key moments.

We provide a concrete example of such a failure in the following reasoning chain for a question below, where the correct answer is B) 28%.

> **An example of reasoning that is close to the correct answer at intermediate thoughts but finally converges to the incorrect answer**
>
> > Q: The original retail price of an appliance was 60 percent more than its wholesale cost. If the appliance was actually sold for 20 percent less than the original retail price, then it was sold for what percent more than its wholesale cost?
> > Options: A) 20% B) 28% C) 36% D) 40% E) 42%
> > The model's output is:
> > 1. "To find the percent more than the wholesale cost for which the appliance was sold, we need to break down the calculations step by step."
> > ...
> > 11. "So, (0.28x / x) * 100 = 28%." $\rightarrow$ The state that is closest to the correct answer, with a distance of 0.077.
> > 12. "Therefore, the appliance was sold for 28% more than its wholesale cost."
> > 13. "The answer is C." $\rightarrow$ Eventually, this state reaches the incorrect answer, with distance as 0.182

## H.3 FURTHER INVESTIGATION ON THE CONSISTENCY METRIC

In the Tab. 2, we analyze the model responses for drawing Fig. 5 and report (1) the average number of thoughts, (2) the average number of tokens in a thought, and (3) the average consistency of different thoughts.

Table 2: The relation of consistency with the number of thoughts and tokens

| Model | Avg. Thoughts | Avg. Tokens | Avg. Consistency |
|---|---|---|---|
| Llama-3.2-1B | 8.07 | 346.81 | 0.51 |
| Llama-3.2-3B | 11.73 | 439.37 | 0.40 |
| Llama-3.1-8B | 21.38 | 715.56 | 0.48 |
| Llama-3.1-70B | 13.55 | 442.72 | 0.51 |

As can be seen, the 8B/70B models produce more thoughts than the 1B/3B models; meanwhile, their intermediate states of correct chains in blue are more consistent than those of the 1B/3B model. The Pearson correlation coefficient between CoT length (thoughts) and consistency is only -0.0185, indicating a very weak negative correlation that is not approaching either +1 or -1. **Hence, higher consistency doesn't correlate with shorter chains. Fewer CoT steps do not necessarily indicate higher consistency.**

As we introduced in Sec. 2.3, the consistency metric is used to understand whether the LLM knows the answer before generating all thoughts. Here, the observation "larger models have higher consistency" actually indicates that a larger model has a higher probability of knowing its final answer in its middle steps of reasoning. We believe that this observation is new and insightful to the community.

In addition, we investigate whether the consistency is meaningful for the reasoning outcome or if it consistently decreases as the thoughts increases. We ask the Llama 3.1 8B Instruct model to generate some random thoughts, using a temperature of 0.7 to encourage more varied responses. For each of the 10 questions we select from AQuA, we then randomly combine different numbers of these thoughts to create 50 chains for each question, with the number of thoughts ranging from 2, 4, 8, 16, or 32. After generating these chains, we calculate the distance matrix and report the consistency, as shown in Tab. 3. **Notably, as the length of the chain of random thoughts increases, the consistency consistently decreases, regardless of the correctness, which justifies that consistency will not increase as $n$ increases.**

Besides, we conduct extra experiments on a harder task across model scales and show that larger models achieve higher consistency than smaller models on both easy and hard tasks. Specifically, we apply the MMLU-Pro (Wang et al., 2024b) as a harder benchmark. MMLU-Pro is a more challenging version of MMLU (adopted in this work), extending the MMLU dataset by integrating more reasoning-focused questions. We sample problems from the MMLU-Pro Math subset and evaluate models of different scales, following the consistency calculation described in equation 5. The experiment results are shown as follows:

**The above results show that larger models have substantially higher consistency on both the easy task (MMLU) and the hard task (MMLU-Pro) than smaller models.** Here are some detailed observations: (1) Notably, on the hard task, the 70B model still has a higher consistency than the 1B/3B/8B model on either the hard task or the easy task. (2) Besides, the 70B model achieves a similar consistency on easy and hard tasks (0.55 and 0.52, respectively). (3) However, the 8B model drops significantly from easy to hard tasks (from 0.41 to 0.20).

## H.4 FURTHER DISCUSSION ON THE STRATEGYQA

The abnormal reasoning behavior, where states cluster on anchors that differ from their final answer in Fig. 4(c), is not due to our visualization method but to the unstable reasoning process in the Llama-3.1-70B using CoT on StrategyQA. This model struggles to reliably represent its self-generated intermediate thoughts, presenting consistency between intermediate thoughts and final predictions, thus leading to the abnormal patterns observed.

Table 3: Consistency metrics across random thoughts

| Consistency | The number of random thoughts | | | | |
|---|---|---|---|---|---|
| | 2 | 4 | 8 | 16 | 32 |
| Correct Paths | 0.77 | 0.80 | 0.80 | 0.75 | 0.66 |
| Incorrect Paths | 0.90 | 0.92 | 0.92 | 0.79 | 0.79 |

Table 4: Accuracy and consistency on MMLU and MMLU-Pro across different models.

| Model | MMLU Accuracy | MMLU Consistency | MMLU-Pro Accuracy | MMLU-Pro Consistency |
|---|---|---|---|---|
| Llama-3.2-1B Instruct | 0.20 | 0.40 | 0.05 | 0.17 |
| Llama-3.2-3B Instruct | 0.46 | 0.41 | 0.30 | 0.26 |
| Llama-3.1-8B Instruct | 0.66 | 0.41 | 0.30 | 0.20 |
| Llama-3.1-70B Instruct | **0.86** | **0.55** | **0.40** | **0.52** |

Specifically, the consistency of incorrect paths declines steadily. This highlights the model's unstable reasoning, as it fails to maintain coherent reasoning even when approaching the final answer. In addition, the landscape exhibits the highest perplexity compared to other models, indicating low confidence in its generated thoughts, which undermines the reliability of the estimated feature matrix used in our visualization.

Further, we provide landscape visualizations for the same dataset using other models and methods in Fig. 9 to Fig. 12. These landscapes do not exhibit the same abnormal density patterns, reinforcing that the issue is specific to Llama-3.1-70B's reasoning instability rather than a flaw in our visualization.

### H.5 Comparing the Perplexity among Different Models

We conduct experiments to calculate the average perplexity of models in our visualization. Consistent with the prior works, we find that different models present similar perplexity when decoding the same set of CoTs. Here, we first generate a set of CoTs from the AQuA dataset using Llama-3.1-70B Instruct. Then, we use models from the same family (*i.e.*, Llama-3.2-1B Instruct, Llama-3.2-3B Instruct, Llama-3.1-8B Instruct, and Llama-3.1-70B Instruct) to compute the average perplexity on decoding the same set of CoTs. This control experiment isolates the effect of a model's inherent perplexity calculation from the variation of its generated thoughts.

As shown in Tab. 5, while there is a slight variation in perplexity, the values all fall within a comparably narrow range (from 1.4 to 2.0). This demonstrates that for decoding the same CoTs, different models in the Llama-3 family produce similar and comparable perplexity scores.

In addition, in Fig. 2, we measure the perplexity of decoding CoTs generated by the models themselves. In this context, perplexity reflects both a model's reasoning capabilities and the comprehension of its generated content. To some extent, the above findings support the validity of the comparison of perplexity across models in our study.

### H.6 Additional Experiments on the Verifier

**Absolute Performance of the Verifier.** In this part, we provide the absolute performance of the experiment conducted in Fig. 8. Shown as Tab. 6, the results demonstrate that our approach consistently provides improvements across different domains and models.

**Variants of Verifier.** In this part, we extend it into a process verifier and validate its effectiveness through additional experiments. Our lightweight verifier functions as an outcome reward model (ORM), assessing the correctness of an entire reasoning path. Specifically, the process verifier predicts the accuracy of each reasoning state using features from the current and all previous thoughts. State accuracy reflects whether the current state is closer to the correct answer (measured

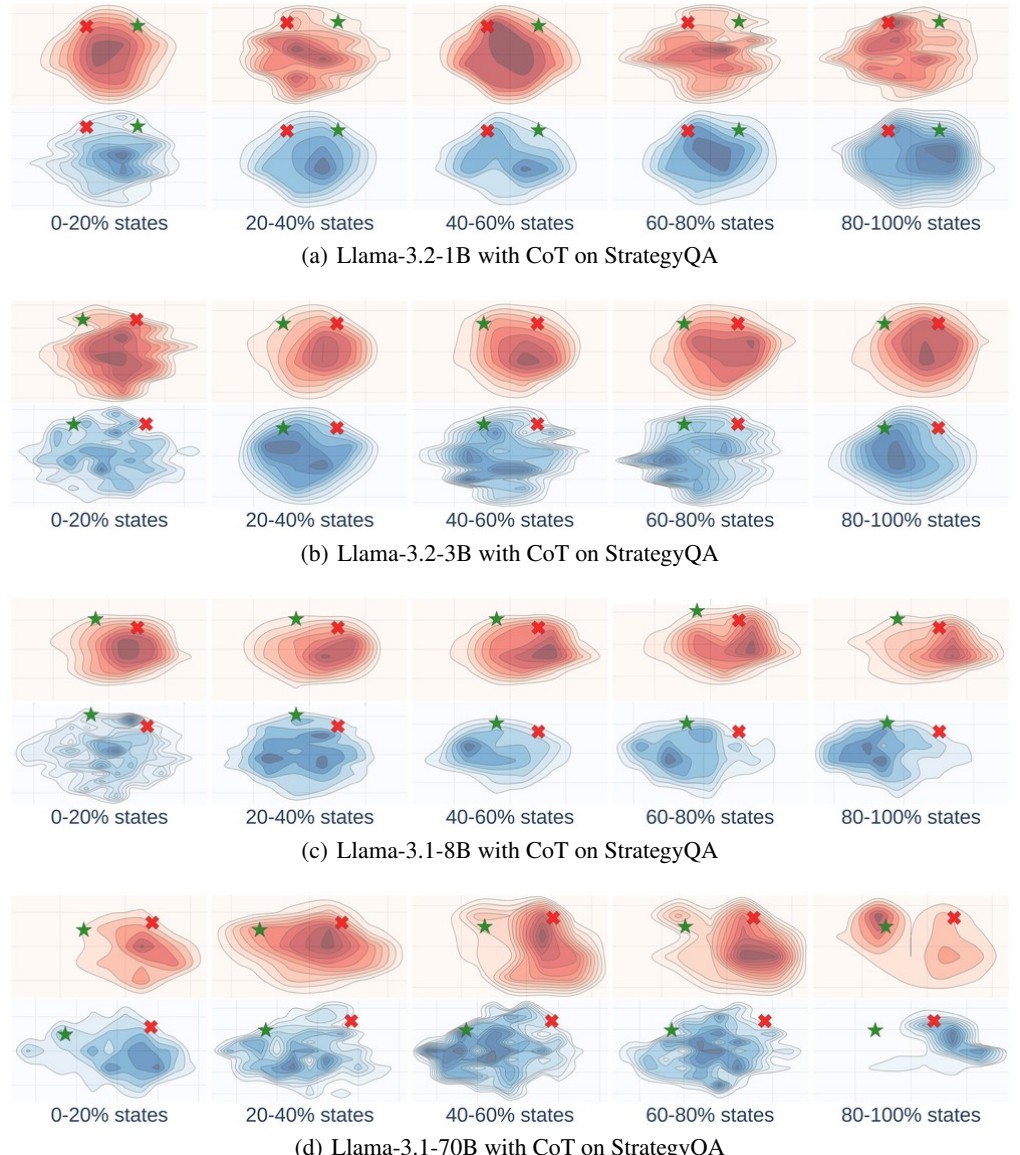

Figure 9: The landscapes of the model across scales (using CoT on the StrategyQA dataset).

Table 5: Comparison of the perplexity of CoTs of correct and incorrect reasoning.

| Model | Avg. Perplexity (Correct CoTs) | Avg. Perplexity (Wrong CoTs) |
|---|---|---|
| Llama-3.2-1B Instruct | 1.68 | 1.96 |
| Llama-3.2-3B Instruct | 1.72 | 1.69 |
| Llama-3.1-8B Instruct | 1.61 | 1.49 |
| Llama-3.1-70B Instruct | 1.56 | 1.42 |

by perplexity) than other answers. We then aggregate these predictions across the chain to estimate overall accuracy.

Empirically, we collect the state-wise data by comparing the state features and the correct answers, and train the process verifier. Note, we do not need to manually annotate the step-wise rewards

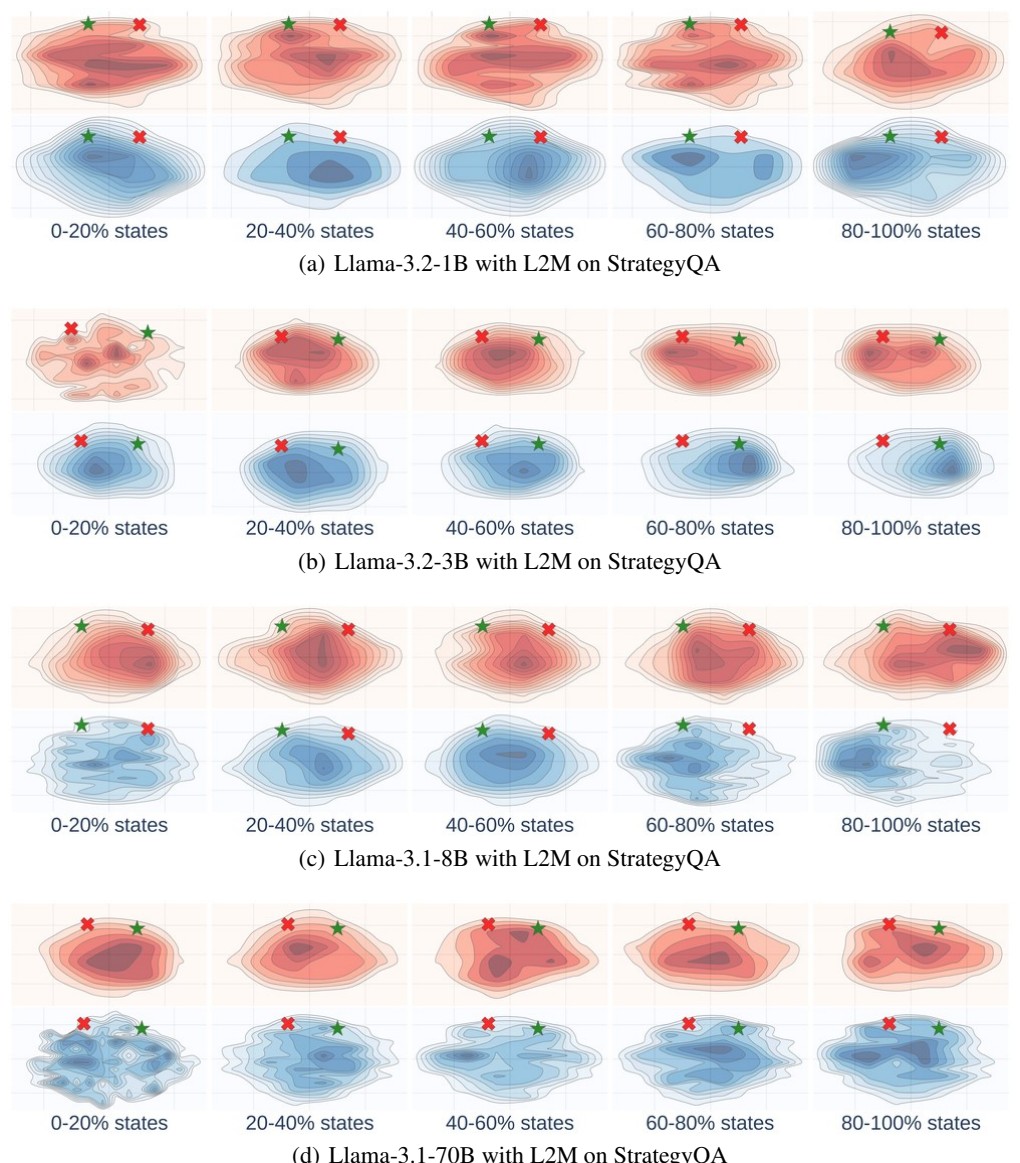

(a) Llama-3.2-1B with L2M on StrategyQA

(b) Llama-3.2-3B with L2M on StrategyQA

(c) Llama-3.1-8B with L2M on StrategyQA

(d) Llama-3.1-70B with L2M on StrategyQA

Figure 10: The landscapes of the model across scales (using L2M on the StrategyQA dataset).

Table 6: Absolute accuracy with the verifier, compared to performance in Fig. 6 (without the verifier).

(a) Across datasets

|  | AQuA | MMLU | StrategyQA | Common SenseQA |
|---|---|---|---|---|
| AQuA | 63.0 (+0.7) | 62.3 (+0.0) | 62.3 (+0.0) | 64.0 (+1.7) |
| MMLU | 53.0 (+0.0) | 53.0 (+0.0) | 53.0 (+0.0) | 53.0 (+0.0) |
| StrategyQA | 41.5 (+4.5) | 40.5 (+3.5) | 43.0 (+6.0) | 37.0 (+0.0) |
| Common SenseQA | 54.0 (+1.0) | 53.0 (+0.0) | 53.0 (+0.0) | 54.0 (+1.0) |

(b) Across models

|  | 1B | 3B | 8B | 70B |
|---|---|---|---|---|
| 1B | 26.0 (+0.5) | 27.5 (+2.0) | 27.5 (+2.0) | 27.5 (+2.0) |
| 3B | 45.5 (+0.0) | 48.0 (+2.5) | 51.0 (+5.5) | 51.0 (+5.5) |
| 8B | 60.0 (+0.0) | 60.0 (+0.0) | 60.0 (+0.0) | 60.0 (+0.0) |
| 70B | 74.0 (+2.0) | 73.0 (+1.0) | 72.5 (+0.5) | 72.5 (+0.5) |

to train conventional PRMs. Results in Tab. 7 show that this process verifier is comparable to the outcome verifier.

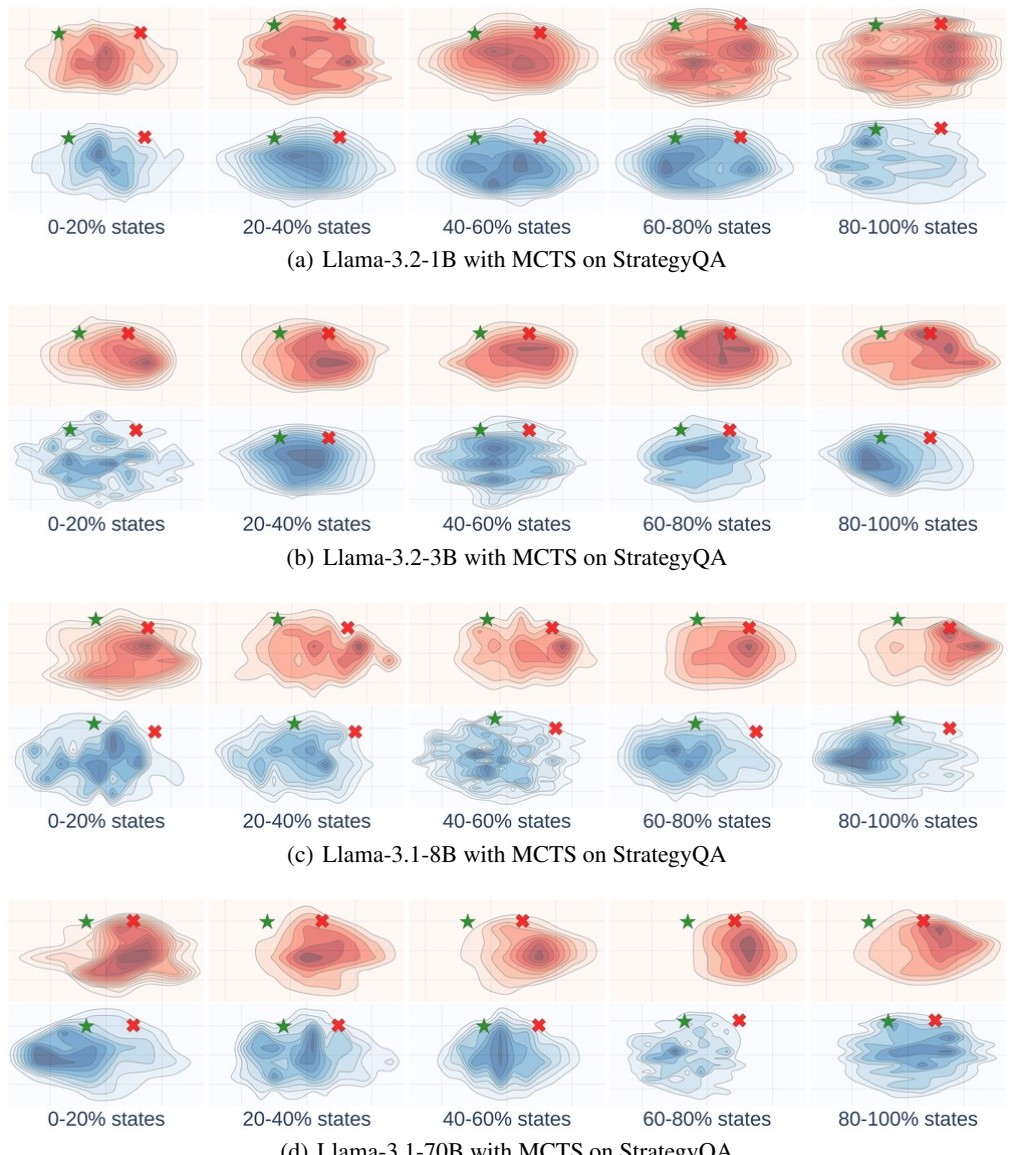

(a) Llama-3.2-1B with MCTS on StrategyQA

(b) Llama-3.2-3B with MCTS on StrategyQA

(c) Llama-3.1-8B with MCTS on StrategyQA

(d) Llama-3.1-70B with MCTS on StrategyQA

Figure 11: The landscapes of the model across scales (using MCTS on the StrategyQA dataset).

**Comparing the lightweight verifier with existing verifiers.** In the following, we compare our lightweight verifier with the other two types of existing verifiers: the LM-based verifier and the model-self verifier.

The LM-based verifier leverages another powerful LLM (not the model to do reasoning) to semantically analyze reasoning trajectories, mimicking human expert evaluation to detect errors in the trajectories. These verifiers rely on extensive, specially curated datasets (e.g., PRM800k (Lightman et al., 2024)) to train a language model for process verification. Here, collecting high-quality training data often requires detailed, fine-grained annotations of reasoning steps, which can be costly and time-consuming. Moreover, training this verifier (often itself a large language model) incurs much additional computational expense. In contrast, our lightweight verifier is much more efficient to train, as it requires no human annotations and only uses easily obtainable data that is collected from the model to do reasoning.

As for the model-self verifier (Li et al., 2023a; Xiong et al., 2024), it utilizes features derived from the model itself, such as uncertainty, perplexity, or entropy, eliminating the need for an external model

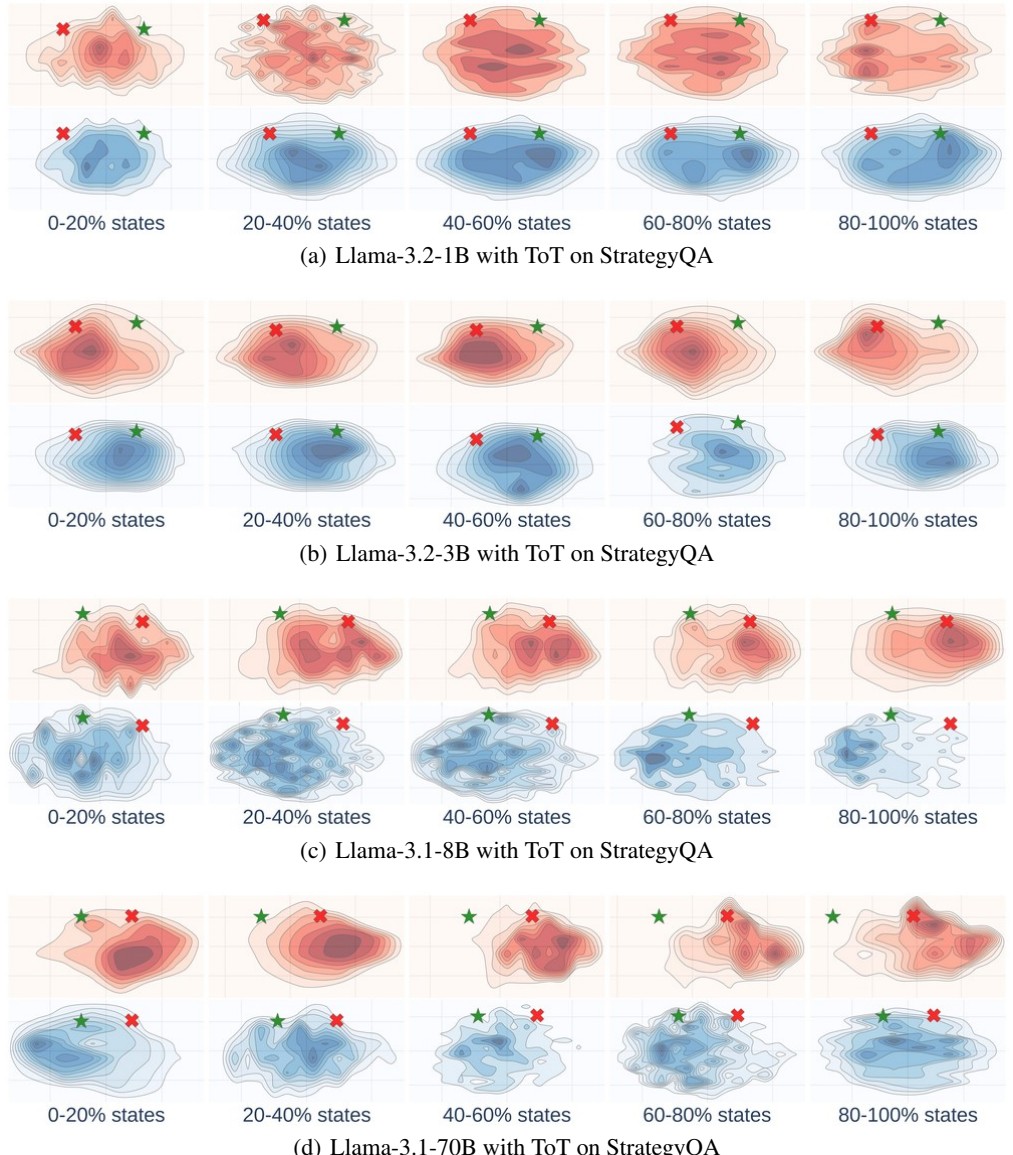

(a) Llama-3.2-1B with ToT on StrategyQA

(b) Llama-3.2-3B with ToT on StrategyQA

(c) Llama-3.1-8B with ToT on StrategyQA

(d) Llama-3.1-70B with ToT on StrategyQA

Figure 12: The landscapes of the model across scales (using ToT on the StrategyQA dataset).

and enhancing efficiency in search-based methods. While these model-self verifiers are training-free and efficient, they lack the learnability to be trained and optimized, as the model is not trained on the downstream task, and thus it can be suboptimal. In contrast, our verifier is specifically trained with the downstream task's data collected from the model, ensuring greater reliability compared to model-self verifiers.

Therefore, our landscape-based lightweight verifier offers distinct advantages in terms of efficiency and reliability over the other two types of verifiers.

**Ablation study on verifier.** We conduct an extra ablation study on training the verifier with either consistency or 2D information. We report the accuracy of reasoning under Least-to-Most with different model scales, averaged across different datasets.

As shown in the Tab. 8, the combination of the consistency score and 2D information delivers the best overall accuracy. This shows that our verifier could utilize the complementary aspects of both kinds of features to access the reasoning chains and thus boost reasoning accuracy.

Table 7: Performance comparison of reasoning methods across model scales on the AQuA dataset, with and without verifiers.

| Model | Method | Without Verifier | With Outcome Verifier | With Process Verifier |
|---|---|---|---|---|
| Llama-3.2-1B | CoT | 0.26 | 0.28 | 0.26 |
| | L2M | 0.22 | 0.24 | 0.29 |
| | ToT | 0.35 | 0.38 | 0.35 |
| | MCTS | 0.29 | 0.32 | 0.31 |
| Llama-3.2-3B | CoT | 0.46 | 0.51 | 0.46 |
| | L2M | 0.29 | 0.31 | 0.31 |
| | ToT | 0.33 | 0.35 | 0.33 |
| | MCTS | 0.35 | 0.36 | 0.35 |
| Llama-3.1-8B | CoT | 0.60 | 0.63 | 0.60 |
| | L2M | 0.58 | 0.62 | 0.58 |
| | ToT | 0.50 | 0.53 | 0.50 |
| | MCTS | 0.50 | 0.51 | 0.50 |
| Llama-3.1-70B | CoT | 0.72 | 0.73 | 0.73 |
| | L2M | 0.72 | 0.72 | 0.73 |
| | ToT | 0.74 | 0.74 | 0.74 |
| | MCTS | 0.72 | 0.73 | 0.72 |

Table 8: Ablation study on data employed for training the verifier.

| | 1B | 3B | 8B | 70B |
|---|---|---|---|---|
| **Consistency only** | 0.21 | 0.31 | 0.59 | 0.71 |
| **2D information only** | 0.20 | 0.31 | 0.61 | 0.71 |
| **Consistency + 2D information** | 0.24 | 0.31 | 0.62 | 0.72 |

## H.7 FURTHER EXPERIMENTS ON THE SCALING EFFECT

We present experiments and demonstrate that combining both information sources is the best choice, with significant gains from more sampled trajectories (i.e., test-time scaling) compared to the verifier trained with either feature, as can be seen in Tab. 9. Here, we report the accuracy using the Llama-3.2-3B Instruct model on the StrategyQA dataset as follows. As can be seen, the advantages of using both information sources increase with more sampled trajectories, especially for more than 20 sampled trajectories. In contrast, verifiers trained only on consistency or 2D information peak earlier, showing no notable performance gains beyond 10 sampled trajectories.

## H.8 LANDSCAPES WITH DIFFERENT METHODS OF DIMENSIONALITY REDUCTION

t-SNE is widely adopted in non-linear projection for visualisations, which makes the plots more interpretable. Beyond t-SNE (Cai & Ma, 2022), several advanced dimensionality reduction techniques have been developed to improve visualization quality and efficiency. UMAP (McInnes et al., 2018) outperforms t-SNE by better balancing local and global structure preservation while offering greater speed and scalability for large datasets. TriMAP (Amid & Warmuth, 2019) prioritizes both local and global preservation but tends to emphasize global structure in practice, potentially at the expense of local details. PaCMAP (Wang et al., 2021) achieves a robust balance between local and global structure preservation by incorporating neighbors, mid-near points, and further points, resulting in high-quality visualizations across diverse scenarios.

In addition, our goal is to develop a visualization tool to help users analyze the reasoning behaviors of LLMs. If necessary, we can change the adopted t-SNE to more advanced methods of dimensionality reduction. Our tool is designed to be compatible with these methods.

Next, we experiment with different dimensionality reduction methods, including t-SNE, UMAP, and PacMAP, to visualize the landscape. **Across all three visualization techniques, we consistently observe the same overarching dynamics in the reasoning process.** In the early stages (0–40% of states), the thought states are widely dispersed. As reasoning progresses, states gradually converge

Table 9: Performance of the verifier given different numbers of sampled paths.

| Sampled Paths | Consistency | 2D Information | Consistency + 2D Information |
|---|---|---|---|
| 1 | 0.32 | 0.32 | **0.32** |
| 10 | 0.32 | 0.32 | **0.34** |
| 20 | 0.32 | 0.30 | **0.46** |
| 30 | 0.32 | 0.36 | **0.56** |
| 40 | 0.32 | 0.34 | **0.68** |
| 50 | 0.32 | 0.30 | **0.66** |

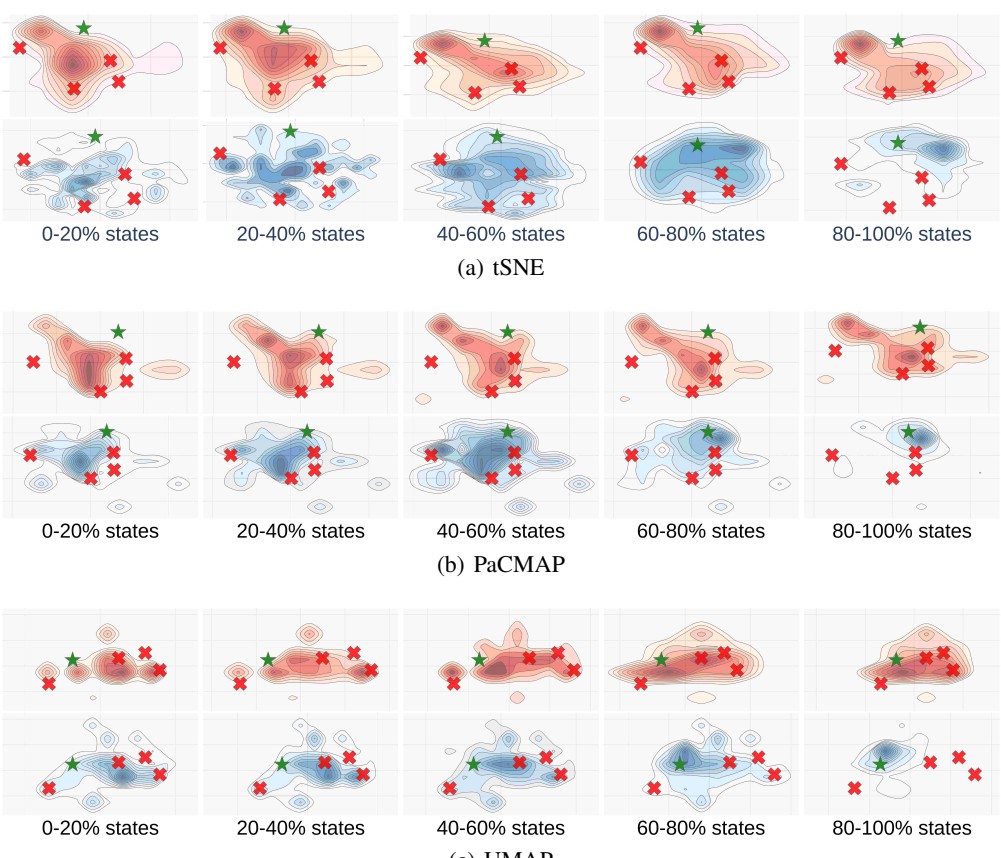

Figure 13: The landscapes of thought visualization with different dimensionality reduction methods (Llama-3.1-70B with CoT on AQuA).

toward the final answer choices. Importantly, a clear distinction emerges between correct and incorrect reasoning paths, regardless of the selection of different dimensionality reduction methods. Incorrect paths tend to converge rapidly toward wrong answers early in the process, while correct paths exhibit a more gradual and deliberate progression, only clustering tightly around the correct answer in the final stages (80–100% of states).

We provide landscape visualizations in Fig. 13 with different dimensionality reduction methods. While the specific geometry and density of clusters may vary between t-SNE, UMAP, and PacMAP, the fundamental narrative is unchanged: the landscape of thoughts consistently reveals that incorrect reasoning solidifies quickly, whereas correct reasoning is characterized by a slower, more refined convergence. This consistency across different dimensionality reduction algorithms demonstrates that our observations are not artifacts of a particular visualization technique, but rather reflect intrinsic properties of the model's reasoning process.

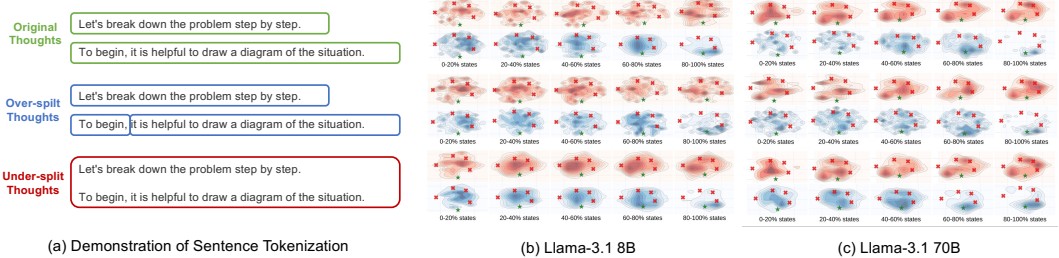

Figure 14: Demonstration of sentence tokenization methods for thoughts splitting.

## H.9 ROBUSTNESS OF SENTENCE TOKENIZATION

To evaluate the robustness of the landscape to the split thoughts' information volume, *i.e.*, the granularity of the sentence tokenization, we conduct a controlled experiment by considering two imperfect cases in thought split, namely over-split thoughts and under-split thoughts.

Specifically, shown as Fig. 14 (a), compared to the original thoughts split that transform sentences to thoughts based on the period, over-split thoughts jointly consider the comma, resulting in additional splits. For the under-split, two adjacent thoughts are merged into one thought. We then visualize the imperfect thought splits using CoT on AQuA following the setting in Fig. 5(a) and Fig. 2(c),

Shown in Fig. 14 (b) and (c), the landscapes are robust to the split thoughts' information volume, which are stable and consistent with our observations. Notably, for over-split thoughts, the states are more visually diverse but eventually converge to the answers. Whereas under-split thoughts, the states show a more compact pattern and exhibit a clear convergence trend toward the answer.

## H.10 GENERALIZATION TO TASKS BEYOND MULTI-CHOICE PROBLEMS

Here, we provide new empirical results on open-ended benchmarks MATH, GSM8k, and StrongReject to show the visualization effectiveness of LoT in *open-ended questions*, where the observations in these datasets highly align with our observations in multiple-choice tasks.

Note that the multiple-choice restriction is pragmatic, not fundamental, and a simple pseudo-option strategy can address this restriction. LoT construction requires anchors to compute relative distances, as defined in Equation (1), where the feature $f_i$ for state $s_i$ quantifies distances to choices $\{c_j\}_{j=1}^k$ via perplexity in Equation (2). In open-ended settings, these do not exist naturally, but they can be created reliably. We tested two complementary families:

- For rule-verifiable reasoning (MATH and GSM8K), we keep the unique correct answer and use Llama-3.1-8B-Instruct to generate three plausible incorrect final answers plus full reasoning chains, followed by manual filtering to retain only non-trivial distractors, yielding clean 4-way multiple-choice versions.

- For non-rule-verifiable or safety tasks (StrongReject jailbreak benchmark), we use True/False as the two options, with ground-truth labels produced by GPT-4o and manually verified.

This approach mirrors the original setup in Section 2.2, ensuring the perplexity-based distances remain meaningful without altering the core visualization pipeline.

Then, following the experiment setting described in Sec. 2, We sampled 10 trajectories per question using the Llama-3.1-8B Instruct and constructed the LoT accordingly. Figures 21 (MATH), 22 (GSM8K), and 23 (StrongReject) show the phenomena replicate strongly. Key observations hold clearly in open-ended settings. We describe them as follows:

- Observation 3.4 (Similar reasoning tasks exhibit similar landscapes). MATH and GSM8K exhibit rich state diversity and phased exploration patterns highly similar to multiple-choice datasets (AQuA, MMLU, and StrategyQA in Fig. 4), whereas StrongReject shows concentrated, low-diversity search regions akin to CommonSenseQA.

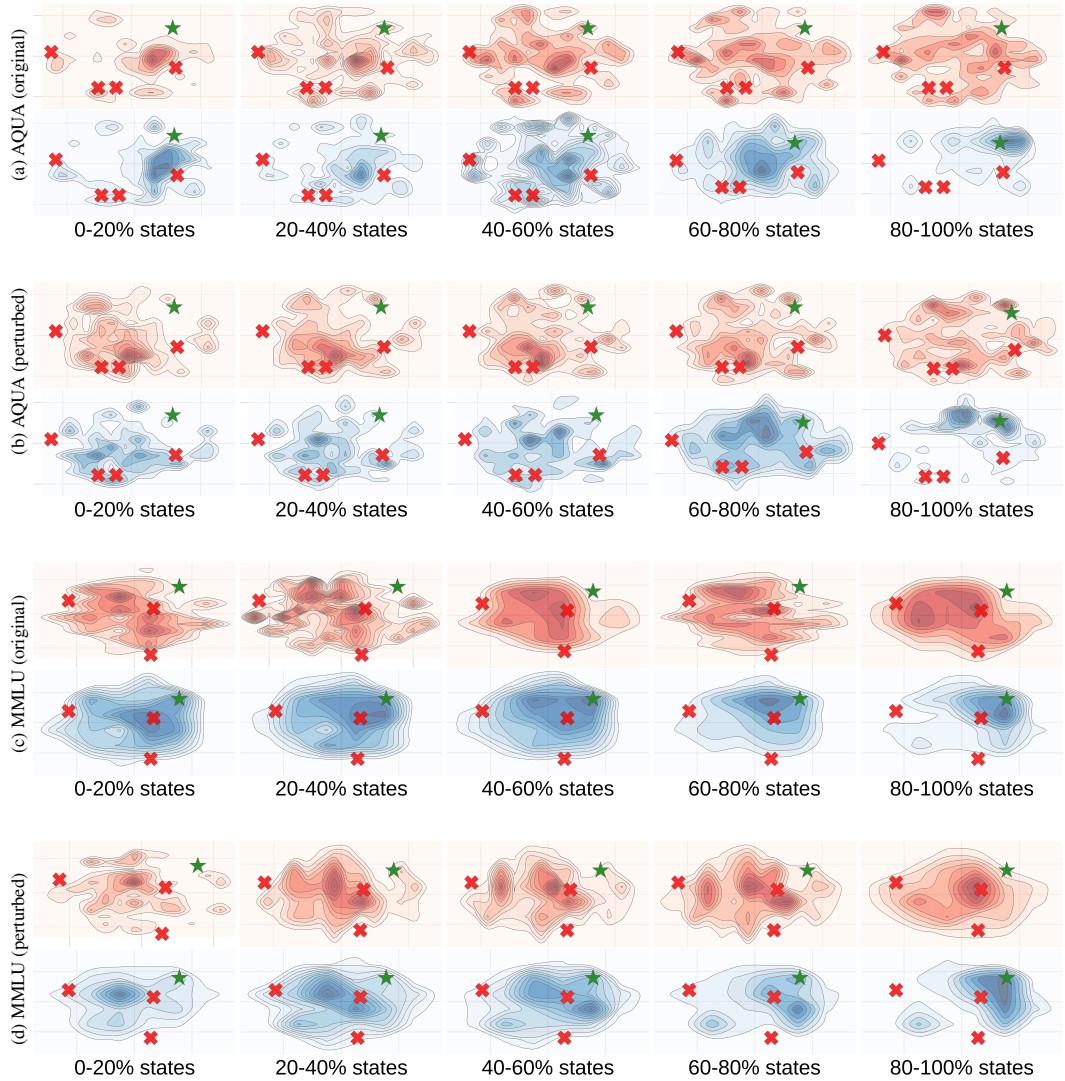

Figure 15: Comparing the LoT of perturbed options (using Llama-3.1-8B with CoT).

- Observation 3.7 (Within-method comparison: For any single method, incorrect trajectories converge faster to wrong answers than correct trajectories converge to right answers.) Across all three open-ended benchmarks, incorrect trajectories rapidly approach wrong anchors in 60–80% of reasoning states, while correct trajectories explore longer and only converge in the final 80-100% of states, mirroring the pattern in Figs. 2, 5, and 4 of multiple-choice questions.

The core insights, therefore, transfer to open-ended questions without choices. These observations support that the LoT can be employed to analyze the reasoning process of open-ended questions.

Further, to eliminate the concern that our current results might depend on the quality of pseudo-options generated by another model or on manual filtering, we envision a fully automatic anchor discovery approach as future work. Specifically, after sampling multiple trajectories for an open-ended question, the final answers can be embedded with a strong embedding model and clustered in semantic space. Task-specific verifiers, e.g., exact-match checks for math problems or safety classifiers for jailbreak evaluation, can automatically label the highest-quality cluster as the correct anchor and the remaining clusters as incorrect anchors. This procedure requires no auxiliary model and no human intervention, making it more scalable while preserving the computations of the perplexity metric.

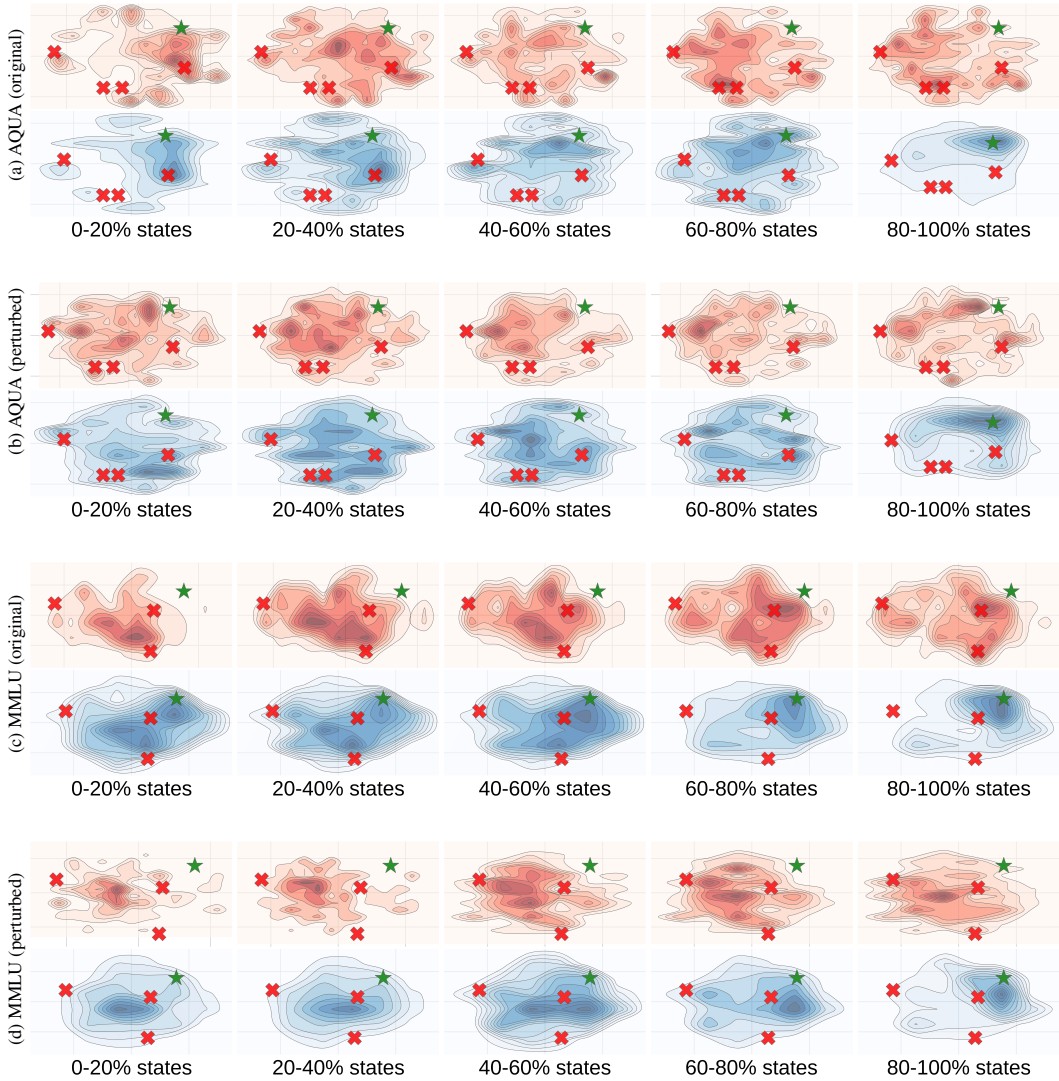

Figure 16: Comparing the LoT of perturbed options (using Llama-3.1-8B with L2M).

## H.11 LANDSCAPE VISUALIZATION WITH CHOICE REORDERING.

We swap the ordering of answer options (for example, A) $c_1$, B) $c_2 \to$ B) $c_1$, A) $c_2$) while keeping the semantic content of each $c_j$ unchanged. We then recompute state features, rebuild the landscapes, and replot the consistency, uncertainty, and perplexity curves. As shown in Fig. 16, convergence patterns, separation between correct and incorrect trajectories, and the overall geometry are effectively unchanged; the metric curves almost overlap. This indicates that LoT is insensitive to superficial label permutations.

## H.12 LANDSCAPE VISUALIZATION WITH MODEL BEYOND LLAMA FAMILY.

To further test the generality of LoT, we apply LoT to `Qwen-2.5-7B Instruct` and `Qwen-2.5-72B Instruct` and visualize their landscapes (Fig. 17). Notalby, we observed that: (1) Early states (0-20% of steps) are more dispersed in the landscape; (2) As states progress (20-80%), they become more organized and form clearer structures in LoT space; (3) Incorrect trajectories (red in the top rows) move toward wrong answers already in early and mid stages (for example, 20-40%), consistent with the option-directed behavior reported in Observation 3.7; The

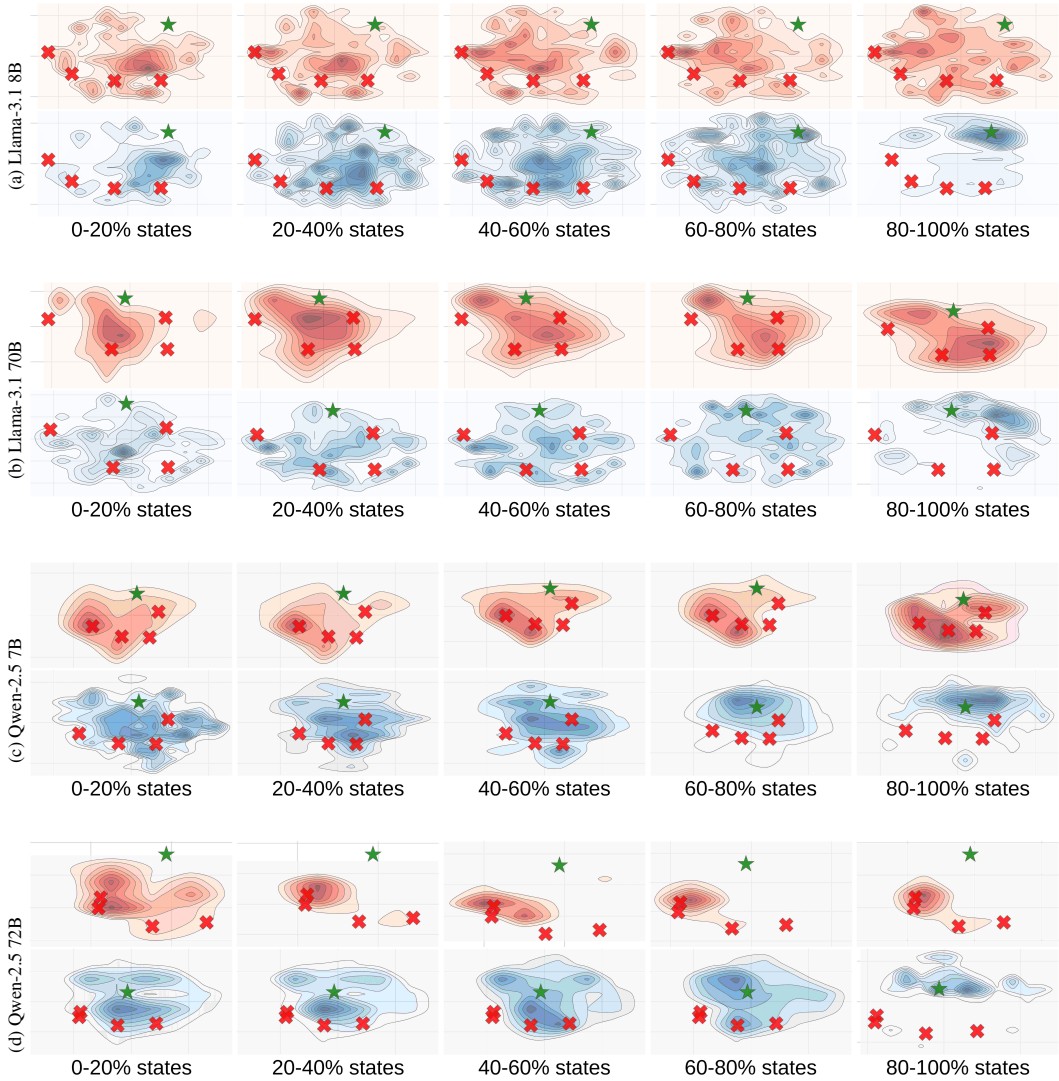

Figure 17: Comparing the LoT of different model families and scales (Sampling with CoT on AQUA).

separation between correct and incorrect trajectories remains evident: correct states stay closer to the correct option over many steps and show higher stability (higher consistency and lower uncertainty), whereas incorrect states exhibit reduced stability.

The fact that these patterns appear both in the Llama family and in Qwen 2.5 models indicates that LoT is capturing broader properties of autoregressive chain-of-thought reasoning, not idiosyncrasies of a specific Llama line. Taken together, the model family comparison (Fig. 17), cross-scale trends (Fig. 2), cross-task differences (Fig. 4), and the predictive utility of LoT features (Fig. 7) provide converging evidence that our observations reflect general reasoning behaviors rather than phenomena restricted to the Llama family.

## H.13 LANDSCAPE GEOMETRY IS ROBUST ACROSS ROLLOUT SAMPLE SIZES

From a methodological perspective, LoT aggregates traces in a way that is inherently stable to sample size. The reason is that LoT operates in a fixed feature space where each state is represented by a distance vector $f_i$ over answer options, and all state features are stacked into a matrix $S$ before dimensionality reduction. Adding more trajectories simply adds more rows to $S$ from the same underlying distribution of states. This increases the sampling density of the manifold but does not

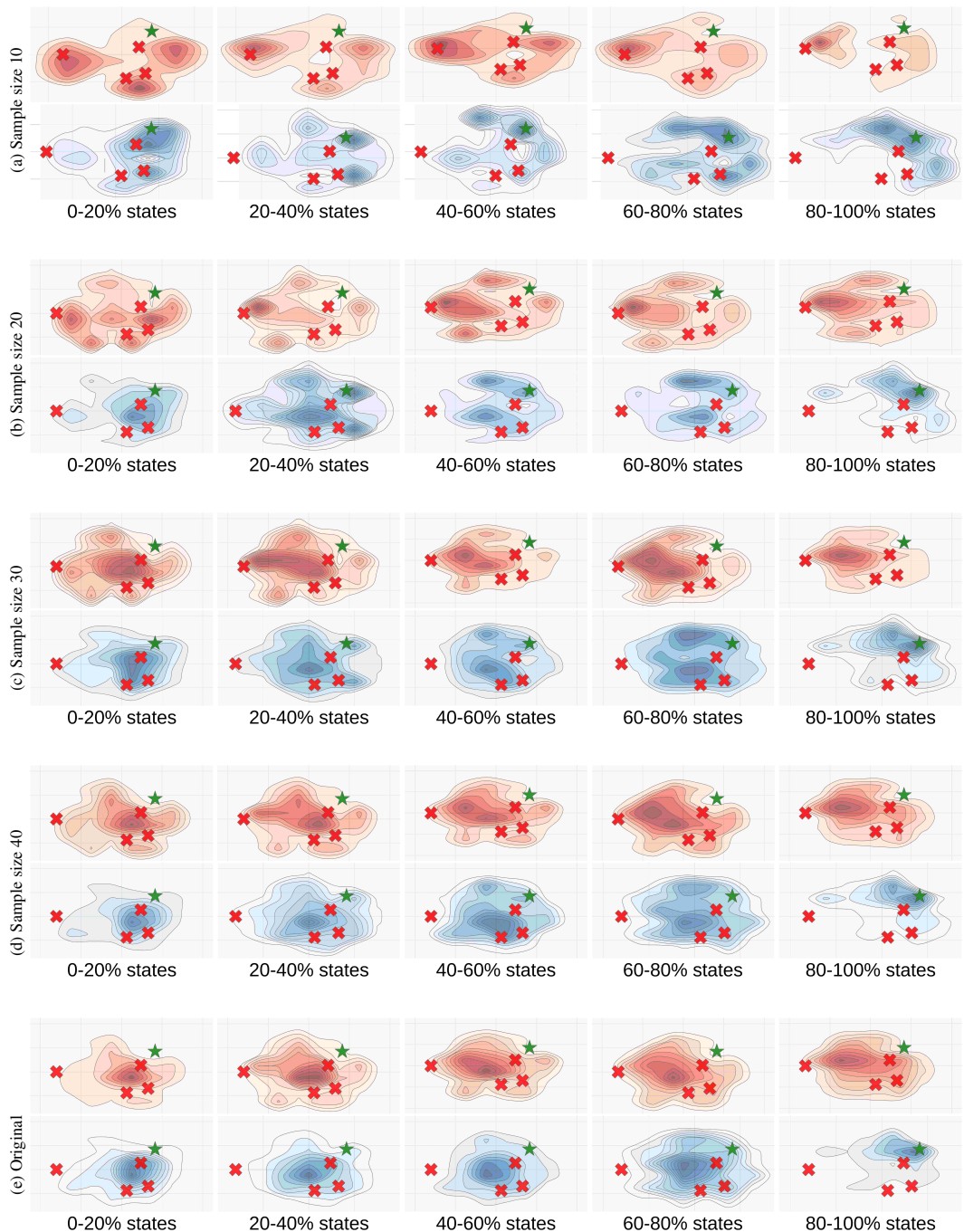

Figure 18: Comparing the LoT of different sample sizes (using Llama-3.1-8B with CoT on AQUA).

redefine the feature space or the embedding function, so the global geometry of the landscape is expected to remain stable as the sample size grows.

Here, we explicitly test this by generating landscapes with different numbers of sampled trajectories per question: 10, 20, 30, 40, and 50, used in the original landscape. Across all these configurations (reported in Fig. 18), the resulting landscapes exhibit the same qualitative structure: (1) Correct trajectories gradually converge toward the correct region; (2) Incorrect trajectories collapse earlier

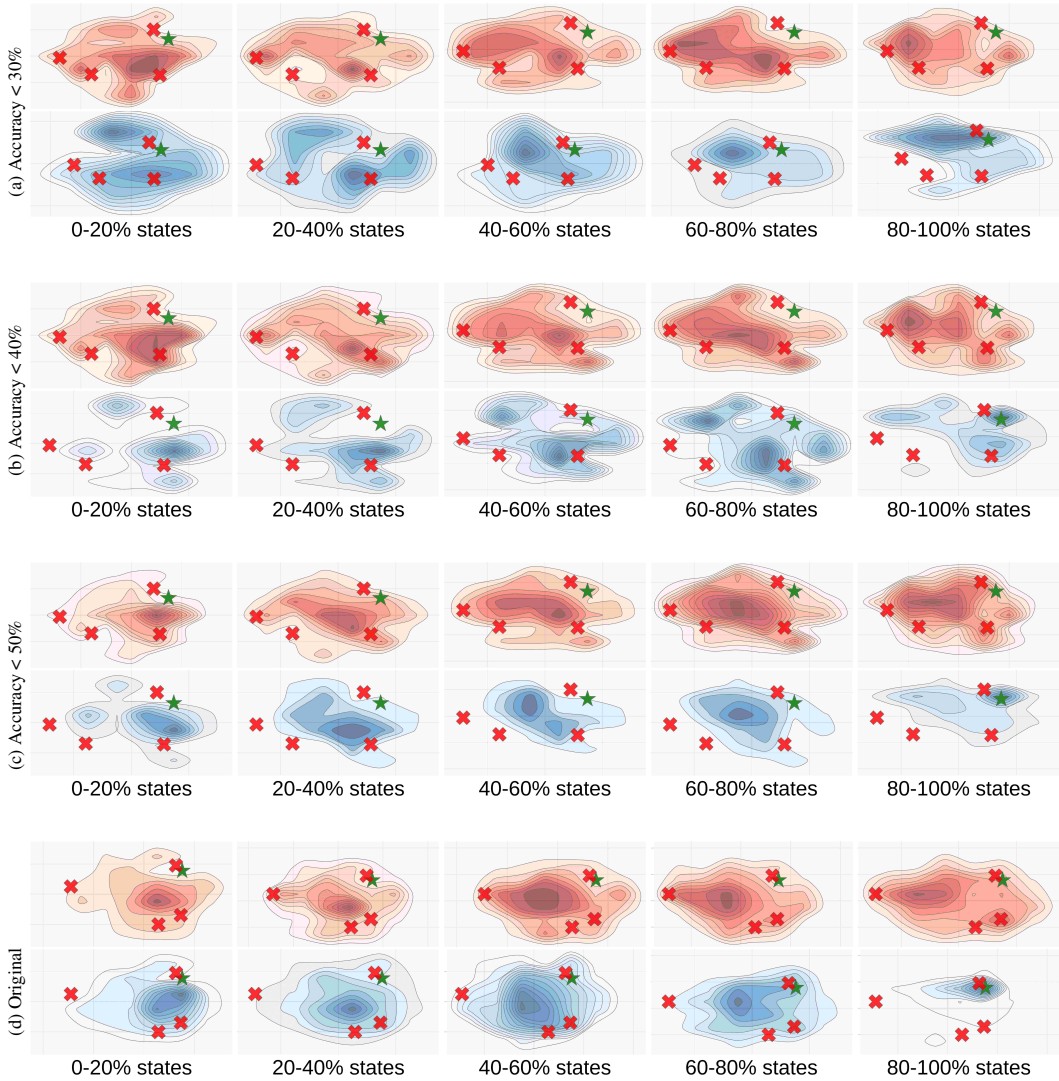

Figure 19: Comparing the LoT of different accuracy thresholds (using Llama-3.1-8B with CoT on AQUA).

toward wrong answer anchors; (3) The relative placement of major clusters is unchanged; (4) the evolution of density over time is smooth and monotonic in sample size.

The only noticeable difference is visual: landscapes with more samples (for example, 40 or 50 rollouts) appear smoother and less noisy because the underlying density is better estimated, but the shapes of clusters, the separation between correct and incorrect trajectories, and the convergence patterns remain the same.

### H.14 LANDSCAPE ON CHALLENGING QUESTIONS.

The main LoT structure remains stable across all difficulty thresholds. We examined LoT on progressively harder subsets by filtering out easy questions using accuracy thresholds of $< 50\%$, $< 40\%$, and $< 30\%$ (Fig. 19). Across all cases, the characteristic patterns documented in Sec. 3 remain visible: early-stage states are more dispersed, mid-stage states exhibit partial organization, and later-stage states move more clearly toward specific answer options. This confirms that the core LoT behaviors persist even when accuracy is low.

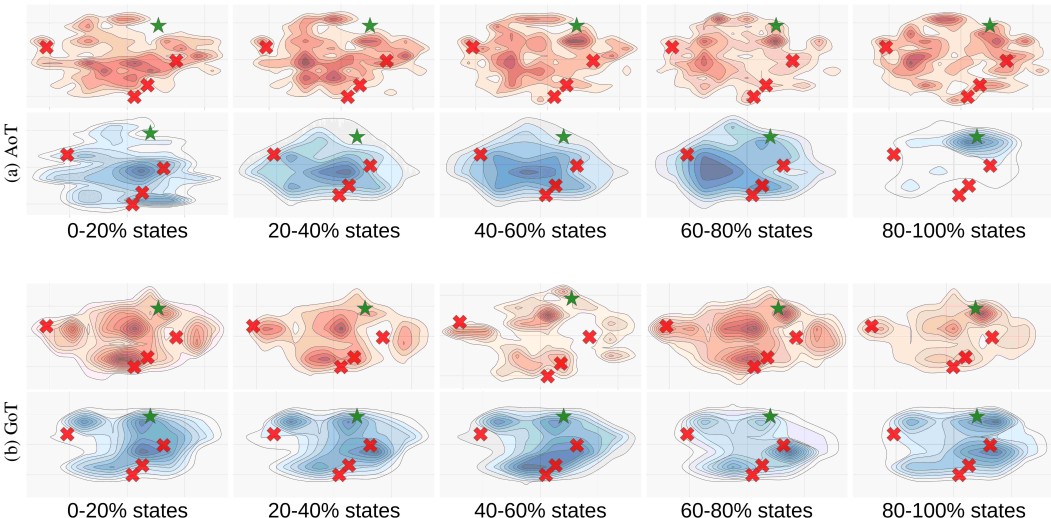

Figure 20: Comparing the LoT of different reasoning methods (using Llama-3.1-8B on AQUA).

Incorrect trajectories continue to show clear option-directed movement. The incorrect trajectories (red) still exhibit recognizable movement toward specific answer choices as the reasoning progresses. This behavior is visible across all state ranges and does not disappear even when correct trajectories become rare. Thus, LoT continues to capture meaningful convergence tendencies for incorrect reasoning.

Late-stage concentration becomes weaker as accuracy decreases. In the original landscape, the later-state incorrect trajectories form more compact regions. However, in the challenging subsets, 60-80% and 80-100% of states show broader and less sharply grouped red regions. This indicates that the model becomes less decisive during late reasoning stages on more challenging questions, as also discussed in Observation 3.6. The filtered landscapes make this phenomenon more pronounced.

The remaining correct trajectories still follow the expected patterns. Although correct trajectories become sparse in harder subsets, the ones that remain continue to align closely with the correct option across most state ranges. Their intermediate states show higher stability, and their later states form clear clusters near the correct anchor. These behaviors are consistent with Observation 3.1 and demonstrate that LoT continues to reflect successful reasoning even when questions are challenging.

The contrast between correct and incorrect trajectories becomes more pronounced on difficult questions. As the questions become harder, incorrect states become increasingly dispersed, while the remaining correct states stay compact. This highlights a clearer distinction between stable reasoning and unstable reasoning in the challenging subsets, reinforcing the interpretive value of LoT even when accuracy is low.

In summary, LoT remains informative and interpretable even when the dataset consists primarily of incorrect trajectories. The landscapes of challenging questions preserve the core patterns described in Sec. 3 while revealing additional difficulty-specific phenomena such as greater mid-stage dispersion and weaker late-stage grouping.

## H.15 LANDSCAPE VISUALIZATION ON NEW REASONING METHODS

We extend the LoT to two additional methods: Algorithm-of-Thoughts (AoT) (Sel et al., 2024) and Graph-of-Thoughts (GoT) (Besta et al., 2024), using their public implementations with Llama-3.1-8B and applying the same visualization procedure described in Sec. 3. The corresponding landscapes are shown in Fig. 20, and the reasoning accuracies for AoT and GoT are 0.54 and 0.30, respectively. In the following, we summarize our observations on these two methods.

Both Algorithm of Thoughts (AoT) and Graph of Thoughts (GoT) produce distinct landscapes but preserve the characteristic LoT patterns reported in the submission. Despite differences in the

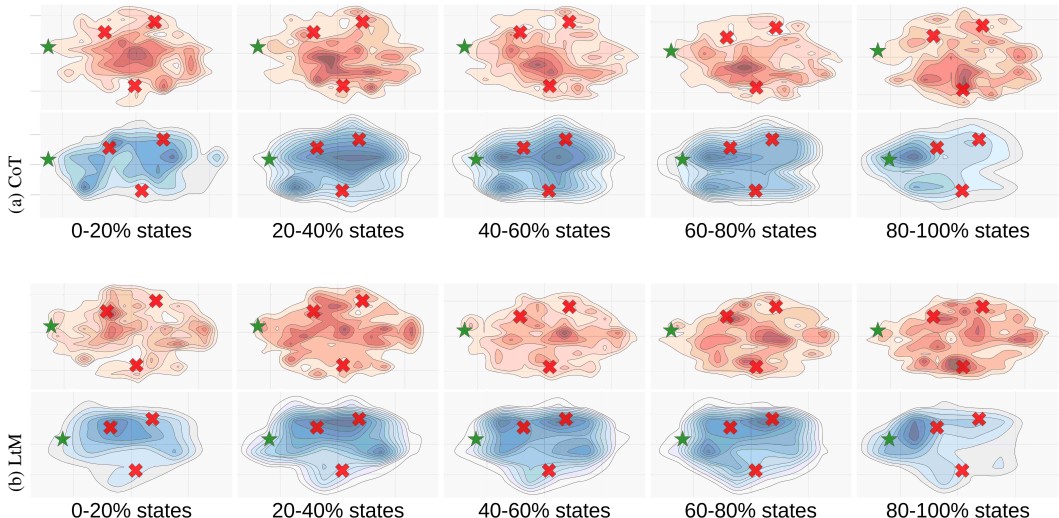

Figure 21: Comparing the LoT of different reasoning methods (using Llama-3.1-8B on MATH).

underlying reasoning strategies, both methods show the same core behaviors described in Sec. 3. Namely, early states (0-20%) are widely dispersed for both correct and incorrect trajectories. Notably, mid-range states (20-60%) begin to exhibit more organization, and later states (60-100%) show clearer movement toward specific answer options. This confirms that LoT continues to reveal a stable structure across diverse reasoning algorithms.

Correct trajectories in AoT and GoT follow the same progression observed in the original analysis. For both reasoning methods, the correct trajectories remain closer to the correct answer across state ranges and display higher stability, consistent with Obs 3.1 and 3.2. AoT, which achieves higher accuracy (0.54), exhibits more concentrated blue regions in the 40-60% and 60-80% state ranges, reflecting more consistent reasoning. GoT, with lower accuracy (0.30), shows correct trajectories that are fewer and more spread out, but still maintains recognizable proximity to the correct option in later states.

Incorrect trajectories show early and clear movement toward wrong answers, consistent with the original LoT observations. In both AoT and GoT, the incorrect trajectories (red) begin to cluster toward specific wrong answer options within the first 20-40% of states. This matches the behavior described in Sec. 3, where incorrect paths tend to settle into wrong answers earlier in the reasoning process. The effect is especially pronounced for GoT, whose lower accuracy results in a broader and less stable distribution of mid-stage states.

In summary, the AoT and GoT experiments show that LoT captures consistent reasoning patterns across methods with different structures and accuracies. The landscapes in Fig. 20 reproduce the key behaviors described in Sec. 3 and additionally reflect the performance differences between methods.

## I VISULIZATIONS

In this part, we provide the full visualization of the verifier performance and landscapes.

In Fig. 24 to Fig. 27, we visualize the average voting accuracy (%) of different LLMs reasoning with and without verification on various datasets and methods. In Fig. 28 to Fig. 31, we display the landscape of different models on various datasets using four methods. We also provide case studies by visualizing the landscape with corresponding states in Fig 32 to Fig. 35.

In addition, we provide the landscape of thoughts on the latest reasoning model. Specifically, we conduct experiments on the DeepSeek-R1-Distill models (Guo et al., 2025) (Llama-70 B and Qwen-1.5 B). As shown in Fig. 37 and Fig. 38, the landscape of the reasoning model also aligns with the

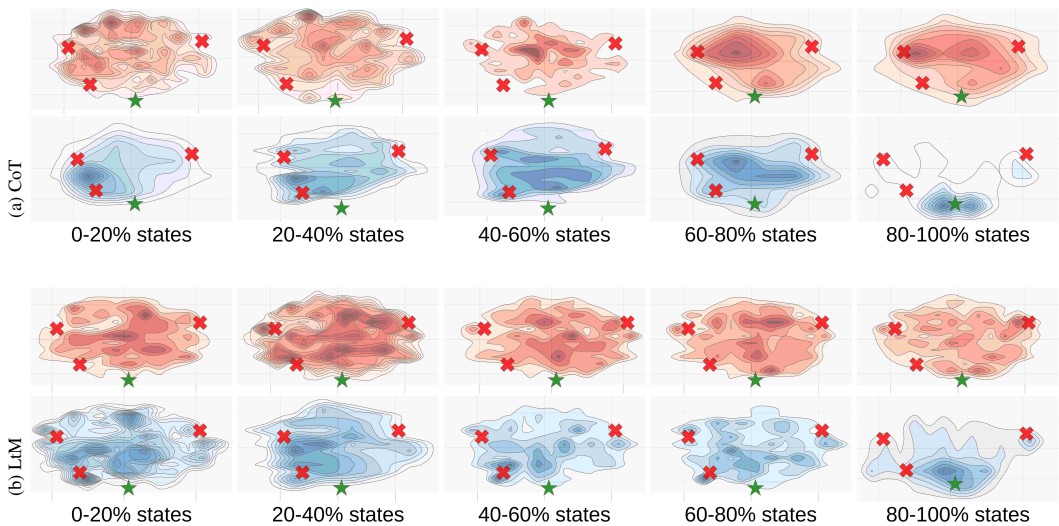

Figure 22: Comparing the LoT of different reasoning methods (using Llama-3.1-8B on GSM8K).

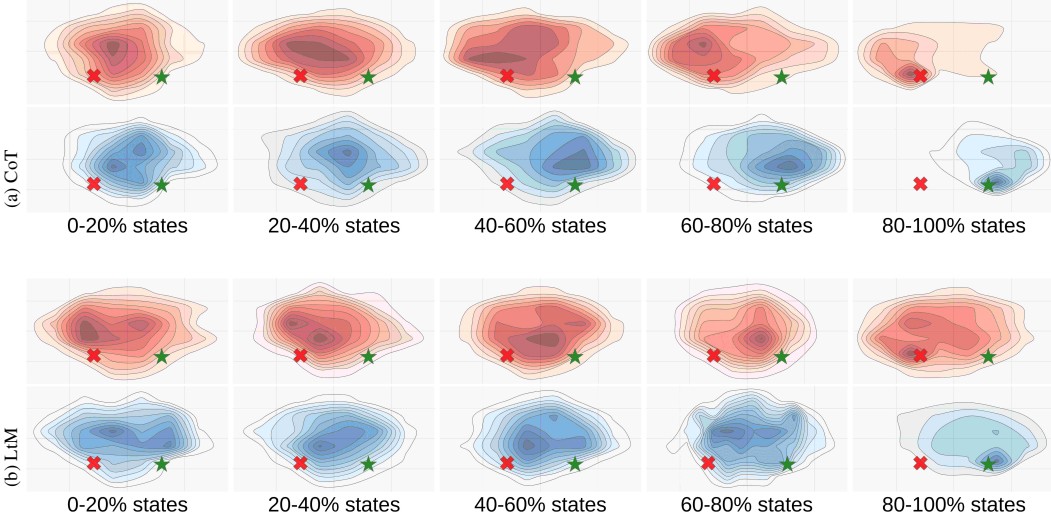

Figure 23: Comparing the LoT of different reasoning methods (using Llama-3.1-8B on STRONGReject).

observation drawn from the general-purpose model, but exhibits more complex reasoning patterns, such as self-evaluation and back-tracking.

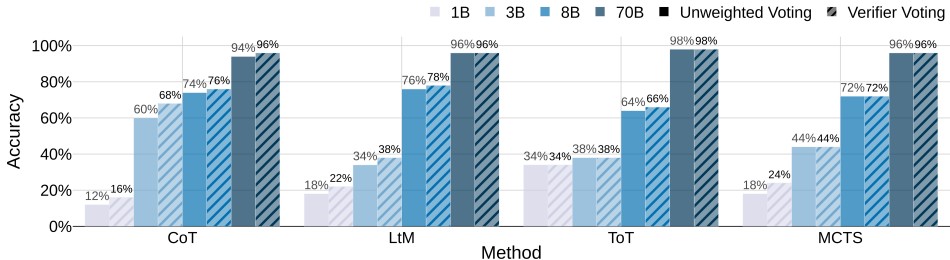

Figure 24: Average voting accuracy (%) of reasoning with and without verification on AQuA.

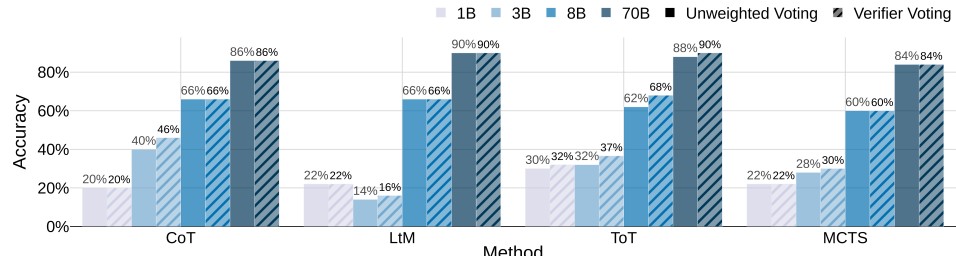

Figure 25: Average voting accuracy (%) of reasoning with and without verification on MMLU.

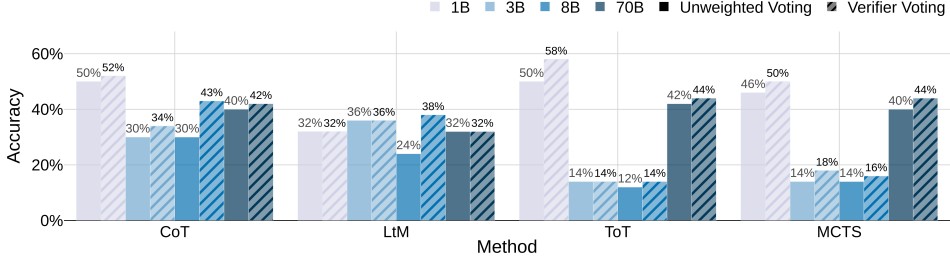

Figure 26: Average voting accuracy (%) of reasoning with and without verification on StrategyQA.

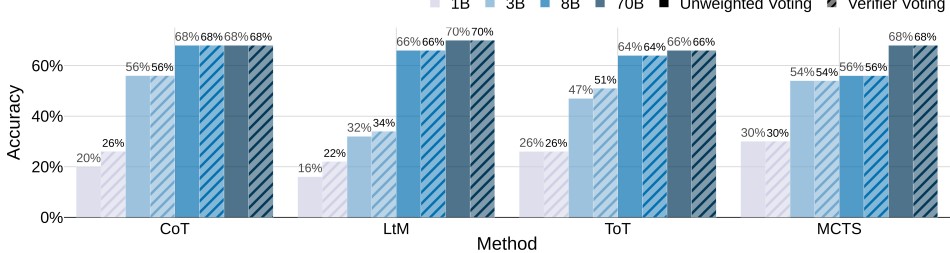

Figure 27: Average voting accuracy (%) of reasoning with and without verification on Common-SenseQA.

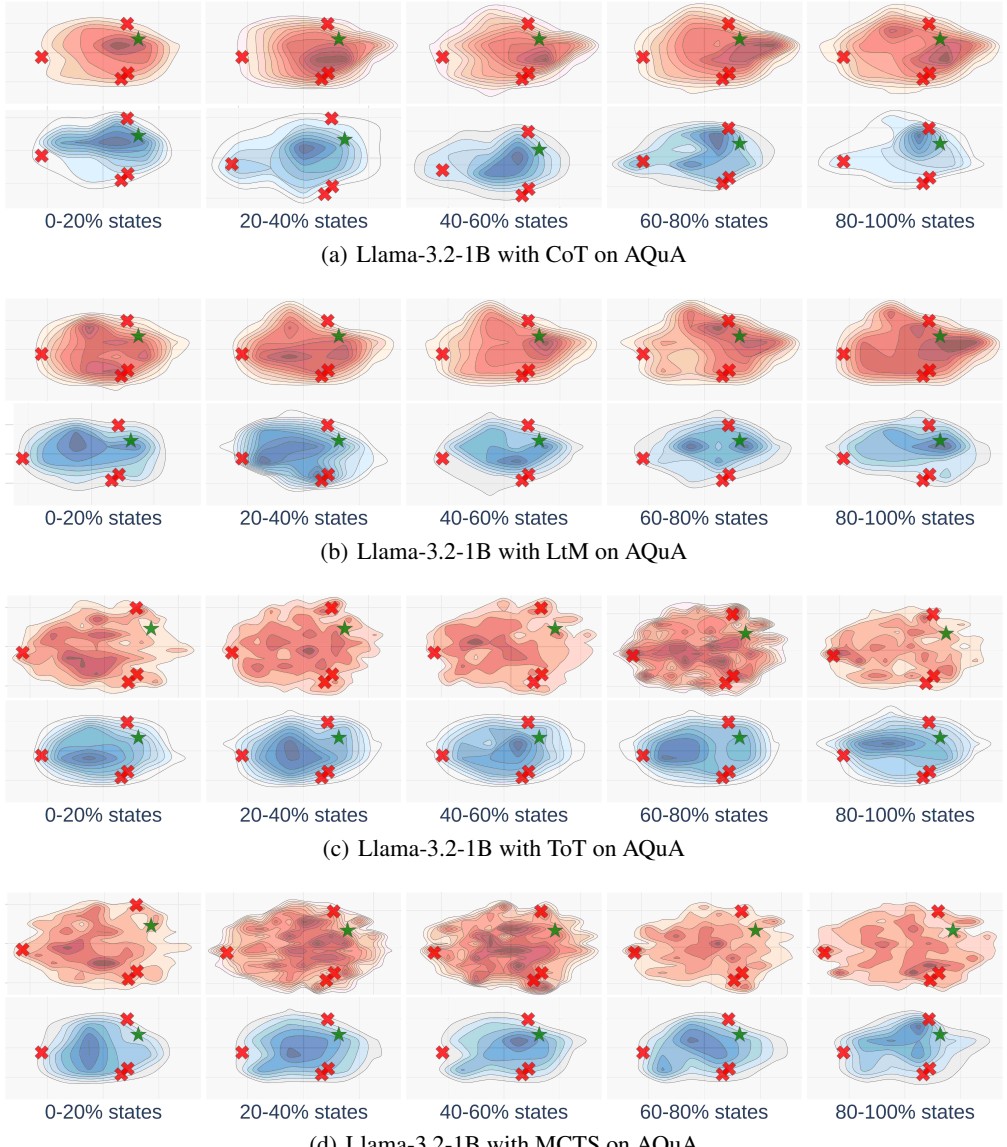

Figure 28: The landscapes of various reasoning methods (using Llama-3.2-1B on the AQuA dataset).

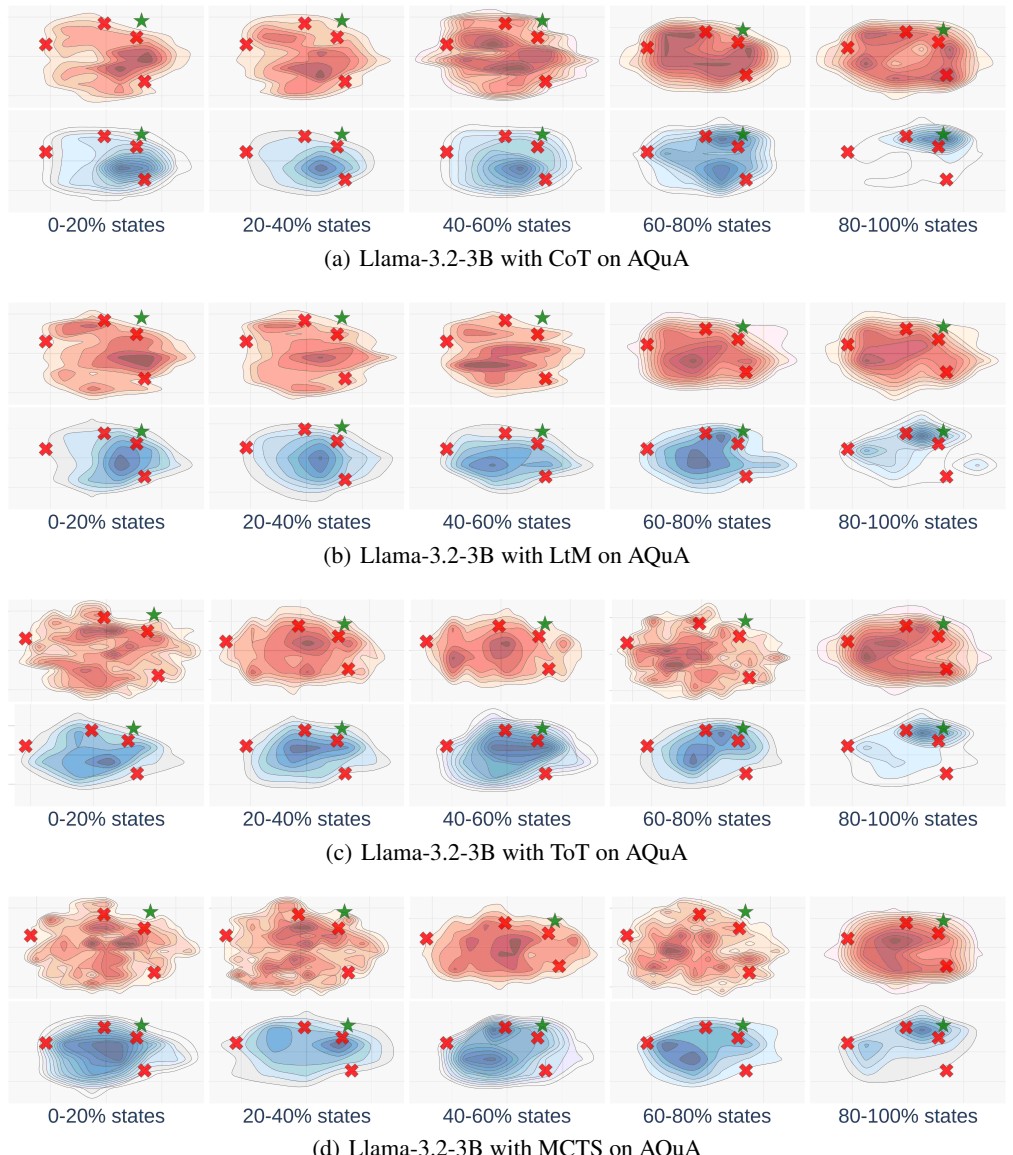

(a) Llama-3.2-3B with CoT on AQuA

(b) Llama-3.2-3B with LtM on AQuA

(c) Llama-3.2-3B with ToT on AQuA

(d) Llama-3.2-3B with MCTS on AQuA

Figure 29: The landscapes of various reasoning methods (using Llama-3.2-3B on the AQuA dataset).

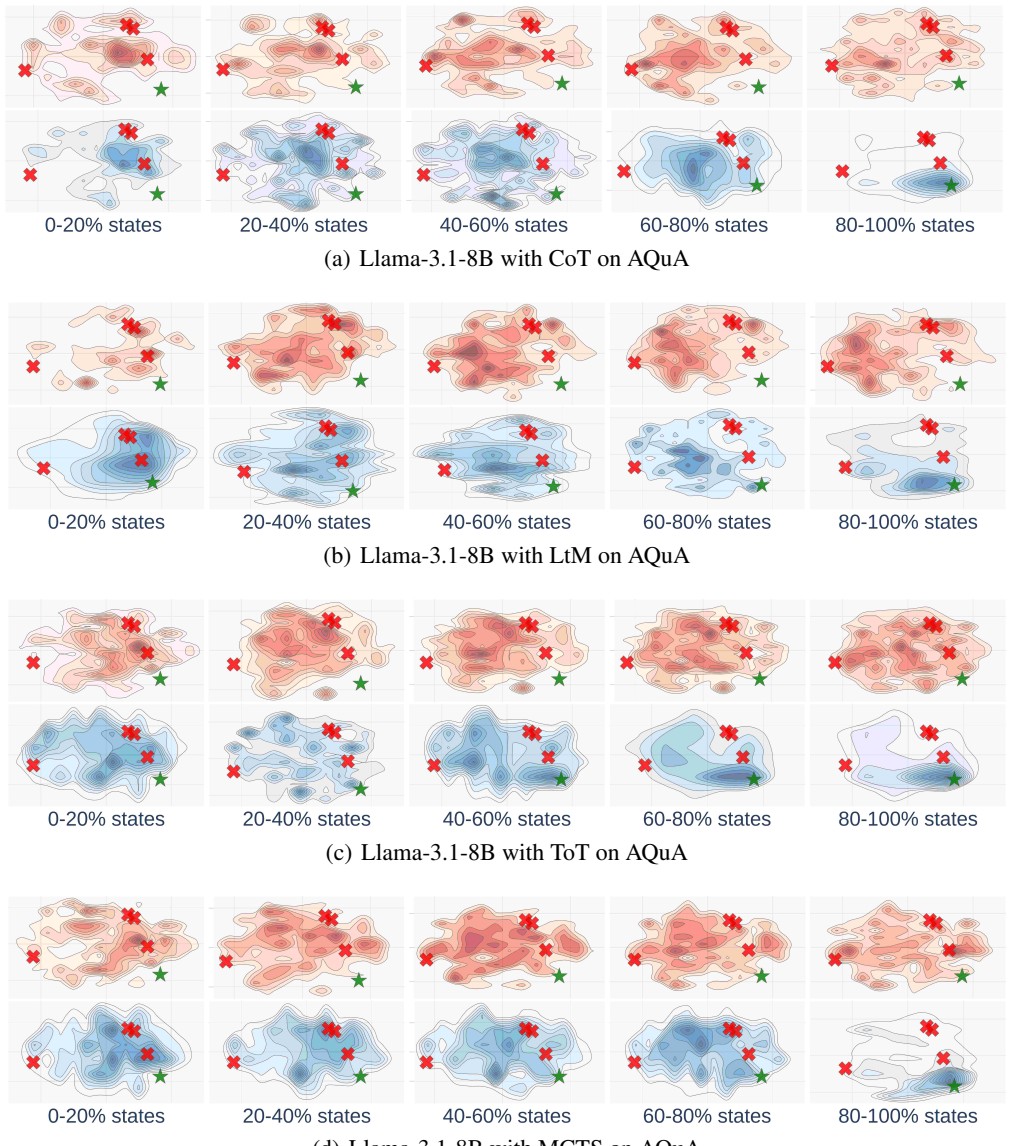

(a) Llama-3.1-8B with CoT on AQuA

(b) Llama-3.1-8B with LtM on AQuA

(c) Llama-3.1-8B with ToT on AQuA

(d) Llama-3.1-8B with MCTS on AQuA

Figure 30: The landscapes of various reasoning methods (using Llama-3.1-8B on the AQuA dataset).

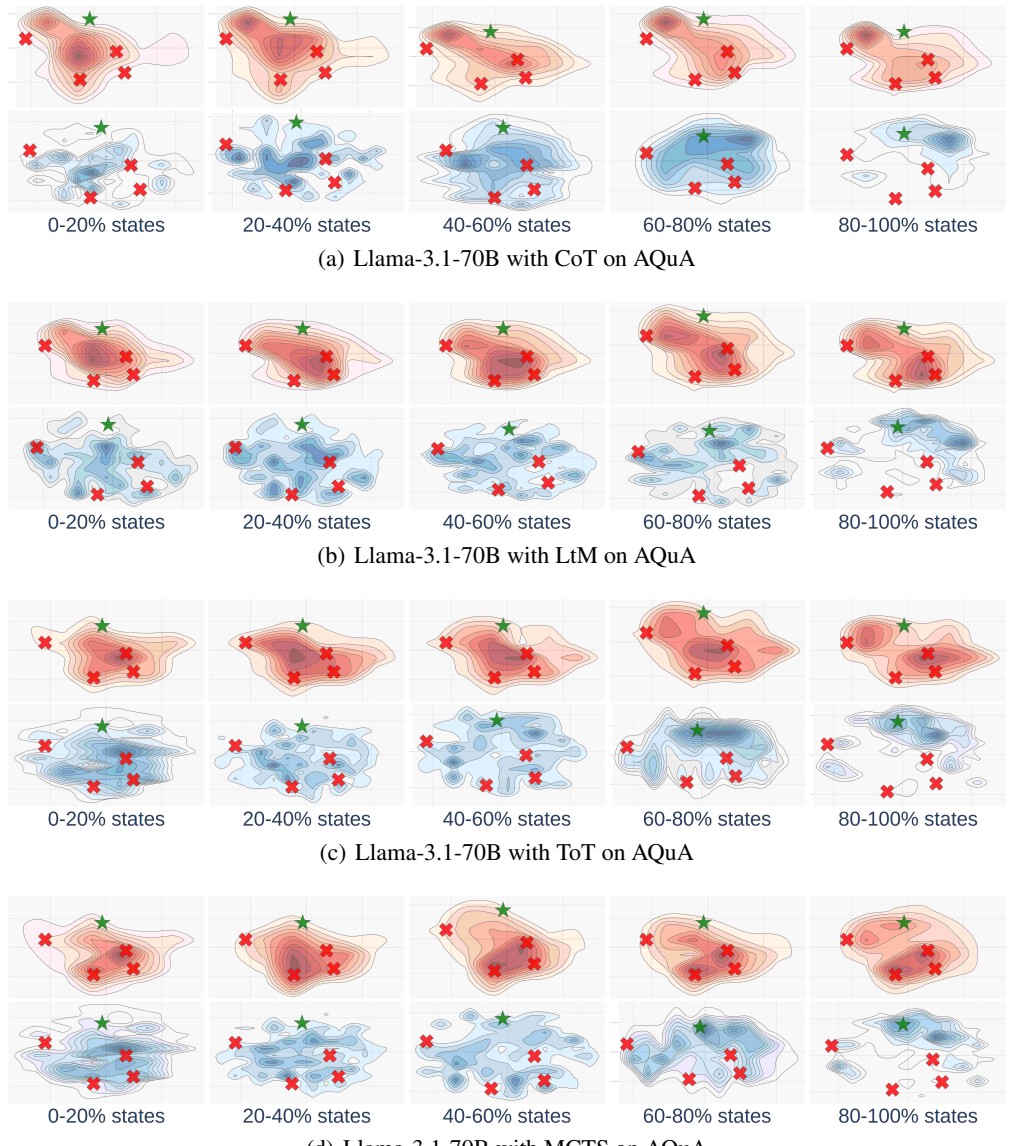

Figure 31: The landscapes of various reasoning methods (using Llama-3.1-70B on the AQuA dataset).

To solve the problem, let's break it down into a series of calculations according to the given property.

2. The perimeter of the other part is 66 cm (perimeter of 16x and 14y).

Hose A fills the pool in 8 hours, so its rate is 1/8 of the pool per hour.

Step 4: Substitute the calculated value for 4/5 of 25 into the expression for the difference.

Conclusion: The original price of the item was approximately $63.32. The answer is A.

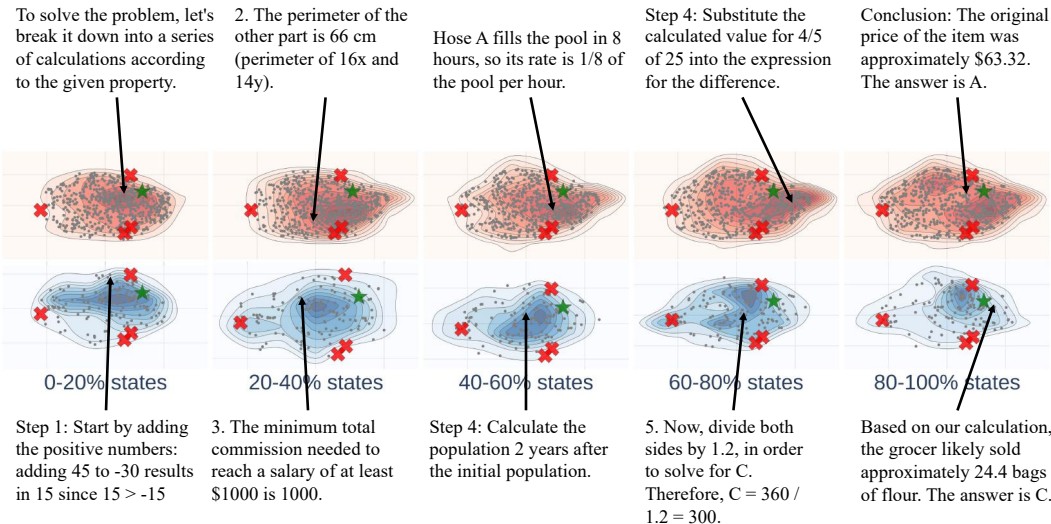

Step 1: Start by adding the positive numbers: adding 45 to -30 results in 15 since 15 > -15

3. The minimum total commission needed to reach a salary of at least $1000 is 1000.

Step 4: Calculate the population 2 years after the initial population.

5. Now, divide both sides by 1.2, in order to solve for C. Therefore, C = 360 / 1.2 = 300.

Based on our calculation, the grocer likely sold approximately 24.4 bags of flour. The answer is C.

Figure 32: Case Study: Landscape of thoughts of Llama-3.2-1B on AQuA using CoT.

Let's break down the steps to calculate the average speed of the plane around the square field.

4. Since the profit is 25% of the selling price, we can calculate the total selling price for all the bags sold.

6. Since we found that A = 1/8, we can solve for B by substituting A into the equation: 1/8 + B = 3/4.

However, the number of toys cannot be a fraction, so we need to round to the nearest whole number.

Step 8: Since Hose B can fill 1/8 of the pool per hour, it can fill the entire pool in 8 hours. Therefore, the answer is D.

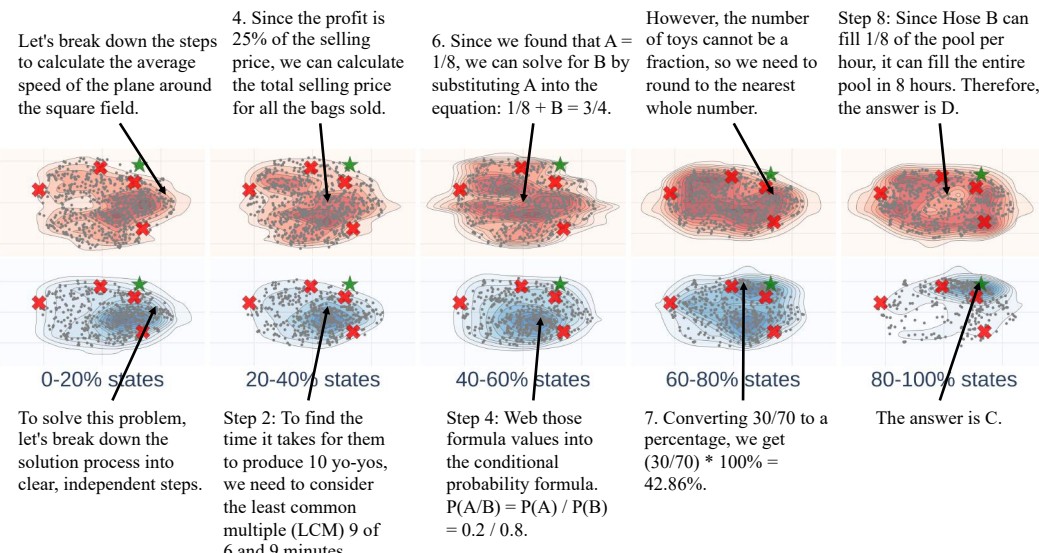

To solve this problem, let's break down the solution process into clear, independent steps.

Step 2: To find the time it takes for them to produce 10 yo-yos, we need to consider the least common multiple (LCM) 9 of 6 and 9 minutes.

Step 4: Web those formula values into the conditional probability formula. P(A/B) = P(A) / P(B) = 0.2 / 0.8.

7. Converting 30/70 to a percentage, we get (30/70) * 100% = 42.86%.

The answer is C.

Figure 33: Case Study: Landscape of thoughts of Llama-3.2-3B on AQuA using CoT.

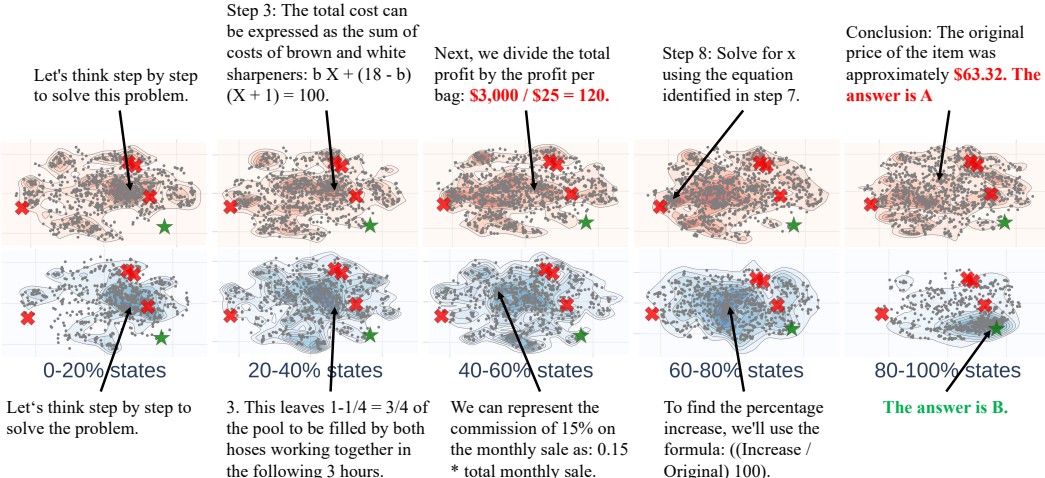

Figure 34: Case Study: Landscape of thoughts of Llama-3.1-8B on AQuA using CoT.

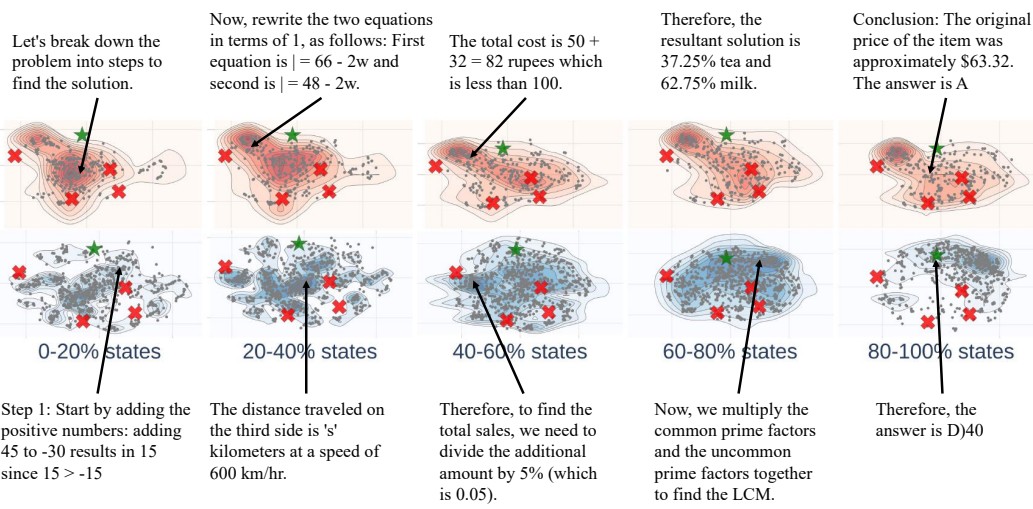

Figure 35: Case Study: Landscape of thoughts of Llama-3.1-70B on AQuA using CoT.

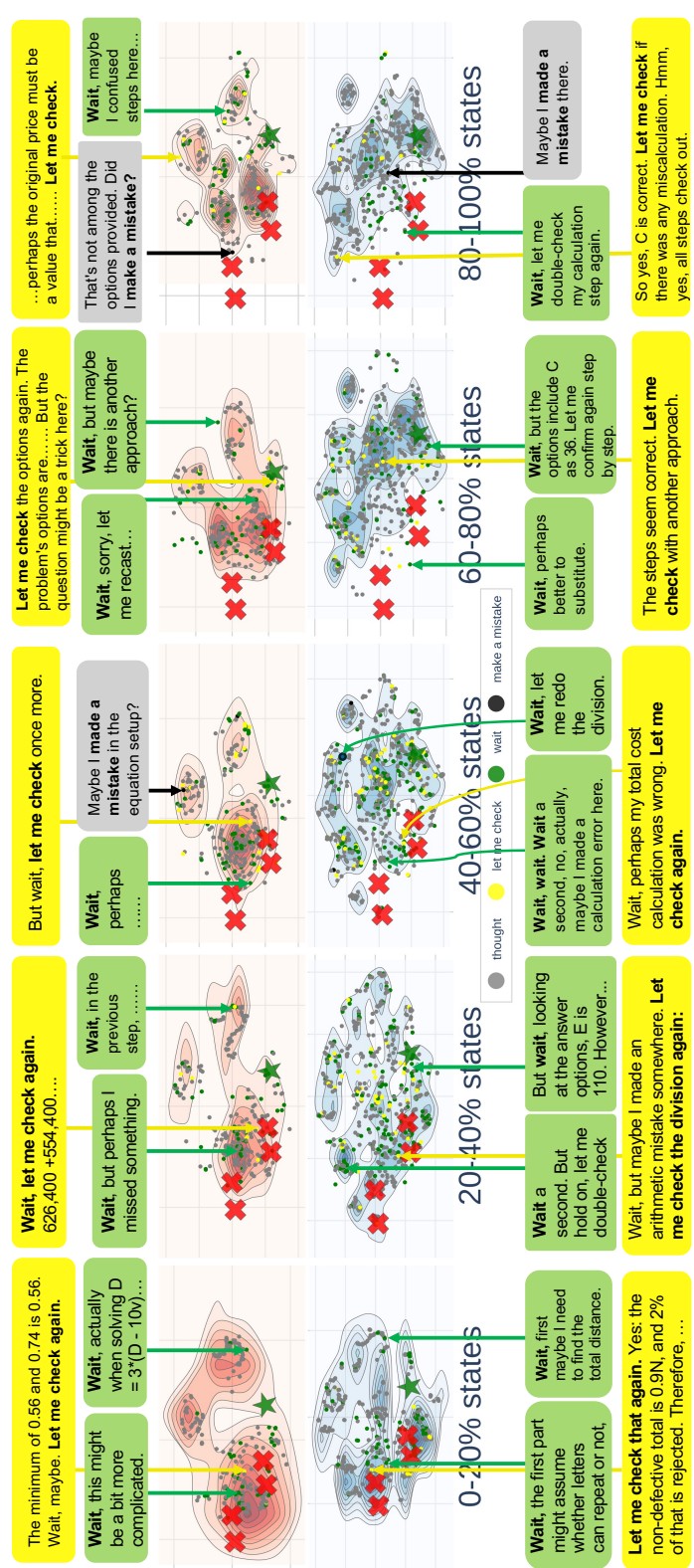

Figure 36: Landscape of QwQ-32B using CoT on AQuA.

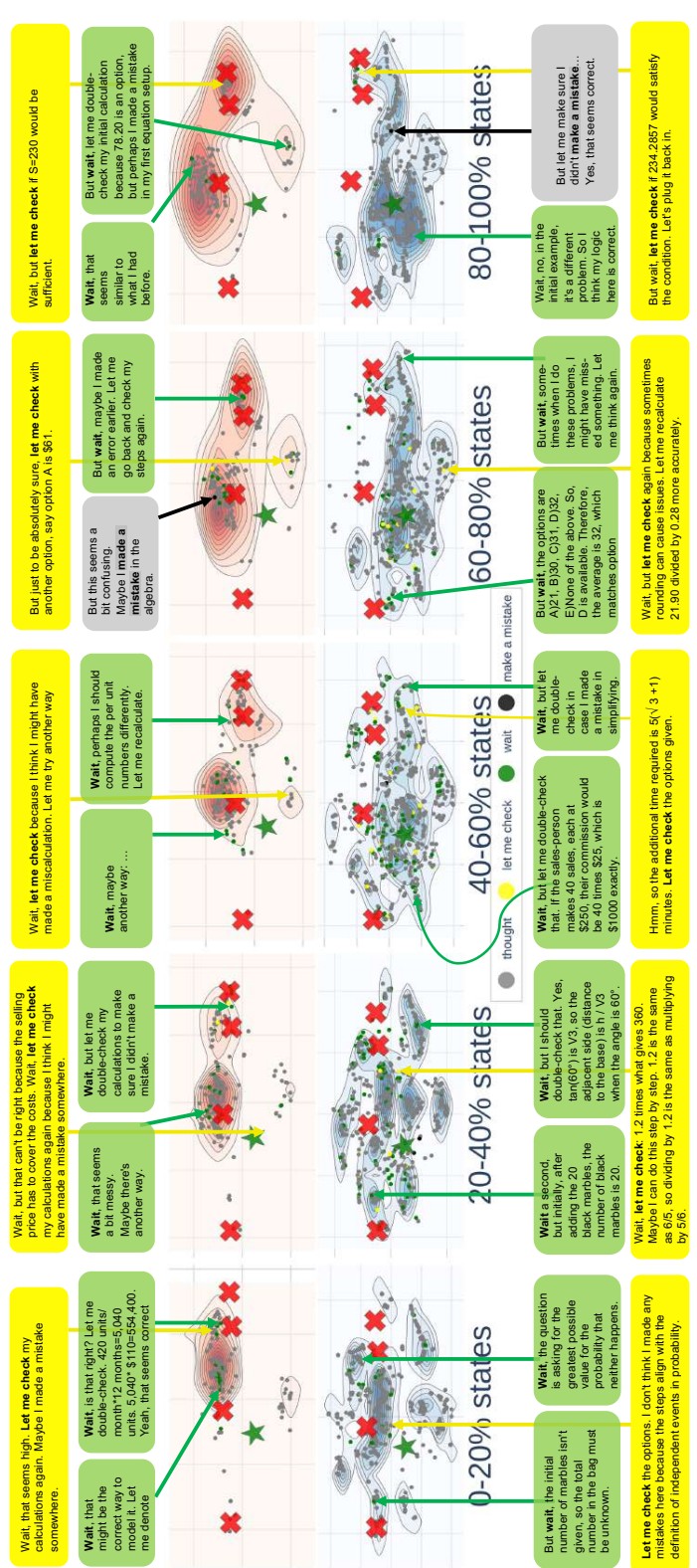

Figure 37: Landscape of DeepSeek-R1-Distill-Llama-70B using CoT on AQuA.

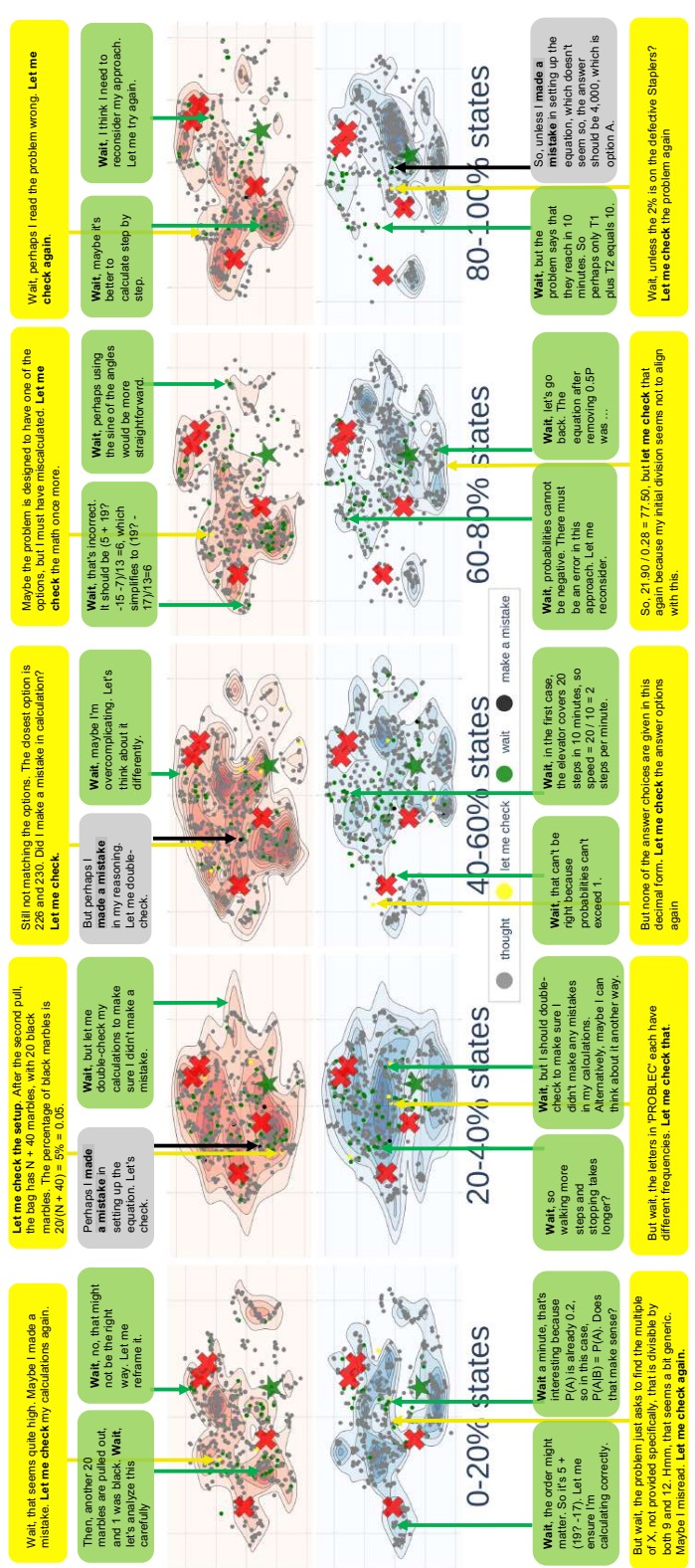

Figure 38: Landscape of DeepSeek-R1-Distill-Qwen-1.5B using CoT on AQuA.

