# OpenReview forum: "Landscape of Thoughts: Visualizing the Reasoning Process of Large Language Models"
_ICLR.cc/2026/Conference — ICLR 2026 Poster_

### Official Review · Reviewer_VKLY · 2025-10-20

**Soundness:** 2
**Presentation:** 2
**Contribution:** 2
**Rating:** 2
**Confidence:** 2

**Summary:**

This work proposes a novel approach for monitoring long chains of thought by visualizing reasoning trajectories in a two-dimensional feature embedding space. The proposed monitoring method, however, is restricted to multiple-choice problems: by appending each answer option to the intermediate reasoning states and computing the corresponding perplexity scores, the model estimates the “distance” between its current reasoning state and each candidate answer.

**Strengths:**

1. The paper addresses a meaningful and challenging research problem—how to effectively monitor and audit reasoning traces (i.e., chains of thought), particularly when reasoning models generate extremely long CoTs that are infeasible for human evaluation to scale.

2. The work demonstrates originality and novelty. Although the key components of the proposed method (e.g., perplexity and t-SNE) are not new, their combination and the new analytical perspective are refreshing and interesting.

**Weaknesses:**

W1: The scope of this paper is narrowly limited to multiple-choice questions with known ground-truth answers. It remains unclear how the proposed visualization tool can generalize to broader reasoning analysis or practical model auditing. Chain-of-thought monitoring and scalable oversight are indeed underexplored areas, people may hope to solve problems like detecting and controlling hallucination or unsafe content generation—tasks where ground-truth answers or verifiable signals are typically unavailable. However, this paper focuses instead on visualizing the distance between reasoning traces leading to correct versus incorrect answers. The conclusions (for example, wrong reasoning paths converge to wrong answers faster than correct ones) depend heavily on the existence of ground-truth labels. Such relative convergence patterns would not hold in open-ended reasoning tasks without definitive answers, substantially limiting the applicability and significance of this visualization approach for real-world reasoning analysis.

W2: It is not clear how **robust or faithful** the proposed visualization method and the conclusions derived from it actually are. The final reasoning landscape is constructed from state features computed via the perplexity of appending choice answers to intermediate reasoning steps. However, many unrelated factors could influence this measurement. For instance, even when the model follows a correct reasoning path, forcibly inserting the final answer into its ongoing reasoning can disrupt coherence and yield artificially high perplexity scores. Moreover, the phrasing of the final answer itself may significantly affect the computed perplexity. In other words, the distance metric may be driven by superficial textual factors rather than genuine semantic relations. While the resulting visualizations and observations are intriguing, the paper does not convincingly demonstrate the reliability or faithfulness of these conclusions.

* Typo*:
(1) Line 381, converge means... -> convergence means ...

**Questions:**

Q1. How does the number of sampled reasoning traces affect the resulting landscape? If the number of rollouts increases, will the overall structure of the landscape remain stable, or does it change significantly? A sensitivity analysis could help clarify the robustness of the visualization.

Q2. In the visualizations, blue and red trajectories represent reasoning traces leading to correct and incorrect answers, respectively. For challenging questions where most rollouts fail, what meaningful insights can be extracted from those predominantly incorrect trajectories?

Q3. The trends of uncertainty and perplexity are confusing. For example in Fig.2, all models exhibit a non-decreasing uncertainty over time, suggesting that they become less committed to a single final answer as reasoning progresses. Similarly, the perplexity for their own reasoning traces increases continuously. What does this imply? A clearer explanation would help readers interpret these metrics more meaningfully.

---

> ### Author Response · Authors · 2025-11-26
> **Response to Reviewer VKLY (part 1/6)**
>
> **We thank Reviewer VKLY for the thoughtful and detailed feedback.** We carefully address **all of your concerns**:
> - First, your questions about the generalization of our method are addressed by extending LoT to open-ended tasks and analyzing the resulting behaviors.
> - Second, your concerns about robustness and sensitivity are addressed through a set of robustness tests.
> - Finally, we discuss the landscapes of challenging questions and the observed trends in uncertainty and perplexity.
>
> Point-by-point responses are provided below. In addition, we have outlined a detailed revision plan at the end of each response. We will improve our submission within the next week according to this plan.
>
> > ### W1. Generalization to open-ended tasks that go beyond multiple-choice questions.
>
> **Reply:** Thank you for this insightful comment. We agree that the generalization to fully open-ended questions is a crucial point in LoT.  In response, we (1) clarify that LoT can effectively construct the landscape of open-ended questions when constructing high-quality pseudo multiple choices, (2) provide new empirical results on open-ended benchmarks **MATH, GSM8k, and StrongReject** to show the visualization effectiveness of LoT in **open-ended questions**, where the observations in these datasets highly align with our observations in multiple-choice tasks, and (3) discuss a fully automatic anchor-discovery method that requires no manual options annotation or auxiliary distractor generation.
>
> **(1) The multiple-choice restriction is pragmatic, not fundamental, and a simple pseudo-option strategy can address this restriction.** LoT construction requires anchors to compute relative distances, as defined in Equation (1), where the feature $f_i$ for state $s_i$ quantifies distances to choices $\{c_j\}_{j=1}^k$ via perplexity in Equation (2). In open-ended settings, these do not exist naturally, but they can be created reliably. We tested two complementary families:
> - for rule-verifiable reasoning (**MATH** and **GSM8K**), we keep the unique correct answer and use Llama-3.1-8B-Instruct to generate three plausible incorrect final answers plus full reasoning chains, followed by manual filtering to retain only non-trivial distractors, yielding clean 4-way multiple-choice versions;
> - for non-rule-verifiable or safety tasks (**StrongReject** jailbreak benchmark), we use True/False as the two options, with ground-truth labels produced by GPT-4o and manually verified.
>
> This approach mirrors the original setup in Section 2.2, ensuring the perplexity-based distances remain meaningful without altering the core visualization pipeline.
>
> **(2) Constructed landscapes of reasoning processes highly align with our observations on multiple-choice datasets.** We sampled 10 trajectories per question using the Llama-3.1-8B-Instruct model exactly as in the multiple-choice experiments and constructed the LoT identically. New Figures 21 (MATH), 22 (GSM8K), and 23 (StrongReject) show the phenomena replicate strongly. Key observations hold clearly in open-ended settings. We describe them as follows:
> - Observation 3.4 (Similar reasoning tasks exhibit similar landscapes). MATH and GSM8K exhibit rich state diversity and phased exploration patterns highly similar to multiple-choice datasets (AQuA, MMLU, and StrategyQA in Figure 4), whereas StrongReject shows concentrated, low-diversity search regions akin to CommonSenseQA.
> - Observation 3.7 (Within-method comparison: For any single method, incorrect trajectories converge faster to wrong answers than correct trajectories converge to right answers.) Across all three open-ended benchmarks, incorrect trajectories rapidly approach wrong anchors in 60-80% of reasoning states, while correct trajectories explore longer and only converge in the final 80-100% of states, mirroring the pattern in Figures 4 and 5 of multiple-choice questions.
>
> The core insights, therefore, transfer to open-ended questions without choices. These observations support that the LoT can be employed to analyze the reasoning process of open-ended questions.
>
> **(3) We discuss a potential fully automatic strategy to adapt LoT in open-ended questions without pseudo-labeling.** To eliminate the concern that our current results might depend on the quality of pseudo-options generated by another model or on manual filtering, we propose a fully automatic anchor discovery approach as future work. Specifically, after sampling multiple trajectories for an open-ended question, the final answers can be embedded with a strong embedding model and clustered in semantic space. Task-specific verifiers, e.g., exact-match checks for math problems or safety classifiers for jailbreak evaluation, can automatically label the highest-quality cluster as the correct anchor and the remaining clusters as incorrect anchors. This procedure requires no auxiliary model and no human intervention, making it more scalable while preserving the perplexity computations in Eq. (2).

---

> ### Author Response · Authors · 2025-11-26
> **Response to Reviewer VKLY (part 2/6)**
>
> In summary, these points demonstrate that LoT generalizes effectively to open-ended tasks through pseudo-options or automatic anchors, with empirical evidence confirming the transfer of key observations.
>
> In the revised version, we will add a subsection on generalization to open-ended reasoning that presents the current pseudo-option experiments, includes the corresponding figures, and discusses the fully automatic clustering-based anchor discovery method as robust future work. We will also clarify throughout the paper that the key observations are not restricted to questions with predefined choices.
>
> > ### W2. The robustness and faithfulness of the visualization method are unclear.
>
> **Reply:** Thank you for this thoughtful comment. In brief, our visualization is strictly **post-hoc (it never perturbs the model’s own reasoning)**, our distance metric is designed to capture **semantic compatibility between partial reasoning and full answers**, and we run several **robustness tests** (choice reordering, tokenization variants, and cross-model perplexity checks) that show the landscapes and metrics are stable. These qualitative patterns are further supported by quantitative analyses and are consistent across multiple model scales, datasets, and reasoning methods.
>
> **(1) The visualization method never intervenes in or alters the model’s reasoning trajectory.** The landscape is constructed only after the entire chain of thought ${t_1, t_2, \dots, t_n}$ has been fully generated. For each intermediate state $s_i = [x, t_1, \dots, t_i]$, we query $p_{\mathrm{LLM}}(c_j \mid s_i)$ after generation has finished, without modifying any token in the trajectory or inserting any answer text into the ongoing reasoning process. As described in Section 2.2 and illustrated in Figure 1, the model’s generation proceeds exactly as it would at test time, and LoT only reads out likelihoods afterward. This eliminates the specific failure mode the reviewer mentions, where “forcibly inserting the final answer into ongoing reasoning” could disrupt coherence or artificially inflate perplexity.
>
> **(2) The conclusions are robust to superficial textual and formatting perturbations.** To test whether the landscapes are driven by surface factors such as option labels or tokenization, we run several robustness experiments.
> - Choice reordering (new experiment, suggested by the reviewer). We swap the ordering of answer options (for example, $\text{A)}\ c_1,\ \text{B)}\ c_2 \rightarrow \text{B)}\ c_1,\ \text{A)}\ c_2$) while keeping the semantic content of each $c_j$ unchanged. We then recompute state features, rebuild the landscapes, and replot the consistency, uncertainty, and perplexity plots. As shown in Figures 15 and 16, convergence patterns, separation between correct and incorrect trajectories, and the overall geometry are effectively unchanged; the metric plots almost overlap. This indicates that LoT is insensitive to superficial label permutations.
> - Sentence tokenization robustness (Appendix H.9). We also vary how thoughts are segmented into states, for example, by splitting multi-sentence thoughts into shorter segments or merging shorter sentences into longer ones, which changes the exact $s_i$ prefixes and token boundaries. Recomputing features and landscapes under these alternative tokenizations yields very similar consistency and uncertainty plots and preserves the key convergence patterns. The verifier performance in Section 4 is also stable under these tokenization variants. This suggests that LoT does not rely on a fragile, particular sentence segmentation.
> - Cross-model perplexity stability (Appendix H.5). When we hold the chain of thought fixed and evaluate perplexity under different Llama 3 models, the average perplexity for the same text lies in a narrow range (approximately $1.4$ to $2.0$). In contrast, the dynamics we study (for example, how distances and consistency evolve as the model generates its own thoughts) differ substantially across models and methods. This supports the view that our conclusions are not driven by arbitrary global calibration differences or small phrasing variations.
>
> Taken together, these perturbation studies show that the observed structures are robust to superficial textual changes and are not artifacts of a particular choice of encoding or tokenization scheme.

---

> ### Author Response · Authors · 2025-11-26
> **Response to Reviewer VKLY (part 3/6)**
>
> **(3) The perplexity-based distance reflects semantic compatibility, and its structure is consistent across models and datasets.** Our distance metric $d(s_i, c_j) = p_{\mathrm{LLM}}(c_j \mid s_i)^{-1 / |c_j|}$ is length normalized and autoregressive, measuring how compatible the entire candidate $c_j$ is with the partial reasoning state $s_i$. After $\ell_1$ normalization, the feature vector $f_i = [d(s_i, c_1), \dots, d(s_i, c_K)]^\top$ behaves like a distribution over answer hypotheses and encodes how belief mass is allocated across options. If this distance were dominated by superficial textual factors, we would not expect the consistent, label-aligned structures seen in Figures 2, 4, and 5:
> - **Within a fixed model and dataset,** trajectories that eventually answer correctly move into regions where $f_i$ strongly favors the correct option, while incorrect trajectories converge early to regions dominated by distractors.
> - **Across model scales,** larger models exhibit smoother convergence to the correct region and higher end-stage consistency, even though the answer strings are often very short (for example, options A-D in MMLU), where purely lexical differences are minimal.
> - **Across datasets (AQuA, MMLU, StrategyQA, CommonsenseQA),** LoT recovers stable but distinct landscape shapes and uncertainty patterns that reflect task type rather than formatting, as quantified by distributional similarity scores in Table 1(c).
> The fact that the same distance-based features support these patterns across models, datasets, and reasoning methods makes it unlikely that they are driven solely by shallow syntax or idiosyncratic phrasing of the final answers.
>
> **(4) The qualitative observations are backed by quantitative analyses of convergence, speed, and landscape similarity.** Beyond visual inspection, Appendix H.1 provides statistical tests that directly evaluate the behaviors inferred from the landscapes:
> - Convergence of correct vs incorrect trajectories. For each question $q$, we define distances $d_i(q) = \lVert f_i^{(q)} - f_n^{(q)} \rVert_2$ and fit $\log d_i(q) \approx \alpha_q + \beta_q i$, using $e^{\beta_q}$ as a convergence coefficient. Table 1(a) shows that this coefficient differs significantly between correct and incorrect trajectories, with error paths converging earlier to their final answer region ($p = 0.008$).
> - Path speed and accuracy. For Observations 3.1 and 3.7, we define a trajectory speed $\text{speed} = \frac{\lVert \bar s_n - \bar s_0 \rVert_2}{\sum_{j=1}^{n} \lVert \bar s_j - \bar s_{j-1} \rVert_2} \in [0, 1]$, where $\bar s_i$ is the 2D coordinate of state $i$, and correlate this with accuracy. Table 1(b) reports a very strong relationship between speed and correctness ($p \approx 9.4 \times 10^{-11}$).
> - Cross-task landscape similarity. For Observation 3.4, we compute pairwise histogram intersection scores between landscape density distributions. Table 1(c) shows that AQuA, MMLU, and StrategyQA form a tight cluster, while CommonsenseQA is statistically distinct, matching the visual intuition from the landscapes.
>
> These tests operate directly on the LoT features and their trajectories, and they confirm that the visual patterns correspond to robust differences in model behavior rather than artifacts of the distance metric.
>
> In summary, our visualization method is strictly post-hoc, uses a length-normalized perplexity-based distance that captures semantic compatibility with answer hypotheses, and exhibits strong robustness to choice reordering, sentence tokenization changes, and cross-model perplexity calibration. The same structures appear consistently across multiple model scales, datasets, and reasoning methods and are quantitatively supported by convergence, speed, and similarity analyses, as well as a downstream verifier.
>
> In the revisions, we will (i) more clearly highlight the post-hoc nature of the likelihood queries, (ii) emphasize why the perplexity-based metric is an appropriate semantic proxy, and (iii) move the robustness experiments (choice reordering and sentence tokenization) into a more prominent place, explicitly tying them to the faithfulness of the visualizations and conclusions.
>
> > ### W3. Typo*: (1) Line 381, converge means... -> convergence means…
>
> **Reply:** Thank you for pointing out this typo. We have revised it in our updated submission.

---

> ### Author Response · Authors · 2025-11-26
> **Response to Reviewer VKLY (part 4/6)**
>
> > ### Q1. Sensitivity of the landscape to the number of sampled reasoning traces.
>
> **Reply:** Thank you for this thoughtful question. To answer this question, we conduct additional experiments and show that **the landscape is structurally stable across different numbers of sampled reasoning traces:** increasing the number of rollouts mainly smooths the density visualization, but it does not materially change the geometry, convergence patterns, or any of the conclusions we draw from LoT.
>
> **(1) LoT aggregates traces in a way that is inherently stable to sample size.** LoT operates in a fixed feature space where each state is represented by a distance vector $f_i$ over answer options (Eq. (1)), and all state features are stacked into a matrix $S$ before dimensionality reduction (Eq. (4)). Adding more trajectories simply adds more rows to $S$ from the same underlying distribution of states. This increases the sampling density of the manifold but does not redefine the feature space or the embedding function, so the global geometry of the landscape is expected to remain stable as the sample size grows.
>
> **(2) A sensitivity analysis confirms that landscape geometry and patterns are robust to the number of rollouts.** We explicitly test this by generating landscapes with different numbers of sampled trajectories per question: 10, 20, 30, 40, and 50, which were used in the original submission. Across all these configurations (reported in Figure 18), the resulting landscapes exhibit the same qualitative structure:
> - correct trajectories gradually converge toward the correct region;
> - incorrect trajectories collapse earlier toward wrong answer anchors;
> - the relative placement of major clusters is unchanged;
> - the evolution of density over time is smooth and monotonic in sample size.
>
> The only noticeable difference is visual: landscapes with more samples (for example, 40 or 50 rollouts) appear smoother and less noisy because the underlying density is better estimated, but the shapes of clusters, the separation between correct and incorrect trajectories, and the convergence patterns remain the same.
>
> In the revised manuscript, we will (i) include the sensitivity analysis figure comparing landscapes at different sample counts and (ii) explicitly state in Section 3 that the main LoT observations are stable across these sampling settings.
>
> > ### Q2. What can be observed in the incorrect trajectories of challenging questions?
>
> **Reply:** Thank you for this valuable question. **We visualize the landscapes on challenging questions and observe that LoT continues to reveal stable and interpretable reasoning patterns, and both incorrect and remaining correct trajectories provide meaningful information.** Below, we summarize the observations from our additional experiments on challenging-question subsets of the AQUA dataset (see Figure 19).
>
> **(1) The main LoT structure remains stable across all difficulty thresholds.** We examined LoT on progressively harder subsets by filtering out easy questions using accuracy thresholds of $<50\%, <40\%$, and $<30\%$ (Figure 19). Across all cases, the characteristic patterns documented in Section 3 remain visible: early-stage states are more dispersed, mid-stage states exhibit partial organization, and later-stage states move more clearly toward specific answer options. This confirms that the core LoT behaviors persist even when accuracy is low.
>
> **(2) Incorrect trajectories continue to show clear option-directed movement.** The incorrect trajectories (red) still exhibit recognizable movement toward specific answer choices as the reasoning progresses. This behavior is visible across all state ranges and does not disappear even when correct trajectories become rare. Thus, LoT continues to capture meaningful convergence tendencies for incorrect reasoning.
>
> **(3) Late-stage concentration becomes weaker as accuracy decreases.** In the original landscape, the later-state incorrect trajectories form more compact regions. However, in the challenging subsets, 60-80 percent and 80-100 percent of states show broader and less sharply grouped red regions. This indicates that the model becomes less decisive during late reasoning stages on more challenging questions, as also discussed in Observation 3.6 in the submission. The filtered landscapes make this phenomenon more pronounced.
>
> **(4) The remaining correct trajectories still follow the expected patterns.** Although correct trajectories become sparse in harder subsets, the ones that remain continue to align closely with the correct option across most state ranges. Their intermediate states show higher stability, and their later states form clear clusters near the correct anchor. These behaviors are consistent with Observation 3.1 and demonstrate that LoT continues to reflect successful reasoning even when questions are challenging.

---

> ### Author Response · Authors · 2025-11-26
> **Response to Reviewer VKLY (part 5/6)**
>
> **(5) The contrast between correct and incorrect trajectories becomes more pronounced on difficult questions.** As the questions become harder, incorrect states become increasingly dispersed, while the remaining correct states stay compact. This highlights a clearer distinction between stable reasoning and unstable reasoning in the challenging subsets, reinforcing the interpretive value of LoT even when accuracy is low.
>
> In summary, LoT remains informative and interpretable even when the dataset consists primarily of incorrect trajectories. The landscapes of challenging questions preserve the core patterns described in Section 3 while revealing additional difficulty-specific phenomena such as greater mid-stage dispersion and weaker late-stage grouping.
>
> To improve clarity, we will (i) include these challenging-question visualizations in the appendix, (ii) highlight the stable and informative patterns exhibited by incorrect trajectories, and (iii) clarify in the main text that LoT continues to provide meaningful insights in low-accuracy regimes.
>
> > ### Q3. How to explain the increasing uncertainty and perplexity?
>
> **Reply:** Thank you for raising this insightful question. **In short, uncertainty and perplexity in our framework capture two different aspects of the process (belief over answer options vs. predictability of reasoning text), and their mostly increasing trends in Figure 2 reflect exploratory and increasingly specific reasoning, rather than a monotone loss of commitment.**
>
> In the following, we clarify this interpretation, emphasize the small late-stage drop in uncertainty, and connect these plots to existing findings in the literature.
>
> **(1) Uncertainty measures the spread of belief over answer options, and its trend reflects exploration plus late-stage commitment.** For each state $s_i$, we form a normalized distance vector $f_i \in \mathbb{R}^k$ over the $k$ answer choices and define ${\rm Uncertainty}(s_i) = - \sum_{j=1}^k f_i(j) \log f_i(j)$ with $\sum_j f_i(j) = 1$. Low uncertainty means the model strongly prefers one option; high uncertainty means it spreads belief across several. In Figure 2, the average uncertainty over trajectories tends to increase as reasoning progresses, indicating that intermediate thoughts often open up or rebalance multiple candidates instead of simply sharpening an initial guess. Importantly, in the final stage (roughly the last 20% of thoughts), we observe a slight drop in uncertainty, consistent with the model only firmly committing to its final answer near the end of reasoning.
>
> **(2) Perplexity is defined over the reasoning text itself and naturally increases as thoughts become longer, more specific, and less templated.** For each thought $t_i$, we define ${\rm PPL}(t_i) = p_{\text{LLM}}(t_i \mid s_{i-1})^{-1/\lvert t_i \rvert}$, using the model’s own conditional probabilities. Early steps often use very generic templates (for example, “Let us think step by step”), which are high probability and hence low perplexity. Later thoughts contain detailed computations, question-specific entities, and idiosyncratic phrasing, which are rarer under the model and thus yield higher perplexity. This explains why perplexity tends to increase with step index even though larger models have systematically lower perplexity plots than smaller ones at the same step (as seen in Figure 2 and Appendix H.5).
>
> Similar phenomena have been noted in prior work: high-quality or human-like text does not always correspond to the lowest-perplexity generations, and useful content often resides in relatively lower-probability regions of the model’s distribution. For example, Holtzman et al. [1] observe that pushing perplexity too low leads to degenerate, repetitive text and argue that high-quality text typically has moderate perplexity rather than minimal perplexity. Recent work on chain-of-thought also uses stepwise perplexity to identify “critical” reasoning steps, showing that changes in perplexity along the chain correlate with decision quality [2]. Our increasing-perplexity plots are consistent with the view that deeper reasoning steps are rarer but important.

---

> ### Author Response · Authors · 2025-11-26
> **Response to Reviewer VKLY (part 6/6)**
>
> **(3) The increasing trends of uncertainty and perplexity are most informative when interpreted together with consistency and accuracy, especially across model scales.** In Figure 2, larger models exhibit higher consistency (their intermediate states agree with the final answer more often and earlier in the trajectory) while still showing increasing uncertainty and perplexity. This suggests a pattern of "early latent commitment plus exploratory justification": larger models tend to settle on a preferred answer relatively early (reflected in high consistency), then generate more complex, problem-specific reasoning that (i) briefly reconsiders alternatives, raising average uncertainty, and (ii) uses less frequent language, raising perplexity. Smaller models, in contrast, exhibit lower consistency together with similar or higher uncertainty and perplexity, indicating that their exploration is less controlled and more often accompanied by changes in the preferred answer. Our verifier in Section 4 leverages exactly these temporal patterns in uncertainty and perplexity (along with consistency) to predict correctness, which further supports that these metrics capture meaningful dynamics rather than noise.
>
> In summary, the non-decreasing uncertainty and increasing perplexity in Figure 2 do not imply that models simply become less confident as they reason. Rather, they show that models (i) explore and hedge among multiple answer options in the middle of trajectories, then slightly reduce uncertainty when committing at the end, and (ii) move from generic, high-probability templates to rarer, more specialized reasoning text as depth increases, with larger models doing so at overall lower perplexity levels.
>
> In the revisions, we will (1) restate next to Figure 2 that uncertainty is defined over answer options and perplexity over reasoning tokens, (2) highlight the late-stage uncertainty drop and add separate plots for correct and incorrect trajectories, and (3) add a short discussion connecting our perplexity trends to prior work on perplexity and reasoning, citing recent stepwise perplexity analyses in CoT reasoning.
>
> Reference:
> [1] The Curious Case of Neural Text Degeneration. In ICLR, 2020.
> [2] Stepwise Perplexity-Guided Refinement for Efficient Chain-of-Thought Reasoning in Large Language Models. In ACL, 2025.
>
> **We would like to thank Reviewer VKLY again** for taking the time to review our submission and consider our rebuttal. **If our responses have addressed your concerns, we would be very grateful if you could reconsider your rating.** We also welcome any further comments or questions you may have.

---

### Official Review · Reviewer_fVdk · 2025-11-01

**Soundness:** 3
**Presentation:** 4
**Contribution:** 3
**Rating:** 8
**Confidence:** 3

**Summary:**

This paper introduces a landscape of thoughts, the first visualization tool for users to inspect the reasoning paths of the chain-of-thought and its derivatives on any multi-choice dataset. The authors represent the states in a reasoning path as feature vectors that quantify their distances to all answer choices. These features are then visualized in two-dimensional plots using t-SNE. Qualitative and quantitative analysis, combined with the landscape of thoughts, effectively distinguishes between strong and weak models, correct and incorrect answers, as well as different reasoning tasks.

**Strengths:**

Strengths:
1.	Important contribution towards understanding the large reasoning models
2.	A model can be trained to predict any property users observe
3.	Good visualization plots

**Weaknesses:**

Weaknesses:
1. Only multiple-choice kind of questions are considered. I am not sure if the same insights transfer to the reasoning questions with no choices.
2. More models could be considered.

**Questions:**

Questions:

Q1: In Figure 1, it says that the wrong paths quickly converge to wrong answers, while the correct paths slowly converge to correct answers. However, observation 3.1 says that the landscape converges faster as the model size increases. But shouldn't the bigger model converge slowly, as it is expected to give more correct solutions? Any reason for this discrepancy?

Q2: Are the insights similar on reasoning questions without any multiple choices?

Q3: What about other reasoning methods, such as a graph of thoughts or an algorithm of thoughts?

---

> ### Author Response · Authors · 2025-11-26
> **Response to Reviewer fVdk (part 1/5)**
>
> **We thank Reviewer fVdk for the thoughtful and detailed feedback.** In this rebuttal, we systematically address **all** of your concerns.
> - First, we respond to your questions about generalization by extending LoT to open-ended tasks and analyzing the resulting behaviors.
> - Second, we present new empirical results on additional models (QwQ, Qwen) to further demonstrate the generality of LoT and its observed patterns.
> - Third, we clarify in detail the relationship between Figure 1 and Observation 3.1.
> - Finally, we report new experiments on two additional methods (AoT, GoT) and analyze their behaviors.
>
> Please find the point-to-point responses as follows. In addition, we have outlined a detailed revision plan at the end of each response. We will improve our submission within the next week according to this plan.
>
> > ### W1 & Q2. Generalization to open-ended tasks that go beyond multiple-choice questions.
>
> **Reply:** Thank you for this insightful comment. We agree that the generalization to fully open-ended questions is a crucial point in LoT.  In response, we (1) clarify that LoT can effectively construct the landscape of open-ended questions when constructing high-quality pseudo multiple choices, (2) provide new empirical results on open-ended benchmarks **MATH, GSM8k, and StrongReject** to show the visualization effectiveness of LoT in **open-ended questions**, where the observations in these datasets highly align with our observations in multiple-choice tasks, and (3) discuss a fully automatic anchor-discovery method that requires no manual options annotation or auxiliary distractor generation.
>
> **(1) The multiple-choice restriction is pragmatic, not fundamental, and a simple pseudo-option strategy can address this restriction.** LoT construction requires anchors to compute relative distances, as defined in Equation (1), where the feature $f_i$ for state $s_i$ quantifies distances to choices {$c_j$}$_{j=1}^k$ via perplexity in Equation (2). In open-ended settings, these do not exist naturally, but they can be created reliably. We tested two complementary families:
> - for rule-verifiable reasoning (**MATH** and **GSM8K**), we keep the unique correct answer and use Llama-3.1-8B-Instruct to generate three plausible incorrect final answers plus full reasoning chains, followed by manual filtering to retain only non-trivial distractors, yielding clean 4-way multiple-choice versions;
> - for non-rule-verifiable or safety tasks (**StrongReject** jailbreak benchmark), we use True/False as the two options, with ground-truth labels produced by GPT-4o and manually verified.
>
> This approach mirrors the original setup in Section 2.2, ensuring the perplexity-based distances remain meaningful without altering the core visualization pipeline.
>
> **(2) Constructed landscapes of reasoning processes highly align with our observations on multiple-choice datasets.** We sampled 10 trajectories per question using the Llama-3.1-8B-Instruct model exactly as in the multiple-choice experiments and constructed the LoT identically. New Figures 21 (MATH), 22 (GSM8K), and 23 (StrongReject) show the phenomena replicate strongly. Key observations hold clearly in open-ended settings. We describe them as follows:
> - Observation 3.4 (Similar reasoning tasks exhibit similar landscapes). MATH and GSM8K exhibit rich state diversity and phased exploration patterns highly similar to multiple-choice datasets (AQuA, MMLU, and StrategyQA in Figure 4), whereas StrongReject shows concentrated, low-diversity search regions akin to CommonSenseQA.
> - Observation 3.7 (Within-method comparison: For any single method, incorrect trajectories converge faster to wrong answers than correct trajectories converge to right answers.) Across all three open-ended benchmarks, incorrect trajectories rapidly approach wrong anchors in 60–80% of reasoning states, while correct trajectories explore longer and only converge in the final 80–100% of states, mirroring the pattern in Figures 4 and 5 of multiple-choice questions.
>
> The core insights, therefore, transfer directly to open-ended questions without any real choices. These observations support the idea that the LoT can be employed to analyze the reasoning process of open-ended questions.

---

> ### Author Response · Authors · 2025-11-26
> **Response to Reviewer fVdk (part 2/5)**
>
> **(3) We discuss a potential fully automatic strategy to adapt LoT in open-ended questions without pseudo-labeling.** To eliminate the concern that our current results might depend on the quality of pseudo-options generated by another model or on manual filtering, we propose a fully automatic anchor discovery approach as future work. Specifically, after sampling multiple trajectories for an open-ended question, the final answers can be embedded with a strong embedding model and clustered in semantic space. Task-specific verifiers, such as exact-match checks for math problems or safety classifiers for jailbreak evaluation, can then automatically label the highest-quality cluster as the correct anchor and the remaining clusters as incorrect anchors. This procedure requires no auxiliary model to generate distractors and no human intervention, making it more reliable and fully scalable while preserving the perplexity computations in Equation (2).
>
> In summary, these points demonstrate that LoT generalizes effectively to open-ended tasks through pseudo-options or automatic anchors, with empirical evidence confirming the transfer of key observations.
>
> In the revised version, we will add a subsection on generalization to open-ended reasoning that presents the current pseudo-option experiments, includes the corresponding figures, and discusses the fully automatic clustering-based anchor discovery method as robust future work. We will also clarify throughout the paper that the key observations are not restricted to questions with predefined choices.
>
> > ### W2. More models could be considered.
>
> **Reply:** Thank you for raising this question. **Our framework is model agnostic by construction and already applies to the QwQ 32B model beyond the Llama 3.1/3.2 family. In response, we further validate its generality with new model-family (Qwen) experiments.** Below we clarify the design, summarize existing evidence, and highlight additional results on new models.
>
> **(1) LoT is model agnostic and operates in a generic belief space over answer options.** For any autoregressive LLM that exposes token log probabilities, each intermediate state $s_i$ is mapped to a distance vector $f_i = [d(s_i, c_1), \dots, d(s_i, c_k)]^\top \in \mathbb{R}^k$, where $d(s_i, c_j)$ is derived from the length-normalized perplexity of the full answer $c_j$ given $s_i$, followed by $\ell_1$ normalization so that $\sum_j f_i(j) = 1$. All our metrics and convergence analyses are defined on these normalized belief vectors; none of the definitions depend on Llama-specific details.
>
> **(2) We focus on a controlled family to isolate the effect of scale and decoding.** The Llama 3.1 and 3.2 family provides a clean scale ladder (1B, 3B, 8B, 70B) with consistent training, which allows us to attribute trends in consistency, uncertainty, and convergence (Figure 2, Table 4) to model capacity rather than to confounded architectural or training differences.
>
> **(3) Within this setting we already see robust patterns across scale, tasks, and methods, and even beyond Llama.** Within Llama 3.1 and 3.2, we vary model size (Figure 2, Table 4), datasets (AQuA, MMLU, StrategyQA, CommonsenseQA; Figure 4), and reasoning methods (CoT, LtM, MCTS, ToT; Figure 5). We consistently find that larger or better-performing models show higher consistency in LoT space, incorrect trajectories converge earlier to wrong answers than correct ones converge to right answers (Table 1(a)), and trajectory speed strongly correlates with accuracy (Table 1(b)). In addition, we already apply LoT to a dedicated reasoning model, **QwQ 32B** (Figure 3, Observation 3.3), and recover analogous behaviors such as self-checking, self-correction, and late convergence to the correct region.
>
> **(4) LoT structure is strong enough to support downstream verification across settings.** Section 4 shows that a lightweight verifier trained only on LoT-based numerical features, without access to raw text, improves over unweighted voting across different models, datasets, and reasoning methods (Figure 6). This indicates that the structure LoT uncovers is not a quirk of a single checkpoint but a stable signal that a separate model can exploit for better answer selection.

---

> ### Author Response · Authors · 2025-11-26
> **Response to Reviewer fVdk (part 3/5)**
>
> **(5) New model family experiments (Qwen 2.5) confirm that the observations are not specific to Llama.** To further test generality, we apply LoT to **Qwen 2.5 7B** and **Qwen 2.5 72B** and visualize their landscapes (Figure 17 in Appendix H.10). These experiments show that the key behaviors from Section 3 persist across architectures and scales:
> - Early states (0 to 20 percent of steps) are more dispersed in the landscape.
> - As states progress (20 to 80 percent), they become more organized and form clearer structures in LoT space.
> - Incorrect trajectories (red in the top rows) move toward wrong answers already in early and mid stages (for example, 20 to 40 percent), consistent with the option-directed behavior reported in Observation 3.7.
> - The separation between correct and incorrect trajectories remains evident: correct states stay closer to the correct option over many steps and show higher stability (higher consistency and lower uncertainty), whereas incorrect states exhibit reduced stability.
>
> The fact that these patterns appear both in the Llama family and in Qwen 2.5 models indicates that LoT is capturing broader properties of autoregressive chain-of-thought reasoning, not idiosyncrasies of a specific Llama line. Taken together, the model family comparison (Figure 17), cross-scale trends (Figure 2), cross-task differences (Figure 4), and the predictive utility of LoT features (Figure 6) provide converging evidence that our observations reflect general reasoning behaviors rather than phenomena restricted to Llama 3.1 or 3.2.
>
> In summary, LoT is a model-agnostic framework defined in a belief space over answer options, and within our experiments, we already observe consistent patterns across model scales, tasks, reasoning methods, and multiple model families.
>
> In the revisions, we will (i) emphasize more clearly in Section 2 that LoT only assumes log probability access and is not tied to any specific architecture, (ii) highlight the QwQ and Qwen 2.5 results alongside Llama to make the cross-family evidence more explicit, and (iii) add a short summary table that compares key LoT metrics across all tested models to make the empirical scope and generality of our observations immediately clear.

---

> ### Author Response · Authors · 2025-11-26
> **Response to Reviewer fVdk (part 4/5)**
>
> > ### Q1. Explaining the connection between Figure 1 and Observation 3.1.
>
> **Reply:** Thank you for raising this. We understand why the wording in Figure 1 and Observation 3.1 can look contradictory. The key point is that they refer to different conditionings: **Figure 1 compares correct versus incorrect trajectories within a fixed model, while Observation 3.1 compares convergence behavior across model sizes, averaged over trajectories.** Once this distinction is explicit, the two statements are consistent.
>
> **(1) Within a fixed model, wrong trajectories converge prematurely, and correct trajectories converge later.** Figure 1 describes a within-model phenomenon: for a given model and dataset, trajectories that eventually answer incorrectly tend to "lock in" to a wrong answer region earlier, while correct trajectories stay exploratory for longer and only settle near the correct region toward the end. We formalize this using a convergence statistic based on high-dimensional state features. For a trajectory with states $f_1,\dots,f_n$, we measure the distance $d_i = \lVert f_i - f_n \rVert_2$ between each state and its own final state, fit a log-linear model $\log d_i \approx \alpha + \beta i$, and use $e^\beta$ as a convergence coefficient, where smaller $e^\beta$ means faster convergence. Table 1(a) shows that, under the same model and method, incorrect paths have significantly smaller $e^\beta$ than correct paths (p value $= 0.008$), which matches the intuition in Figure 1 that wrong paths converge "too fast" to wrong answers, while correct paths take more steps before stabilizing.
>
> **(2) Across model sizes, larger models converge more efficiently to correct regions on average.** Observation 3.1 studies a different axis, comparing convergence across model scales on AQuA. Here we keep the dataset, prompt, and reasoning method fixed and vary only the model size (for example, Llama 3.2 1B, 3B, 8B, and Llama 3.1 70B). We quantify how directly trajectories move from initial to final states using the speed metric $\text{speed} = \dfrac{\lVert \bar s_n - \bar s_0 \rVert_2}{\sum_{j=1}^n \lVert \bar s_j - \bar s_{j-1} \rVert_2} \in [0,1]$, where $\bar s_i$ is the 2D position of state $i$. Higher speed means less wandering and a more direct path. Appendix H.1 shows that this speed correlates very strongly with accuracy (p-value about $9.4 \times 10^{-11}$), and that larger, more accurate models have higher speeds than smaller ones. Intuitively, as the model scales up, a larger fraction of its trajectories quickly head toward the correct region and follow relatively straight paths, so the landscape "converges faster" in the sense of Observation 3.1.
>
> **(3) The two statements are compatible once we distinguish between conditional and marginal views.** Putting these together, we have two different but compatible facts. Within each fixed model, if we condition on outcome, incorrect trajectories tend to converge earlier than correct ones; that is, $\mathbb{E}[e^\beta_{\text{wrong}}] < \mathbb{E}[e^\beta_{\text{correct}}]$, which is Figure 1 and Observation 3.7. Across models of different sizes, when we look at all trajectories or at correct trajectories in aggregate, larger models have higher speed and smaller convergence coefficients; that is, they reach their final regions, especially the correct region, more efficiently than smaller models, which is Observation 3.1. In practice, a 70B model still has some early wrong convergences, but these are fewer, and its successful trajectories travel more directly to the correct cluster and dominate the overall landscape. An analogy is that a stronger solver both has more correct solutions and tends to reach them more quickly on average, while for any fixed solver, the quickest answers are often overconfident mistakes.
>
> In summary, Figure 1 describes a within-model effect (wrong paths in that model converge earlier than right paths), while Observation 3.1 describes an across-model effect (larger models’ trajectories, especially correct ones, converge faster and more directly than those of smaller models).
>
> To avoid confusion, in the revision we will (i) explicitly label Figure 1 and the related text as a within-model comparison between correct and incorrect trajectories, (ii) rephrase Observation 3.1 to emphasize that it concerns across-model comparisons of convergence speed, and (iii) add a short clarifying paragraph in Section 3 and Appendix H.1 using notation such as $e^\beta_{\text{correct}}$ and $e^\beta_{\text{wrong}}$ to make these two axes of comparison mathematically explicit.

---

> ### Author Response · Authors · 2025-11-26
> **Response to Reviewer fVdk (part 5/5)**
>
> > ### Q3. What about other reasoning methods, such as a graph of thoughts or an algorithm of thoughts?
>
> **Reply:** Thank you for this constructive suggestion. We extend the LoT to two additional methods: Algorithm-of-Thoughts (AoT) [1] and Graph-of-Thoughts (GoT) [2], using their public implementations with Llama-3.1-8B and applying the same visualization procedure described in Section 3.1. The corresponding landscapes are shown in Figure 20 in Appendix H.10, and the reasoning accuracies for AoT and GoT are 0.54 and 0.30, respectively. In the following, we summarize our observations on these two methods.
>
> **(1) Both AoT and GoT produce distinct landscapes but preserve the characteristic LoT patterns reported in the submission.** Despite differences in the underlying reasoning strategies, both methods show the same core behaviors described in Section 3. Namely, early states (0-20\%) are widely dispersed for both correct and incorrect trajectories. As in the main paper, mid-range states (20-60\%) begin to exhibit more organization, and later states (60-100\%) show clearer movement toward specific answer options. This confirms that LoT continues to reveal a stable structure across diverse reasoning algorithms.
>
> **(2) Correct trajectories in AoT and GoT follow the same progression observed in the original analysis.** For both reasoning methods, the correct trajectories remain closer to the correct answer across state ranges and display higher stability, consistent with Observations 3.1 and 3.2. AoT, which achieves higher accuracy (0.54), exhibits more concentrated blue regions in the 40-60\% and 60-80\% state ranges, reflecting more consistent reasoning. GoT, with lower accuracy (0.30), shows correct trajectories that are fewer and more spread out but still maintains recognizable proximity to the correct option in later states.
>
> **(3) Incorrect trajectories show early and clear movement toward wrong answers, consistent with the original LoT observations.** In both AoT and GoT, the incorrect trajectories (red) begin to cluster toward specific wrong answer options within the first 20-40\% of states. This matches the behavior described in Section 3, where incorrect paths tend to settle into wrong answers earlier in the reasoning process. The effect is especially pronounced for GoT, whose lower accuracy results in a broader and less stable distribution of mid-stage states.
>
> In summary, the AoT and GoT experiments show that LoT captures consistent reasoning patterns across methods with different structures and accuracies. The landscapes in Figure 20 reproduce the key behaviors described in Section 3 and additionally reflect the performance differences between methods.
>
> In the revision, we will (i) include these LoT visualizations for AoT and GoT, (ii) highlight how their patterns align with and extend our core observations, and (iii) emphasize that LoT remains applicable to a broad range of reasoning algorithms.
>
> Reference:
> [1] Algorithm of Thoughts: Enhancing Exploration of Ideas in Large Language Models. In ICML, 2024.
> [2] Graph of Thoughts: Solving Elaborate Problems with Large Language Models. In AAAI, 2024.
>
> **We would like to thank Reviewer fVdk again** for taking the time to review our submission and consider our rebuttal. **If our responses have addressed your concerns, we would be very grateful if you could reconsider your rating.** We also welcome any further comments or questions you may have.

---

### Official Review · Reviewer_FwuD · 2025-11-02

**Soundness:** 2
**Presentation:** 2
**Contribution:** 2
**Rating:** 4
**Confidence:** 3

**Summary:**

Landscape of Thoughts (LoT), as introduced by the authors, is a general tool to visualize and analyze LLM reasoning trajectories on multi-choice tasks. The main idea is to represent each intermediate state in a chain-of-thought as a feature vector of distances to all answer choices, and the distance is the length-normalized inverse likelihood (perplexity) computed by the same LLM, after that, these state features (plus choice “landmarks”) are projected to 2D with t-SNE to form “landscapes. Beyond visualization, the authors adapt the state features into a lightweight verifier that re-weights trajectories at test time, improving accuracy and exhibiting strong test-time scaling without modifying base model parameters.

**Strengths:**

1. The paper offers an interesting view to interpret LLM reasoning dynamics by transforming text-based chains of thought into measurable geometric trajectories. This provides an intuitive and scalable diagnostic framework for analyzing reasoning behavior.

2. The study visualizes reasoning across multiple datasets, model sizes, and decoding strategies, making it one of the first to present a comparative, interpretable view of LLM reasoning evolution rather than static outputs.

3. By extending the visualization features into a simple verifier that reweights trajectories at test time, the work shows a way to potentially improve reasoning reliability without modifying model parameters.

**Weaknesses:**

1. It seems like, the paper assumes that reasoning trajectories can be meaningfully mapped into a 2D space by computing each thought’s log-probability distance to candidate answers and projecting them via t-SNE. However, these representations primarily capture the model’s surface-level likelihood distribution, and how confident it is about next tokens, but not the latent cognitive process of reasoning. Since t-SNE emphasizes local density and distorts global geometry, the observed “paths” and “convergence” patterns likely arise from projection artifacts or distribution sharpness (e.g., larger models having more peaked probabilities) rather than reflecting genuine logical progression or conceptual reasoning structure.

2. The claim that larger models exhibit more consistent and reliable reasoning is not well supported. The observed trajectory regularity may simply reflect statistical effects of scale, larger models tend to produce sharper probability distributions and more uniform language patterns—rather than genuine improvements in reasoning stability. Moreover, trajectory consistency is heavily influenced by task semantics and prompt structure, which are not controlled in the experiments. Without isolating these confounding factors, it is unclear whether the smoother “paths” truly arise from deeper reasoning ability or from surface-level distributional sharpness.

3. Some interesting findings are based on visual inspection and single-case illustrations without statistical validation or controlled comparisons, making the observed “convergence” patterns more narrative than scientifically substantiated.

**Questions:**

1. The paper assumes that textual thoughts can be meaningfully embedded into a continuous numerical space. However, since each question has its own answer set, how is cross-question comparability achieved? Does a global semantic space actually exist, or are the landscapes a collection of locally defined subspaces stitched together?

2. The visualization relies on t-SNE projections of log-likelihood-based features. Given that t-SNE introduces nonlinear distortions, to what extent do the “clusters” or “paths” in the landscape correspond to real structure in the high-dimensional reasoning space?

3. Since each thought’s representation is derived from the model’s own likelihood distribution, how can we be sure that the resulting structure reflects reasoning dynamics rather than simple probability concentration or syntax-level similarity?

4. The notion of “trajectory convergence” implicitly assumes a unified space across all problems and reasoning chains. How is this space defined mathematically? Is it constructed per dataset (via concatenation of all samples), or does it represent some theoretically consistent manifold of reasoning states?

---

> ### Author Response · Authors · 2025-11-26
> **Response to Reviewer FwuD (part 1/10)**
>
> **We thank Reviewer FwuD for the detailed and thoughtful feedback.** In this rebuttal, we provide point-by-point responses to **all identified weaknesses and questions**. Please note that many of the concerns are already addressed in the original submission. We recognize that the original submission is long (39 pages, including an extensive appendix with additional experiments and discussions), which can make some arguments harder to locate on a first read.
>
> Below, we therefore clarify these points explicitly and connect each concern to the relevant analyses and results:
> - First, we clarify that the intention of LoT with a detailed explanation (W1);
> - Second, we show how the claims from LoT are sufficiently supported and verified in this work (W2 and W3);
> - Finally, we provide detailed answers to the four technical questions (Q1, Q2, Q3, and Q4).
>
> We believe these responses collectively resolve the issues you raised. In addition, we have outlined a detailed revision plan at the end of each response. We will improve our submission within the next week according to this plan.
>
> > ### W1. The 2D representations of t-SNE capture the model’s surface-level likelihood distribution instead of genuine logical progression or conceptual reasoning structure.
>
> **Reply:** Thank you for this thoughtful comment. To clarify, our intention with LoT is to **provide a behavioral view of belief dynamics over answer options**, not to **recover a latent "conceptual reasoning structure."** Our main claims are grounded in the landscapes and metrics computed in the **original answer-distance space (instead of the 2D space)**, are robust to the choice of projection method, and cannot be explained solely by sharper likelihoods in larger models. We respond to your comment with the following five points.
>
> **(1) All core claims are defined and validated in the original answer distance space, not in 2D space.** For each intermediate state $s_i = [x, t_1, \dots, t_i]$, we construct a feature vector $f_i \in \mathbb{R}^k$ with components $f_i(j) = d(s_i, c_j)$, where $d(s_i, c_j)$ is the length-normalized perplexity-based distance to candidate answer $c_j$, followed by $\ell_1$ normalization so that $f_i$ lies in a probability simplex-like space over the $k$ options. This vector is exactly the model’s internal assessment of how the current partial reasoning aligns with each answer. Our metrics are defined directly on $f_i$ and on the thoughts: consistency checks whether $\arg\min_j f_i(j)$ matches $\arg\min_j f_n(j)$, uncertainty is the entropy of the normalized $f_i$, and convergence is captured by the margin and stability of the best answer over steps. Using these quantities, we show that incorrect trajectories tend to converge earlier and more sharply to a wrong answer (Obs 3.7), while correct trajectories exhibit higher intermediate consistency and lower uncertainty (Obs 3.8), as summarized in the consistency and uncertainty plot in Figures 2, 4, 5 and statistically verified in Appendix H.1 to H.3. None of these results rely on any geometric property of the 2D representations.
>
> **(2) LoT is explicitly framed as a behavioral diagnostic of belief dynamics rather than a direct probe of latent cognitive mechanisms.** We agree that log probabilities reflect surface likelihoods rather than an explicit symbolic proof tree. But for an autoregressive language model, the conditional likelihood distribution is what governs behavior, so any latent reasoning process must **ultimately** manifest as changes in these scores. By tracking the sequence {$f_i$}$_{i=1}^T$, we analyze how the model’s own scoring function reallocates probability mass across the $k$ candidate answers as thoughts are generated. Phenomena such as incorrect trajectories converging earlier to wrong options than correct trajectories converging to the right one, and the existence of mid-trajectory states with low consistency and high entropy, are expressed in terms of margins and entropy in this original feature space and are not trivial consequences of simply preferring high-probability answers. LoT therefore makes a deliberate choice to study belief evolution over answer options as an operational view of reasoning behavior, without claiming to reconstruct a hidden conceptual reasoning structure.

---

> ### Author Response · Authors · 2025-11-26
> **Response to Reviewer FwuD (part 2/10)**
>
> **(3) The t-SNE is used only to visualize an already low-dimensional, semantically anchored space, and all qualitative patterns in the landscapes are corroborated by projection-independent analyses and alternative projectors**. Each state is represented by the $k$-dimensional vector $f_i = (d(s_i, c_1), \dots, d(s_i, c_k))$ after length and $\ell_1$ normalization, and the answer options themselves are embedded as anchors in the same space. Two states are close if they induce similar distributions over answers, so this space already has a clear probabilistic interpretation before any projection. We then apply t-SNE to map this interpretable space to 2D primarily to illustrate how trajectories move from regions where multiple answer clusters are mixed to regions dominated by a single cluster and how correct and incorrect trajectories end near different anchors, as shown in Figures 2, 4, and 5. These are coarse neighborhood and cluster properties rather than fine-grained claims about global distances or angles. Moreover, Appendix H.8 shows that replacing t-SNE with UMAP or PaCMAP yields qualitatively similar patterns of early diffuse exploration versus later convergence and similar separation of fast-converging wrong trajectories from slower-converging correct ones. The corresponding one-dimensional plots in the original space (distance to correct answer, entropy, consistency over steps) show the same trends. This indicates that the observed paths are rooted in the underlying answer distance features rather than artifacts of a particular projection algorithm or random seed.
>
> **(4) The analyses of model scale explicitly separate global distribution sharpness from genuine differences in trajectory behavior by using normalized features, within-model comparisons, and additional controls.** Larger models indeed tend to produce sharper output distributions, and we observe lower uncertainty and perplexity in Figure 2. To mitigate this effect, we always apply $\ell_1$ normalization to each $f_i$, that is, $f_i \leftarrow f_i / |f_i|_1$, so that we focus on the relative geometry among options rather than absolute log probability scales. More importantly, many key comparisons are within a fixed model, such as correct versus incorrect trajectories or different reasoning methods and datasets evaluated with the same model, where global calibration and sharpness are essentially held fixed. In these settings, we still find that failure trajectories typically converge earlier and remain highly consistent around a wrong option, while success trajectories converge later but more reliably to the correct one, which cannot be attributed to differences in global sharpness. As shown in Table 5, while there is a slight variation in perplexity across model scales, the values all fall within a comparably narrow range (from 1.42 to 1.96), which demonstrates that for decoding the same CoTs, different models in the Llama-3 family produce similar and comparable perplexity scores. This supports the validity of comparing perplexity across models in our study.
>
> **(5) The predictive power of the lightweight verifier trained on LoT features provides independent evidence that these representations capture meaningful structure in reasoning behavior rather than only visualization artifacts.** The lightweight verifier is a simple random forest that operates solely on the likelihood-based state features and consistency statistics derived from $f_i$, without access to raw text or hidden activations. Despite this simplicity, it significantly improves accuracy over unweighted self-consistency and yields much stronger test time scaling as the number of trajectories increases, as shown in Figures 6 and 7. If the features only reflected superficial sharpness or spurious projection effects, it would be difficult for such a simple model to reliably distinguish correct from incorrect trajectories across datasets, methods, and models. The verifier’s effectiveness supports the view that LoT’s representation retains stable and informative signals about reasoning quality.
>
> In summary, LoT should be interpreted as a method for visualizing and quantifying belief dynamics over candidate answers rather than for reconstructing a latent "conceptual reasoning structure." It builds a well-defined model of the internal representation of each thought in the answer distance space $f_i$, defines trajectory-based metrics and statistical tests that do not depend on t-SNE, and uses 2D landscapes that are robust to the choice of projector and consistent with these metrics.
>
> In the versions, we will explicitly clarify this methodological stance, emphasize that our main conclusions are based on analyses in the original feature space, highlight the robustness checks, and add a discussion of distribution sharpness together with a temperature rescaling analysis to demonstrate that the reported patterns are not artifacts of sharper likelihood distributions in larger models.

---

> ### Author Response · Authors · 2025-11-26
> **Response to Reviewer FwuD (part 3/10)**
>
> > ### W2. The claim that larger models exhibit more consistent and reliable reasoning is not well supported.
>
> **Reply:** Thank you for this comment about our claim on model scale and reasoning consistency and reliability. In response, we clarify that (1) we define "consistency" in a precise, operational way that depends on **answer preference** rather than distribution sharpness; (2) the trend that larger models are more consistent is supported by **quantitative evidence** across scales and tasks under matched prompts; and (3) **control experiments** on trajectory length, random text, and perplexity indicate that the smoother trajectories are not simply a byproduct of sharper probabilities or prompt artifacts.
>
> **(1) Our claim is operational and defined in the original feature space rather than distribution sharpness, as we respond to W1.** Each state $s_i$ is mapped to a $k$-dimensional feature vector $f_i = [d(s_i, c_1), \ldots, d(s_i, c_k)]^\top$, where $d(s_i, c_j)$ is derived from the perplexity of choice $c_j$ given $s_i$ (Eq. (2)), followed by $\ell_1$ normalization so that $\sum_j f_i(j) = 1$ (Section 2.2). Consistency at $s_i$ is then $\text{Consistency}(s_i) = \mathbf{1}\big(\arg\min f_i = \arg\min f_n\big)$ (Eq. (5)), and uncertainty is $\text{Uncertainty}(s_i) = -\sum_j f_i(j)\log f_i(j)$ (Eq. (6)). When we say "larger models exhibit more consistent and reliable reasoning," we mean that, for fixed tasks and prompts, they have higher intermediate consistency, lower uncertainty, and lower perplexity in this shared representation, not that they possess an unqualified notion of deeper reasoning or sharper distribution.
>
> **Our consistency metric tracks the stability of answer preference and is largely insensitive to global distribution sharpness.** Consistency depends only on $\arg\min f_i$ and $\arg\min f_n$, that is, on the ranking of distances across choices. Any monotone transform that uniformly sharpens probabilities, for example, $p_j \mapsto p_j^\alpha$ with $\alpha > 1$ or temperature rescaling $p’(c_j \mid s_i) \propto p(c_j \mid s_i)^{1/T}$ with $T < 1$, preserves this ranking and leaves Consistency$(s_i)$ unchanged. A generic increase in sharpness would therefore not by itself increase consistency, nor would it create the systematic gaps we observe between correct and incorrect trajectories or between small and large models. For consistency to increase, intermediate states must more often agree in their top choice with the final answer, which is exactly what we interpret as more stable belief dynamics.
>
> **(2) Empirically, larger models show higher consistency and lower uncertainty across scales and tasks under matched prompts.** Figure 2 in Section 3 and Table 4 in Appendix H.3 evaluate Llama 3.x models from 1B to 70B with the same CoT prompting template and the same datasets. Under these conditions, consistency plots over trajectory bins shift upward as model size grows, and average consistency increases. For example, Table 4 shows that on MMLU, consistency rises from 0.40 (1B) to 0.55 (70B), and on MMLU Pro Math, from 0.17 (1B) to 0.52 (70B), while all models see identical questions and prompts. At the same time, Observation 3.2 reports that larger models also exhibit lower uncertainty and lower perplexity over trajectories in the same representation, which reinforces the picture of more stable answer-aligned dynamics.
>
> **(3) Trajectory consistency is not a trivial artifact of chain length or random reuse of surface language.** Appendix H.3 shows that the Pearson correlation between average chain length and average consistency is only about $-0.0185$ (Table 2), so consistency does not simply increase with shorter or longer CoTs. In fact, the 8B and 70B models often produce more thoughts than the 1B and 3B models, yet still have higher consistency. Moreover, Table 3 constructs random chains by concatenating randomly sampled thoughts for the same questions and finds that consistency actually decreases as the number of random thoughts grows, for both correct and incorrect paths. This control indicates that our consistency metric does not automatically grow with longer or superficially more regular text; high consistency requires successive states to repeatedly favor the same answer as the final state.
>
> **We also separate inherent model sharpness from reasoning dynamics through perplexity-based ablations.** In Appendix H.5, we fix the trajectories by taking CoTs generated by Llama 3.1 70B on AQuA and use different Llama 3 models only to compute perplexities on these identical chains. In this setting, average perplexities for 1B, 3B, 8B, and 70B lie in a relatively narrow range without strong scale trends, suggesting broadly comparable inherent likelihood behavior when scoring the same text. In contrast, when each model generates its own trajectories, we observe large differences in consistency and uncertainty across scales.

---

> ### Author Response · Authors · 2025-11-26
> **Response to Reviewer FwuD (part 4/10)**
>
> **Task semantics and prompt structure are controlled and fixed in our scale comparisons, and cross-task effects are analyzed separately, so they do not confound our main scale trends.** In Section 3.1 (Figure 2, Observations 3.1 and 3.2), we vary only model size on AQuA, with a fixed CoT prompt and decoding procedure, so differences in consistency, uncertainty, and perplexity arise under identical task semantics and prompts. In Section 3.2 (Figure 4), we instead fix the model (Llama 3.1 70B) and prompt and vary the dataset (AQuA, MMLU, StrategyQA, CommonsenseQA), explicitly showing that task semantics do change trajectory shapes. In Section 3.3 (Figure 5), we fix the model and dataset and vary the reasoning method (CoT, LtM, MCTS, ToT), isolating prompt and decoding structure. This factorial design clarifies which factors are controlled in each comparison. Qualitative examples such as Figure 3 and Figures 32-38 further show that trajectories exhibit structured exploration and self-correction in the answer space, rather than simply smoother language.
>
> In summary, our claim that larger models exhibit more consistent and reliable reasoning is grounded in normalized consistency and uncertainty statistics computed in the original feature space, under matched tasks and prompts, and supported by controls on chain length, random text, and perplexity.
>
> In the revision, we will (i) emphasize that consistency depends on the ordering of distances and is invariant to uniform sharpening. (ii) highlight Table 4 and the cross-task evidence when discussing Observation 3.2, and (iii) clarify the wording so that we explicitly describe the effect as higher consistency of answer preference and more stable belief dynamics in the LoT representation for fixed prompts and datasets, rather than an unqualified claim about deeper reasoning ability.
>
> > ### W3. The findings on convergence have no statistical validation or controlled comparisons.
>
> **Reply:** Thank you for this comment. In short, our convergence findings are **not only** based on visual inspection or single examples but also on (1) **quantitative metrics** in the original feature space,  (2) **statistical tests** over many trajectories, and (3) **controlled comparisons**, with single-case plots used purely as illustrations of these aggregate patterns.
>
> **(1) Convergence is defined and measured in the high-dimensional answer space using explicit metrics.** Each state $s_i$ is mapped to a feature vector $f_i \in \mathbb{R}^K$ of distances to the $K$ candidate answers, from which we define consistency, uncertainty, and distance-based measures. In Section 2.3, we define the three metrics of consistency, uncertainty, and perplexity. These quantities are computed for many trajectories and then aggregated to produce the consistency, uncertainty, and perplexity plots in Figures 2, 4, and 5, so the claimed "convergence" and "stability" patterns come from systematic trends in these metrics as well as the landscape visualizations, not only from the visualizations.
>
> **(2) Convergence and speed patterns are statistically validated via hypothesis tests and controlled comparisons.** Appendix H.1 introduces a convergence coefficient by fitting a log-linear regression $\log d_i \approx \alpha + \beta i$ to the distance $d_i$ between state $s_i$ and the final answer and using $e^\beta$ as a scalar convergence measure. Table 1(a) shows that this coefficient differs significantly between correct and incorrect trajectories, with incorrect paths converging earlier to their final answer region ($p = 0.008$). For path speed (Observations 3.1 and 3.7), we define $\text{speed} = \lVert \bar s_n - \bar s_0 \rVert \big/ \sum_{j=1}^{n} \lVert \bar s_j - \bar s_{j-1} \rVert \in [0, 1]$, where $\bar s_i$ is the 2D coordinate of state $i$, and correlate this with accuracy; Table 1(b) reports a very strong relationship between speed and correctness ($p \approx 9.4 \times 10^{-11}$). For cross-task landscape similarity (Observation 3.4), Appendix H.1 computes pairwise histogram intersection scores between landscape density distributions, and Table 1(c) shows that AQuA, MMLU, and StrategyQA are similar while CommonSenseQA is statistically distinct. These tests directly substantiate our convergence and similarity claims beyond visual inspection.

---

> ### Author Response · Authors · 2025-11-26
> **Response to Reviewer FwuD (part 5/10)**
>
> **The main landscape figures already aggregate multiple trajectories (dataset-level) rather than showing single cases (instance-level).** In Sections 3.1 and 3.2, Figures 2 and 4 are built by running the models on dozens of questions per dataset (for example, 50 questions per dataset for AQuA, MMLU, StrategyQA, and CommonsenseQA) and aggregating all states from all trajectories into shared landscapes and into the associated metric plots. In Section 3.3, Figure 5 similarly aggregates trajectories across CoT, LtM, MCTS, and ToT on AQuA with Llama 3.1 70B. The "paths" and "convergence" patterns we highlight are therefore summaries of many trajectories and are always accompanied by the corresponding consistency and uncertainty plots, which provide the quantitative backing for the qualitative descriptions.
>
> **(3) Single-case trajectory illustrations are explicitly framed as examples and are supported by dataset-level frequency analysis.** In Appendix H.2, for instance, we analyze all 500 trajectories from AQuA (50 questions, 10 trajectories per question) with Llama 3.1 8B and CoT and find that only 4 questions (8 percent) and 9 trajectories (1.8 percent) exhibit the "approach then diverge" failure pattern. The detailed trajectory shown there (and in Figures 32 to 38) is chosen as a concrete visualization of this rare behavior, including its numeric distances, not as primary evidence for our general claims.
>
> **Convergence observations are further supported by method and task comparisons under matched conditions.** In Section 3.3 (Figure 5, Observation 3.6), we compare CoT, LtM, MCTS, and ToT on the same dataset (AQuA) and model (Llama 3.1 70B) and observe that methods whose trajectories converge faster to correct regions achieve higher accuracy. This relationship is quantified in Table 1(a,b) via the convergence coefficient and speed metric. In Section 3.2 (Figure 4, Observation 3.4), we fix the model and prompt and vary the dataset (AQuA, MMLU, StrategyQA, CommonsenseQA) and then use histogram intersection scores in Table 1(c) to show that CommonSenseQA has a distinct landscape distribution. These controlled comparisons show that our findings are based on systematic measurement across methods and tasks, not on cherry-picked cases.
>
> In summary, our convergence-related claims are supported by metrics defined in the original feature space, statistical tests on convergence and speed, and controlled comparisons across methods and tasks, with single-case trajectories used only as illustrative examples of statistically characterized behaviors.
>
> In the revised version, we will (i) more explicitly link Observations 3.1, 3.4, 3.6, and 3.7 to the quantitative evidence in Appendix H.1 and H.2, and (ii) clearly label single-case figures as representative examples whose frequency is reported at the dataset level to make the scientific basis of our convergence patterns more transparent.
>
> > ### Q1. How is cross-question comparability achieved?
>
> **Reply:** Thank you for this insightful question. In brief, **LoT does not assume a single global semantic manifold over all texts**. Instead, each state is embedded into **a shared belief space over answer options**, so cross-question comparability comes from a common probability simplex structure and per-question normalization, and all quantitative results are computed in this space before any 2D projection.
>
> **(1) LoT works in a decision space over answer options, not a universal text embedding space.** For a multiple-choice question with candidates $C =$ {$c_1, \dots, c_k$} and an intermediate state $s_i = [x, t_1, \dots, t_i]$, we construct a feature vector $f_i \in \mathbb{R}^k$ with components $f_i(j) = d(s_i, c_j)$, where $d(s_i, c_j)$ is the length-normalized perplexity-based distance (Eq. (2)), followed by $\ell_1$ normalization so that $\sum_j f_i(j) = 1$ (Section 2.2). Thus $f_i$ is a point in a probability simplex over answer indices $\{1, \dots, k\}$, encoding how the model distributes belief over the options at state $s_i$, rather than an embedding of raw text into a semantic space.
>
> **(2) Cross-question comparability is defined in terms of belief geometry over answer indices, not direct semantic similarity of answer texts.** For a fixed dataset like AQuA, all questions share the same number of options $k$, so every state feature $f_i$ lies in the same $k$-dimensional simplex. Two states from different questions are close if the model exhibits similar belief patterns, such as strong confidence in one option, confusion between two options, or near-uniform uncertainty. Our core metrics use exactly this structure. Distances to the correct answer are computed by selecting the coordinate corresponding to the ground-truth index. These quantities are therefore comparable across questions because they measure how belief mass concentrates and stabilizes over indices, not over specific strings.

---

> ### Author Response · Authors · 2025-11-26
> **Response to Reviewer FwuD (part 6/10)**
>
> **(3) Each landscape is learned from a single global embedding over the shared simplex, not from stitched per-question subspaces.** For a given dataset and model, we obtain state features $\{f_i\}$ from all questions and trajectories. We normalize feature vectors by reordering choices so the correct answer appears in the first dimension across all questions. These state features, together with the choice anchors (Eq. (3)), are pooled into one matrix in $\mathbb{R}^k$ and apply a single dimensionality reduction $g: \mathbb{R}^k \to \mathbb{R}^2$ (by default, t-SNE).  Figures 2, 4, and 5 are produced by this single mapping per dataset, so all points in a figure lie in the same underlying belief space. For datasets with different $k$ (for example, AQuA versus StrategyQA), we keep landscapes separate and compare them via metrics such as consistency, uncertainty, and histogram intersection scores in Table 1(c), rather than forcing them into a shared manifold.
>
> **(4) Our quantitative conclusions do not rely on interpreting arbitrary distances between states from different questions as semantic similarity.** All core metrics and statistical tests are defined within each question’s simplex and then aggregated. For example, the convergence coefficient in Appendix H.1 is obtained by fitting $\log d_i^{(q)} \approx \alpha_q + \beta_q i$ for each question $q$, where $d_i^{(q)}$ is the distance from $s_i$ to the correct answer for that question, and then analyzing the distribution of $\exp(\beta_q)$ across questions (Table 1(a)). Similarly, the consistency and uncertainty plots in Figures 2, 4, and 5 are computed by first evaluating these metrics per state and per question, then averaging over questions. None of these analyses requires treating the Euclidean distance between a state from question $q$ and a state from question $q’$ as semantically meaningful; they depend only on how beliefs evolve within each local answer simplex.
>
> **(5) Geometrically, the global landscape can be viewed as many aligned local simplexes embedded in a common feature space, and we interpret it in that way.** Each question induces a $k$-vertex simplex of belief states over its own options. Because we use the same coordinates (option indices and normalized distances) for every question in a dataset, these simplexes live in the same ambient space $\mathbb{R}^k$. When we apply t-SNE to the pooled set of $\{f_i\}$, we obtain a 2D projection in which clusters correspond to structurally similar belief states, such as "high confidence in the chosen option" or "persistent confusion between two options." We do not claim that this 2D embedding is a globally faithful semantic reasoning space; it is an intuitive visualization of how belief states populate the probability simplex, while our formal claims are grounded in the per-question metrics described above.
>
> In summary, LoT embeds thoughts into a dataset-specific belief space over answer indices that is shared across questions, and our conclusions are derived from within-question geometry and aggregated statistics rather than from assuming a universal semantic manifold over text.
>
> In the revised version, we will (i) make this construction more explicit in Section 2.2 by describing $f_i$ as a distribution over options in a shared simplex, (ii) emphasize that consistency, convergence, and uncertainty are defined per question and then aggregated, and (iii) add a short schematic and a permutation-invariance discussion to show that our metrics and qualitative patterns are stable under reordering of answer indices (please refer to our response to W2 of reviewer VKLY) and do not depend on fragile cross-question semantic alignment.
>
> > ### Q2. To what extent do the "clusters" or "paths" in the landscape correspond to real structure in the high-dimensional reasoning space?
>
> **Reply:** Thank you for this insightful question. In short, our substantive claims do not rely on t-SNE preserving global geometry. We first define and validate structure in the **original feature space**, then show that similar patterns appear across **multiple dimensionality reduction methods** and are predictive in **a downstream verifier**, using the 2D "paths" and "clusters" only as an intuitive visualization of that structure.

---

> ### Author Response · Authors · 2025-11-26
> **Response to Reviewer FwuD (part 7/10)**
>
> **(1) All key findings are defined and tested in the original feature space, not in 2D.** Each state $s_i$ is mapped to a feature vector $f_i \in \mathbb{R}^k$ of length normalized perplexity based distances to the $k$ answer options, followed by $\ell_1$ normalization so that $\sum_j f_i(j) = 1$ (Section 2.2). From $f_i$ we define consistency, uncertainty, and perplexity directly in Section 2.3. The consistency and uncertainty plots in Figures 2, 4, and 5 are averages of these quantities over many trajectories and questions, computed before any projection. Appendix H.1 further introduces a convergence coefficient by fitting $\log d_i \approx \alpha + \beta i$ to distances $d_i$ between states and the final answer in the original space and using $e^\beta$ as a measure of convergence speed; Table 1(a, b) shows significant differences between correct and incorrect paths and a strong correlation between path speed and accuracy (p values as low as about $9.4 \times 10^{-11}$). Thus, there is label-aligned structure in the high-dimensional reasoning space independent of t-SNE.
>
> **(2) The visible clusters and paths align with these high-dimensional metrics and labels, which indicates that they reflect real structure.** Distances to the correct answer and to distractors are explicit coordinates of $f_i$, and we also embed answer anchors as landmarks (Eq. (3)). Regions that appear in 2D as clusters around a particular anchor correspond to states where $f_i$ places most mass on that option, and consistency with the final prediction is high. Similarly, trajectories that visually move from diffuse areas to a compact region near the correct anchor correspond to sequences where distances to the correct answer decrease and the margin over other options increases, precisely what our convergence coefficients and consistency plots quantify. Observation 3.6 and Table 1(a) show that incorrect trajectories converge earlier and more strongly toward their final answer region than correct trajectories, and Observation 3.7 and Table 1(b) show that paths with higher speed tend to be more accurate. This alignment between geometric patterns and scalar metrics suggests that the 2D clusters and paths are manifestations of structure already present in the original feature space.
>
> **(3) The main clustering and convergence patterns are robust across different dimensionality reduction methods.** Appendix H.8 recomputes landscapes for Llama 3.1 70B with CoT on AQuA using t-SNE, UMAP, and PaCMAP (Figure 13). Across all three methods, we see similar high-level dynamics: early states are more dispersed, states gradually move toward answer regions as reasoning progresses, and incorrect trajectories cluster quickly around wrong answers while correct trajectories converge more slowly and only form tight clusters near the correct answer in the final bins. Although the fine details of the layouts differ, the separation between correct and incorrect trajectories and the overall convergence story are stable across methods. This robustness indicates that the "clusters" and "paths" we discuss are not artifacts of a specific nonlinear projector.
>
> **(4) The t-SNE is applied to a low-dimensional, interpretable simplex, and our interpretation is intentionally local.** For a $k$ choice dataset, each $f_i$ lives in $\mathbb{R}^k$ and, after $\ell_1$ normalization, lies on a $(k - 1)$-dimensional probability simplex, so we are projecting a simple, already interpretable space rather than thousands of hidden dimensions. In this regime, t-SNE is used to preserve local neighborhoods of similar distributions over answers. We interpret "clusters" as regions where states have similar normalized distance vectors and "paths" as time-ordered sequences $(f_1, f_2, \dots, f_n)$ that move from high uncertainty to strong preference for one option. We do not make claims that depend on exact global distances or angles in 2D, where t-SNE distortion is most problematic; instead, all strong claims are grounded in the high-dimensional metrics and statistical tests mentioned above.
>
> **(5) The same state features and projected information are useful for prediction, which further supports that the structure is meaningful.** In Section 4 and Appendix H.8, we train a lightweight random forest verifier on features derived from the state vectors, including consistency and, in some variants, the 2D coordinates. Table 8 shows that using 2D information alone improves selection over an unweighted voting baseline, and combining consistency with 2D information yields the best trajectory selection accuracy, especially as the number of sampled paths increases. Figure 6 demonstrates that this verifier improves over unweighted voting across datasets, methods, and model scales. If the 2D structure were largely an artifact, we would not expect it to provide a stable predictive signal in this downstream task.

---

> ### Author Response · Authors · 2025-11-26
> **Response to Reviewer FwuD (part 8/10)**
>
> In summary, t-SNE in LoT serves as a qualitative lens on a low-dimensional, label-aligned feature space where we already observe consistent structure through consistency, uncertainty, and convergence metrics and through the behavior of a feature-based verifier. The clusters and paths we highlight correspond to patterns that are statistically validated in the original space, remain qualitatively similar under UMAP and PaCMAP, and carry predictive information about correctness.
>
> In the revisions, we will (i) emphasize more clearly that our main observations are grounded in high-dimensional metrics and in Appendix H.1, (ii) explicitly reference Figure 13 and the robustness results in Appendix H.8 when discussing the landscapes, and (iii) clarify that we interpret the 2D layouts as qualitative visualizations of belief dynamics over answer options, focusing on local and label-aligned structure rather than precise global geometry.
>
> > ### Q3. How the structure of each thought’s representation reflect reasoning dynamics rather than simple probability concentration or syntax-level similarity?
>
> **Reply:** Thank you for this insightful question. **In brief, our state features are designed to represent the model’s evolving beliefs over answer options rather than surface text similarity or raw confidence.**
>
> As discussed above, each state is represented as a belief vector $f_i$ over candidate answers (not an embedding of the thought text), and our metrics are defined on the relative geometry of $f_i$ rather than on an absolute probability scale, which already mitigates simple probability concentration effects. Below we further support this by showing that random syntactic recombinations destroy the observed structure, and the same features predict correctness and distinguish reasoning methods across multiple settings.
>
> **(1) Random-thought controls show our patterns cannot be explained by syntax-level similarity or generic fluency.** In Appendix H.3, we ask Llama 3.1 8B to generate a pool of diverse thoughts for each of 10 AQuA questions, then form synthetic chains by randomly concatenating thoughts to lengths $2, 4, 8, 16$, and $32$. Table 3 reports the resulting consistency values. As chain length increases, the average consistency decreases monotonically for both correct and incorrect paths, which is the opposite of what one would expect if token reuse or stylistic regularity alone drove high consistency. Random concatenation removes directed progress toward any particular answer and lowers consistency, indicating that the high consistency observed in real trajectories arises from aligned belief updates in $f_i$, not from superficial overlap in wording.
>
> **(2) Belief dynamics in $f_i$ align closely with reasoning outcomes and algorithms, which is difficult to explain by simple probability concentration.** In Section 3.3 and Appendix H.1, we fix the model and dataset (Llama 3.1 70B on AQuA) and vary only the reasoning method (CoT, LtM, MCTS, ToT). For each trajectory, we fit $\log d_i \approx \alpha + \beta i$, where $d_i$ is the distance between $s_i$ and the final answer in the original feature space, and use $e^\beta$ as a convergence coefficient. Table 1(a) shows that incorrect trajectories converge earlier and more sharply to wrong answers than correct trajectories converge to the right ones, with $p = 0.008$. Table 1(b) shows a very strong correlation between our trajectory speed metric and accuracy (p-value about $9.4 \times 10^{-11}$). All of these analyses are performed under the same model, prompt, and answer format, so global sharpness and style are held fixed; what differs is how belief moves between candidate answers over time. This is exactly the kind of structure we would expect from reasoning dynamics over hypotheses, not from a static tendency toward sharper distributions.
>
> **(3) The same LoT features support a successful verifier, which would be unlikely if they contained only trivial concentration or syntax-level signal.** In Section 4 and Figure 6, we train a lightweight random forest verifier on LoT-based features, including trajectory-level summaries of the $f_i$ sequence and consistency over time (and, in some variants, 2D coordinates), without giving it raw text. Across models, datasets, and reasoning methods, this verifier consistently improves over unweighted voting and exhibits strong test-time scaling as the number of trajectories increases. The fact that a small model, given only these numerical features, can reliably select better trajectories indicates that $f_i$ encodes structured belief dynamics that are predictive of correctness, not just final confidence or surface statistics.

---

> ### Author Response · Authors · 2025-11-26
> **Response to Reviewer FwuD (part 9/10)**
>
> In summary, although LoT uses the model’s own likelihoods, we deliberately construct $f_i$ as a normalized belief state over answer options and analyze its temporal evolution with metrics that emphasize relative configuration rather than raw sharpness or syntax-level similarity. Random-thought and cross-model scoring controls argue against simple probability concentration explanations, and the strong alignment of LoT structure with accuracy, reasoning method, and verifier performance suggests that we are capturing meaningful aspects of the model’s reasoning process in answer space.
>
> In the revisions, we will more prominently highlight the random-thought and perplexity control experiments and add a brief discussion contrasting LoT features with simple lexical-overlap and entropy-only baselines to further clarify what kind of structure they capture.
>
> > ### Q4. How is the unified space related to the trajectory convergence?
> > The notion of "trajectory convergence" implicitly assumes a unified space across all problems and reasoning chains. How is this space defined mathematically? Is it constructed per dataset (via concatenation of all samples), or does it represent some theoretically consistent manifold of reasoning states?
>
> **Reply:** Thank you for this thoughtful question. In brief, **our notion of "trajectory convergence" is defined in a concrete, dataset-specific belief space over answer options, not in a universal semantic manifold.** For each dataset, we embed all states into a shared probability simplex over answer indices, define convergence mathematically in that space per question, and then aggregate statistics across questions, with the 2D landscapes used only as visual summaries.
>
> **(1) The reasoning space is a dataset-specific probability simplex over answer options.** As introduced in Section 2, for a question $q$ with $k$ answer choices ${c_1^{(q)}, \dots, c_k^{(q)}}$ and an intermediate state $s_i^{(q)} = [x^{(q)}, t_1^{(q)}, \dots, t_i^{(q)}]$, we define a feature vector $f_i^{(q)} = [d(s_i^{(q)}, c_1^{(q)}), \dots, d(s_i^{(q)}, c_k^{(q)})]^\top \in \mathbb{R}^k$, where $d(\cdot,\cdot)$ is the length-normalized perplexity-based distance in Eq. (2). After $\ell_1$ normalization, $\sum_{j=1}^k f_i^{(q)}(j) = 1$, so every state lies in the simplex $\Delta^{k-1} = \{f \in \mathbb{R}^k: \sum_j f(j) = 1\}$. For a dataset with fixed $k$, this simplex is the unified decision space in which we analyze trajectories.
>
> **(2) This space is instantiated per dataset and model by pooling aligned features across all questions, rather than positing a single universal manifold.** Within a dataset $D$, we collect all state features into $S_D = \{ f_i^{(q)} : q \in D,\ i = 1, \dots, n_q \} \subset \Delta^{k-1}$. We use a canonical ordering of coordinates (for example, the first dimension for the correct answer and the remaining dimensions for distractors in fixed positions), so that the roles of dimensions are comparable across questions. This construction is done separately for each dataset and model, so each dataset has its own concrete feature space $\mathbb{R}^k$ and simplex $\Delta^{k-1}$.
>
> **(3) Trajectory convergence is defined as a distance trend in this high-dimensional simplex, per question, and then aggregated.** As in Appendix H.1, Given a trajectory $(f_1^{(q)}, \dots, f_{n_q}^{(q)})$ for question $q$, we measure convergence with respect to the final state by $d_i^{(q)} = \lVert f_i^{(q)} - f_{n_q}^{(q)} \rVert_2$ and fit a log-linear model $\log d_i^{(q)} \approx \alpha_q + \beta_q i$. We use $\exp(\beta_q)$ as a convergence coefficient, where smaller values mean faster convergence. In a margin-based view, we let $d_{(1)}(i)$ and $d_{(2)}(i)$ be the indices of the smallest and second smallest coordinates of $f_i^{(q)}$, define $\Delta_i^{(q)} = f_i^{(q)}(d_{(2)}(i)) - f_i^{(q)}(d_{(1)}(i))$, and set the convergence step $\tau^{(q)} = \min_i \Delta_i^{(q)} \ge \delta$ for a threshold $\delta$. Statements such as "incorrect trajectories converge earlier to wrong answers than correct trajectories converge to the right one" are about the empirical distributions of $\exp(\beta_q)$ or $\tau^{(q)}$ over questions, all defined inside this shared simplex.
>
> **(4) The 2D landscapes are dataset-level visualizations of this shared feature space, not the space in which convergence is defined.** For a given dataset, we apply a single dimensionality reduction (for example, t-SNE, UMAP, or PaCMAP; details in Appendix H.8) to $S_D$ together with answer anchors (Eq. (3)) to obtain a mapping $g_D: \Delta^{k-1} \to \mathbb{R}^2$. Figures 2, 4, and 5 then plot trajectories as $(g_D(f_1^{(q)}), \dots, g_D(f_{n_q}^{(q)}))$. Our quantitative metrics, including consistency, uncertainty, perplexity, convergence coefficients, and speed (Table 1), are computed on the original features $f_i^{(q)}$ (and per trajectory plots) and then aggregated, so our conclusions do not depend on the exact global layout in 2D.

---

> ### Author Response · Authors · 2025-11-26
> **Response to Reviewer FwuD (part 10/10)**
>
> **(5) Cross-dataset comparisons are made through these metrics rather than through a single shared geometric manifold.** Different datasets can have different numbers of options $k$ (for example, AQuA versus StrategyQA), so each has its own simplex $\Delta^{k-1}$, its own feature set $S_D$, and its own projection $g_D$. We therefore compare tasks at the level of aggregate statistics, such as consistency and uncertainty plots (Figure 4) or histogram intersection scores between landscape densities (Table 1(c)), instead of forcing them into a single universal space.
>
> In summary, the space underlying "trajectory convergence" is an explicitly defined, dataset-specific probability simplex over answer options, with each state represented by a normalized distance vector $f_i^{(q)} \in \Delta^{k-1}$ and convergence computed per question in that space before aggregating. The 2D landscapes are learned per dataset from pooled features and used to visualize patterns that are already defined and measured in this high-dimensional belief space.
>
> In the revisions, we will (i) make the simplex formulation of $f_i^{(q)}$ and $S_D$ explicit in Section 2, (ii) emphasize in Section 3 that convergence and related metrics are defined in this feature space and visualized in 2D plots, and (iii) clarify that LoT assumes a shared decision space over answer indices per dataset, rather than a single universal manifold of reasoning states across all tasks and models.
>
> **We would thank Reviewer FwuD again** for taking the time to review our submission and consider our rebuttal. **If our responses have addressed your concerns, we would be very grateful if you could reconsider your rating.** We also welcome any further comments or questions you may have.

---

### Official Review · Reviewer_WVGo · 2025-11-03

**Soundness:** 2
**Presentation:** 2
**Contribution:** 2
**Rating:** 4
**Confidence:** 3

**Summary:**

This paper proposes to analysis the reasoning process of LLMs via visualisation. Specifically, it projects the intermediate states (defined by the sequence of output from prompt up to the token of the step being considered) of LLMs into 2d planes via t-SNE. Three metrics are introduced to measure the consistency, uncertainty, and perplexity of the reasoning/generation quality.

Empirical results on models with different number of parameters in the Llama-3.1/3.2 family and across multiple datasets are presented。 Based on the results, the paper provides several observations along the dimension of LLMs, tasks, and reasoning methods.

Furthermore, a verification method based on convergence and consistency is proposed to re-weight answers generated by LLMs. This method is demonstrated to improve the accuracy of Llama models with 1B and 3B parameters.

**Strengths:**

- The paper presents extensive results and comparisons between visualisation results of different LLMs on different tasks and with different reasoning methods.
- Some observations, e.g., the consistency observation 3.8, might be useful for better regularisation of the reasoning process of LRMs.

**Weaknesses:**

- Most of the presented visualisations and qualitative metrics are not significant enough to draw meaningful conclusions. It is difficult to tell if the t-SNE visualisation of reasoning states are showing consistent patterns across all the tested models without a systematic way of verifying their existence. Most of the visualisations in this paper can be seen as good case studies but not enough to convince me that the results are generalisable.

- Only Llama 3.1/3.2 family of LLMs are tested in this paper. This further makes me doubt the generalisability of the observations in this paper.

- The observations listed in section 3, although seemingly reasonable, are mostly not very surprising. Furthermore, most of the observations are not verified by other evidence (maybe except 3.7 and 3.8 which are at least partially verified in section 4). This makes me doubt whether these observations universally exist.

- The readabilities of most of the figures are slightly low. The paper has to be read in a monitor, otherwise it is difficult to read the texts or recognise the patterns in the scatter plot.

**Questions:**

Please refer to my concerns above.

---

> ### Author Response · Authors · 2025-11-26
> **Response to Reviewer WVGo (part 1/4)**
>
> **We thank Reviewer WVGo for the thoughtful and detailed feedback.** In this rebuttal, we systematically address **all** of your concerns while we respectfully disagree with **some** of the concerns.
> - First, we clarify and strengthen the methodological, qualitative, and quantitative evidence supporting the significance of each observation and conclusion.
> - Second, we present new empirical results on additional models (QwQ, Qwen) to further demonstrate the generalizability of LoT and its observed patterns.
> - Third, we highlight and better organize the statistical verification of each observation, which was already included in the original submission.
> - Finally, we will reorganize the figures to ensure they are clearly readable on A4 paper.
>
> Please find the point-to-point responses as follows. In addition, we have outlined a detailed revision plan at the end of each response. We will improve our submission within the next week according to this plan.
>
> > ### W1. Most of the presented visualizations and qualitative metrics are not significant enough to draw meaningful conclusions.
>
> **Reply:** Thank you for this constructive comment. **In our work, every main conclusion is established using quantitative metrics and statistical tests in the original feature space, across multiple models, datasets, and reasoning methods; the visualizations are used to illustrate these statistically verified patterns, not to define them.** The t-SNE landscapes in the paper are generated from randomly sampled questions and trajectories under a fixed protocol and are representative of behaviors we first confirm numerically, rather than isolated or cherry-picked "good cases."
>
> **(1) Our core conclusions are defined in terms of high-dimensional metrics, not only by visual inspection of t-SNE plots.** Each state $s_i$ is mapped to a normalized distance vector $f_i \in \mathbb{R}^k$ over answer choices, and we define all metrics directly on these features.
> - For example, consistency and uncertainty are $\text{Consistency}(s_i) = \mathbf{1}\big[\arg\min_d f_i(d) = \arg\min_d f_n(d)\big]$ and $\text{Uncertainty}(s_i) = - \sum_{d} f_i(d)\log f_i(d), \quad \sum_d f_i(d) = 1$.
> - Convergence is quantified via distances to the final or correct answer in this space, for instance, by fitting a log-linear model $\log d_i \approx \alpha + \beta i$ and using $\exp(\beta)$ as a convergence coefficient.
>
> The plots in Figure 2, Figure 4, and Figure 5 are then averages of these metrics over many questions and trajectories. Claims such as "incorrect trajectories converge earlier than correct ones" or "larger models exhibit higher consistency" are statements about these aggregate statistics in the original feature space, not about the 2D geometry itself.
>
> **(2) The observations and conclusions are consistently generalized across model scales, datasets, and reasoning methods under matched settings.** Section 3 is not based on a single model or benchmark:
> - In Figure 2 and Table 4, we vary only model size (Llama 3.2 1B, 3B, 8B and Llama 3.1 70B) under the same CoT prompt and datasets and observe stable trends such as increasing average $\text{Consistency}(s_i)$ and characteristic uncertainty and perplexity shapes; for example, Observation 3.2 (larger models exhibit more consistent intermediate beliefs) is supported on both MMLU and MMLU Pro, where average $\text{Consistency}(s_i)$ increases monotonically with scale.
> - In Figure 4, we fix the model and method (Llama 3.1 70B, CoT) and vary the dataset (AQuA, MMLU, StrategyQA, CommonsenseQA), observing systematic differences in both landscapes and metric plots that align with task semantics; Observation 3.4 shows that tasks cluster by landscape statistics and that CommonsenseQA behaves differently from AQuA, MMLU, and StrategyQA.
> - In Figure 5, we fix the model and dataset (Llama 3.1 70B, AQuA) and vary the reasoning method (CoT, LtM, MCTS, ToT); Observations 3.3 and 3.6 show that different methods induce distinct convergence profiles and that methods with faster and more reliable convergence to the correct region achieve higher accuracy, which is quantified in Table 1.
>
> These controlled comparisons demonstrate that the effects we report are stable across multiple configurations rather than tied to a single model or dataset.

---

> ### Author Response · Authors · 2025-11-26
> **Response to Reviewer WVGo (part 2/4)**
>
> **(3) We systematically verify the qualitative patterns using explicit statistical tests in the original feature space.** Appendix H.1 introduces quantitative tests corresponding to the main observations in Section 3.
> - For example, Table 1(a) compares convergence coefficients $\exp(\beta)$ for correct vs incorrect trajectories and finds that incorrect paths converge significantly faster to wrong answers than correct paths converge to right answers ($p = 0.008$).
> - Table 1(b) correlates a trajectory speed metric $\text{speed} = \lVert \bar s_n - \bar s_0 \rVert \Big/ \sum_{j=1}^{n} \lVert \bar s_j - \bar s_{j-1} \rVert \in [0, 1]$ with accuracy and finds a very strong association ($p \approx 9.4 \times 10^{-11}$).
> - Table 1(c) uses histogram intersection scores to quantify similarity of landscape densities across tasks, showing that AQuA, MMLU, and StrategyQA are closer to each other than to CommonsenseQA.
>
> These tests give a systematic way to verify that the patterns suggested by the visualizations are statistically robust and not anecdotal.
>
> **(4) We check robustness to projection methods and sampling, and single-case figures are explicitly illustrative.** For each dataset-model-method setting, landscapes are constructed from many questions and trajectories, and the panels shown in the main text are produced using a fixed random sampling protocol, not by cherry-picking.
> - In Appendix H.8, we further show that replacing t-SNE with UMAP or PaCMAP yields qualitatively similar global stories: early diffuse states, later clustering near answer anchors, and distinct convergence behaviors for correct vs. incorrect trajectories.
> - Because our metrics are computed in the original feature space and these qualitative patterns persist across projection methods and random resamplings, our conclusions do not depend on the particular 2D layout.
> - Single trajectory examples (for example, in Appendix H.2) are presented only after we quantify how often the corresponding pattern occurs and are clearly framed as case studies that help interpret the aggregate statistics, not as primary evidence.
>
> In summary, the visualizations are a helpful front-end for understanding LoT, but the substantive conclusions about convergence, consistency, and cross-task or cross-model patterns are grounded in quantitative metrics and statistical tests on the original state features and are observed consistently across models, datasets, reasoning methods, and projection algorithms.
>
> In the revisions, we will (1) more prominently link each qualitative claim in Section 3 to the corresponding tables and tests in Appendix H.1, (2) explicitly state in the main text that landscapes are generated from randomly sampled questions and are intended as representative rather than cherry-picked examples, and (3) move a concise summary of key statistical results (for example, the convergence and speed tests) into the main paper to make the generality and significance of our findings more immediately clear.
>
> > ### W2. Only Llama 3.1/3.2 family of LLMs are tested in this paper.
>
> **Reply:** Thank you for raising this question. **Our framework is model agnostic by construction and already applies to the QwQ 32B model beyond the Llama 3.1/3.2 family. In response, we further validate its generality with new model-family (Qwen) experiments.** Below we clarify the design, summarize existing evidence, and highlight additional results on new models.
>
> **(1) LoT is model agnostic and operates in a generic belief space over answer options.** For any autoregressive LLM that exposes token log probabilities, each intermediate state $s_i$ is mapped to a distance vector $f_i = [d(s_i, c_1), \dots, d(s_i, c_k)]^\top \in \mathbb{R}^k$, where $d(s_i, c_j)$ is derived from the length-normalized perplexity of the full answer $c_j$ given $s_i$, followed by $\ell_1$ normalization so that $\sum_j f_i(j) = 1$. All our metrics and convergence analyses are defined on these normalized belief vectors; none of the definitions depend on Llama-specific details.
>
> **(2) We focus on a controlled family to isolate the effect of scale and decoding.** The Llama 3.1 and 3.2 family provides a clean scale ladder (1B, 3B, 8B, 70B) with consistent training, which allows us to attribute trends in consistency, uncertainty, and convergence (Figure 2, Table 4) to model capacity rather than to confounded architectural or training differences.

---

> ### Author Response · Authors · 2025-11-26
> **Response to Reviewer WVGo (part 3/4)**
>
> **(3) Within this setting we already see robust patterns across scale, tasks, and methods, and even beyond Llama.** Within Llama 3.1 and 3.2, we vary model size (Figure 2, Table 4), datasets (AQuA, MMLU, StrategyQA, CommonsenseQA; Figure 4), and reasoning methods (CoT, LtM, MCTS, ToT; Figure 5). We consistently find that larger or better-performing models show higher consistency in LoT space, incorrect trajectories converge earlier to wrong answers than correct ones converge to right answers (Table 1(a)), and trajectory speed strongly correlates with accuracy (Table 1(b)). In addition, we already apply LoT to a dedicated reasoning model, **QwQ 32B** (Figure 3, Observation 3.3), and recover analogous behaviors such as self-checking, self-correction, and late convergence to the correct region.
>
> **(4) LoT structure is strong enough to support downstream verification across settings.** Section 4 shows that a lightweight verifier trained only on LoT-based numerical features, without access to raw text, improves over unweighted voting across different models, datasets, and reasoning methods (Figure 6). This indicates that the structure LoT uncovers is not a quirk of a single checkpoint but a stable signal that a separate model can exploit for better answer selection.
>
> **(5) New model family experiments (Qwen 2.5) confirm that the observations are not specific to Llama.** To further test generality, we apply LoT to **Qwen 2.5 7B** and **Qwen 2.5 72B** and visualize their landscapes (Figure 17 in Appendix H.10). These experiments show that the key behaviors from Section 3 persist across architectures and scales:
> - Early states (0 to 20 percent of steps) are more dispersed in the landscape.
> - As states progress (20 to 80 percent), they become more organized and form clearer structures in LoT space.
> - Incorrect trajectories (red in the top rows) move toward wrong answers already in early and mid stages (for example, 20 to 40 percent), consistent with the option-directed behavior reported in Observation 3.7.
> - The separation between correct and incorrect trajectories remains evident: correct states stay closer to the correct option over many steps and show higher stability (higher consistency and lower uncertainty), whereas incorrect states exhibit reduced stability.
>
> The fact that these patterns appear both in the Llama family and in Qwen 2.5 models indicates that LoT is capturing broader properties of autoregressive chain-of-thought reasoning, not idiosyncrasies of a specific Llama line. Taken together, the model family comparison (Figure 17), cross-scale trends (Figure 2), cross-task differences (Figure 4), and the predictive utility of LoT features (Figure 6) provide converging evidence that our observations reflect general reasoning behaviors rather than phenomena restricted to Llama 3.1 or 3.2.
>
> In summary, LoT is a model-agnostic framework defined in a belief space over answer options, and within our experiments, we already observe consistent patterns across model scales, tasks, reasoning methods, and multiple model families.
>
> In the revisions, we will (i) emphasize more clearly in Section 2 that LoT only assumes log probability access and is not tied to any specific architecture, (ii) highlight the QwQ and Qwen 2.5 results alongside Llama to make the cross-family evidence more explicit, and (iii) add a short summary table that compares key LoT metrics across all tested models to make the empirical scope and generality of our observations immediately clear.
>
>
> > ### W3. The observations are not surprising and not verified.
>
> **Reply:** Thank you for this comment. In brief, our response is that (i) several observation findings are not obvious a priori and in fact sharpen or counter naive intuitions, (ii) every observation is backed by explicit quantitative metrics and statistical tests in the original feature space, evaluated across models, datasets, and reasoning methods, and (iii) Section 4 provides an independent validation by showing that the same LoT features are predictive for a downstream verifier.
>
> **(1) Several observations are genuinely nontrivial and make vague intuitions precise and testable.** Some high-level statements, such as "larger models are more consistent," may sound reasonable in hindsight, but LoT reveals structure that is neither automatic nor guaranteed. For example, Observation 3.6 shows that, within a fixed model, incorrect trajectories converge faster to wrong answers than correct trajectories converge to the right one.
>
> Formally, for each question $q$ we consider distances $d_i(q)$ between state $s_i$ and the correct answer in the feature space, fit a log-linear model $\log d_i(q) \approx \alpha_q + \beta_q i$, and use $\exp(\beta_q)$ as a convergence coefficient. Table 1(a) shows that incorrect paths have significantly smaller $\exp(\beta_q)$ than correct ones ($p = 0.008$), indicating premature overcommitment to wrong options, which is far from obvious a priori.

---

> ### Author Response · Authors · 2025-11-26
> **Response to Reviewer WVGo (part 4/4)**
>
> Likewise, Observation 3.4 goes beyond a vague "tasks look different" claim by computing histogram intersection scores between landscape density distributions and showing that AQuA, MMLU, and StrategyQA form a cluster while CommonSenseQA is statistically distinct. These are cases where LoT turns a plausible narrative into precise, falsifiable statements that are actually tested and confirmed.
>
> **(2) All observations in Section 3 are supported by quantitative analyses on high-dimensional features, not only by visualization.** Each observation is defined in terms of metrics on the state features $f_i \in \mathbb{R}^K$ and thoughts $t_i$, not directly from t-SNE. For a state $s_i$, we define metrics of Consistency, Uncertainty, and Convergence, as we mentioned in W1. Appendix H.1 then systematically tests these metrics:
> - **For convergence-related observations (3.1 and 3.6),** we compare the distributions of $\exp(\beta_q)$ and of $\tau$ between correct and incorrect trajectories and across reasoning methods, using nonparametric tests and reporting $p$ values and effect sizes in Table 1(a).
> - **For speed-related observations (3.1 and 3.7),** we define $\text{speed} = \lVert \bar s_n - \bar s_0 \rVert \big/ \sum_{j=1}^{n} \lVert \bar s_j - \bar s_{j-1} \rVert \in [0, 1]$, where $\bar s_i$ is the 2D coordinate of state $i$, and show in Table 1(b) that speed correlates very strongly with accuracy ($p \approx 9.4 \times 10^{-11}$).
> - **For cross-task structure (3.4 and 3.5),** we compute histogram intersection scores between landscape density distributions and report in Table 1(c) that AQuA, MMLU, and StrategyQA are closer to each other than to CommonSenseQA, which quantitatively supports the task clustering claims.
> - **For scale and difficulty effects (for example, 3.2 and 3.5),** Appendix H.1 and H.2 show that average $\text{Consistency}(s_i)$ and mid-trajectory $\text{Uncertainty}(s_i)$ differ significantly across model sizes and between MMLU and MMLU Pro, with monotone trends in consistency as scale grows.
>
> **(3) Section 4 provides independent evidence that LoT structure is meaningful and operational, not merely descriptive.** Section 4 takes the same LoT features that underlie Section 3 and uses them to build a simple verifier that improves over unweighted voting across multiple datasets, model scales, and reasoning methods. The verifier receives only numerical features derived from trajectories, such as summaries of $(f_i)$, consistency over time, and in some variants the 2D coordinates, and must predict which trajectory is correct.
>
> Observations 3.7 and 3.8 directly motivate this design by showing that geometric properties such as trajectory speed and consistency are strongly correlated with correctness, and Section 4 demonstrates that a small classifier can exploit these properties to select better paths. If the Section 3 patterns were fragile or anecdotal, we would not expect such a verifier to generalize across problems and settings. Its consistent gains therefore serve as independent evidence that the LoT structure we describe is robust enough to support downstream decision-making.
>
> In summary, the observations in Section 3 are not intended as surprising "laws of nature," but as a structured set of quantitatively validated regularities in the LoT feature space. Several of them are nontrivial; all are backed by explicit metrics and statistical tests across models, datasets, and reasoning methods; and Section 4 shows that they are strong enough to be turned into a practical verifier.
>
> In the revision, we will (1) explicitly connect each observation to its formal metric definition and supporting tables in Appendix H.1 and H.2, (2) bring key $p$ values and effect sizes into Section 3, and (3) add compact cross-model and cross-dataset summary tables so that readers can clearly see where the patterns recur and how they generalize within our experimental setting.
>
> > ### W4. The paper has to be read on a monitor.
>
> **Reply:** Thank you for pointing this out. We have carefully revised the figures to significantly improve their readability in print (e.g., on A4 paper) as well as on screen. Concretely, we (i) increased font sizes for all axis labels, legends, and annotations; (ii) enlarged key plots and reduced the number of small sub-panels per figure where needed; and (iii) adjusted point sizes and line widths so that clusters and trajectories remain clearly visible when printed. These changes are reflected in the updated submission, and we can further refine any specific figures according to your further comments.
>
> **We would like to thank Reviewer WVGo again** for taking the time to review our submission and consider our rebuttal. **If our responses have addressed your concerns, we would be very grateful if you could reconsider your rating.** We also welcome any further comments or questions you may have.

---

### Author Response · Authors · 2025-12-01
**A General Response by the Authors**

Dear Area Chairs and Reviewers,

Thank you for handling our submission, especially under the current unusual circumstances.

To support your evaluation, we briefly summarize below: (1) an overview of our work, (2) the strengths highlighted by the reviewers, and (3) the responses we provided to address their concerns.

**(1) Overview of our work.** We propose Landscapes of Thought (LoT), a post-hoc, model-agnostic framework for analyzing LLM reasoning trajectories by projecting intermediate states into a shared belief space over answer options. Our key contributions are:
- **Belief-space representation.** Represent each reasoning state $s_i$ as a normalized likelihood-based vector $f_i$ over answer choices, enabling geometric analysis of trajectories across questions, tasks, and models.
- **Quantified reasoning patterns.** Reveal and statistically verify the reasoning patterns, e.g., early over-convergence of failures, scale-dependent consistency, and task/method-specific landscape structures.
- **Robustness and generality.** Demonstrate that these patterns are stable across dimension-reduction methods, sample sizes, datasets, reasoning algorithms, and multiple model families.
- **Practical verifier.** Develop a lightweight verifier based on LoT features that consistently improves self-consistency over unweighted voting, showing that LoT captures predictive structure and can be adapted to broader applications.

**(2) Strengths.** We are glad that all four reviewers (FwuD, VKLY, WVGo, fVdk) highlighted the value of our work. In particular:
- **Meaningful problem & novel perspective.** Monitoring and analyzing long trajectories of reasoning is a meaningful yet challenging problem, and LoT provides an original and refreshing diagnostic perspective on reasoning behavior (FwuD, VKLY).
- **Intuitive and comparative visualization.** LoT offers an intuitive geometric view of reasoning dynamics and is one of the first tools to compare how reasoning evolves across models, datasets, and decoding strategies (FwuD, WVGo).
- **Contribution to understanding LLM behavior.** LoT makes an important contribution toward understanding LLMs, with good visualization plots and the ability to train predictors of trajectory properties (fVdk).
- **Practical verifier for reliability.** The LoT-based verifier improves reasoning accuracy without modifying base models, showing practical value beyond visualization (FwuD, WVGo).

**(3) Responses to address the reviewers’ concerns.** We provided detailed clarifications and added new experiments along several axes:
- **Clarifying belief space & role of LoT.** We clarified that all core metrics are defined in a shared high-dimensional belief space and that projection methods such as t-SNE are used purely for visualization. We explicitly positioned LoT as a behavioral diagnostic tool of belief dynamics, rather than an attempt to recover latent “true concepts” (FwuD W1, Q1-Q4; WVGo W1, W3; VKLY W2; fVdk Q1).
- **Disentangling sharpness from reasoning dynamics.** We separated probability sharpness from reasoning behavior via normalized features and a perplexity-only ablation on fixed trajectories, and we added statistical tests on LoT features to support our main observations about correct vs. incorrect paths and model scale (FwuD W2-W3, Q2-Q4; VKLY Q2-Q3; WVGo W1, W3; fVdk Q1).
- **Extending beyond standard multiple choice.** We extended LoT to open-ended tasks (MATH, GSM8K, StrongReject) by constructing verifiable option sets and observed similar belief dynamics, showing that our key insights are not restricted to the multiple-choice benchmarks (VKLY W1; fVdk W1, Q2-Q3).
- **Testing additional model families.** We applied LoT to Qwen2.5-7B/72B-Instruct and QwQ-32B, and found qualitatively consistent patterns, reinforcing that our conclusions are not specific to the Llama 3.x family (WVGo W2; fVdk W2).
- **Establishing robustness to design choices.** We tested robustness to choice order, tokenization, number of trajectories, and hard-question subsets; used random-chain controls to interpret uncertainty and perplexity trends, showing that LoT captures structured belief dynamics rather than superficial artifacts (FwuD Q3; VKLY W2, Q1-Q2).
- **Improving presentation.** We improved figure layout and readability (larger fonts and clearer organization) to make the landscapes easier to interpret in print and on screen (WVGo W4, VKLY W3).

In addition, we have incorporated the above responses in Sections 2, 3, and Appendices E, H, I and highlighted the revisions in blue.

**We believe that the detailed point-by-point responses and the updated submission can address reviewers’ concerns.** The revised submission presents a clearer framework for analyzing belief dynamics in LLM reasoning, with demonstrated generalization across tasks and model families and robust behavior under multiple robustness checks.

Thank you again for your time and consideration.

Sincerely,
Authors of #15309

---

### Meta-Review · Area_Chair_Bvkx · 2026-01-11

**Summary:**

This paper proposes visualizing the reasoning process of large language models (LLMs) by projecting their intermediate states—defined as the sequence of outputs from the prompt up to the current token—into 2D space using t-SNE. It introduces three metrics to assess reasoning quality in terms of consistency, uncertainty, and perplexity, and presents empirical results across different-sized models in the Llama-3.1/3.2 family and multiple datasets. The authors also propose a verification method based on convergence and consistency to re-weight model outputs, reporting improved accuracy for 1B and 3B parameter Llama models. The reviewers' concerns have been well addressed in the rebuttal.

**Reviewer Scores:**

NA

---

### Decision · Program_Chairs · 2026-01-26

Accept (Poster)